# Does Spatial Cognition Emerge in Frontier Models?

Santhosh Kumar Ramakrishnan[*]    Erik Wijmans    Philipp Krähenbühl    Vladlen Koltun

Apple

## Abstract

Not yet. We present SPACE, a benchmark that systematically evaluates spatial cognition in frontier models. Our benchmark builds on decades of research in cognitive science. It evaluates large-scale mapping abilities that are brought to bear when an organism traverses physical environments, smaller-scale reasoning about object shapes and layouts, and cognitive infrastructure such as spatial attention and memory. For many tasks, we instantiate parallel presentations via text and images, allowing us to benchmark both large language models and large multimodal models. Results suggest that contemporary frontier models fall short of the spatial intelligence of animals, performing near chance level on a number of classic tests of animal cognition.

## 1 Introduction

Frontier models have achieved impressive performance in mathematics, coding, general knowledge, and commonsense reasoning (Hendrycks et al., 2021a;b; Chen et al., 2021; Sakaguchi et al., 2021; Yue et al., 2024). This remarkable progress has inspired characterizations of frontier models as possessing the intelligence of a smart high schooler and predictions of the imminent arrival of super-intelligence (Aschenbrenner, 2024). These characterizations are often underpinned by the premise that competence (or even mastery) in some aspects of cognition is symptomatic of broad cognitive competence. This is not self-evident. To quote Brooks's first law of artificial intelligence, "When an AI system performs a task, human observers immediately estimate its general competence in areas that seem related. Usually that estimate is wildly overinflated." (Brooks, 2024).

Our work focuses on spatial cognition, a foundational form of intelligence that is present in a broad spectrum of animals including humans (Marshall & Fink, 2001; Waller & Nadel, 2013; Mallot, 2024). Spatial cognition refers to the ability of animals to perceive and interact with their surroundings, build mental representations of objects and environments, and draw upon these representations to support navigation and manipulation. Decades of research in animal cognition have characterized the spatial cognition of mice, rats, bats, pigeons, corvids, dogs, wolves, elephants, marmosets, tamarins, howler monkeys, baboons, chimpanzees, and humans (Tolman, 1948; Menzel, 1973; Peters, 1974; Gillner & Mallot, 1998; Marshall & Fink, 2001; Noser & Byrne, 2007; Tommasi et al., 2012; Porter & Garber, 2013; Blaser et al., 2013; Geva-Sagiv et al., 2015; Presotto et al., 2019; de Guinea et al., 2021; Payne et al., 2021; Xu et al., 2024; Xavier et al., 2024; Welklin et al., 2024). Human infants already possess rudimentary spatial cognition, which subsequently improves along developmental schedules that have been characterized (Blades & Spencer, 1994; Newcombe, 2000; Vasilyeva & Lourenco, 2012). Spatial cognition is known to underpin more advanced cognitive abilities (Kozhevnikov et al., 2007; Newcombe, 2010; Young et al., 2018).

The emergence of spatial cognition has been linked to embodiment (Smith & Gasser, 2005; Jansen & Heil, 2010; Frick & Möhring, 2016), without which the development of spatial cognition may be impaired (Foreman et al., 1990; Anderson et al., 2013). However, frontier models are typically trained in a disembodied manner on corpora of text, images, and video. Does spatial cognition emerge in disembodied frontier models? To study this question systematically, we develop SPACE, a benchmark that builds on decades of research in cognitive science. Our benchmark comprises two

---

[*]Corresponding author: s_ramakrishnan@apple.com

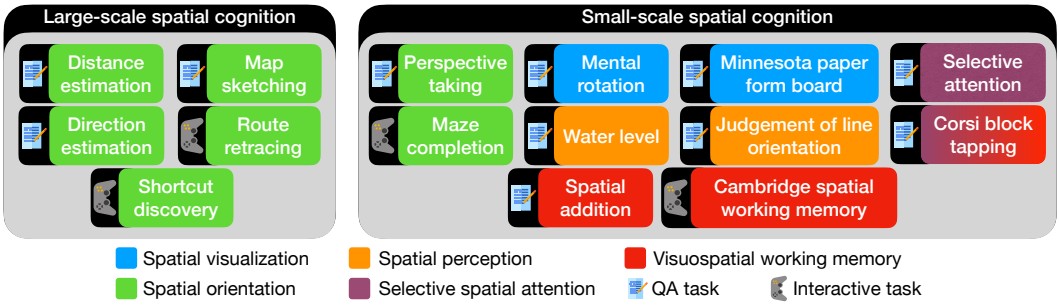

Figure 1: **SPACE: Spatial Perception And Cognition Evaluation.** We design a suite of spatial cognition tasks based on the cognitive science literature. These are broadly classified into large-scale and small-scale spatial cognition. Large-scale tasks require understanding space at the level of environments and evaluate spatial orientation and cognitive mapping abilities. Small-scale tasks require understanding space at the level of objects or object arrangements and evaluate skills such as spatial visualization, spatial orientation, spatial perception, selective spatial attention and visuospatial working memory. These tasks can be multiple-choice question answering, or interactive games. We develop multimodal as well as purely textual presentations, which support evaluation of both large language models (LLMs) and vision-language models (VLMs).

broad classes of tasks, covering large-scale and small-scale spatial cognition (Hegarty et al., 2006; Meneghetti et al., 2022; Newcombe, 2024). See Figure 1 for an overview.

Large-scale spatial cognition has to do with a model's ability to understand its surroundings. In large-scale tasks, the model is familiarized with an environment and is then asked to estimate distances and directions to landmarks, sketch a map of the environment, retrace a known route, or discover a shortcut to a goal. Small-scale spatial cognition has to do with a model's ability to perceive, imagine, and mentally transform objects in two or three dimensions. Together, large-scale and small-scale tasks evaluate core cognitive abilities such as spatial perception, visualization, orientation, selective attention, and visuospatial memory (Lacroix et al., 2021; Meneghetti et al., 2022).

We design text-based and image-based presentations to evaluate both large language-only and vision-language models (LLMs and VLMs, respectively). Our results indicate that contemporary frontier models have not yet reached competency – let alone mastery – in spatial cognition. On key large-scale spatial cognition tasks, frontier multimodal models perform near chance level, even when presented with an allocentric (map) view of the environment. The strongest models exhibit much better performance on some small-scale tasks that evaluate selective spatial attention and visuospatial working memory, especially with purely textual presentations via character arrays, but perform near chance on other tasks such as mental rotation (Vandenberg & Kuse, 1978), perspective taking (Kozhevnikov & Hegarty, 2001), maze completion (Lacroix et al., 2021), or the classic Minnesota Paper Form Board test (Likert & Quasha, 1941; 1969).

## 2 RELATED WORK

**Spatial cognition.** Spatial cognition is a branch of cognitive science that seeks to understand how humans and animals perceive, interpret, represent, and interact with objects and environments (Marshall & Fink, 2001; Landau, 2002; Waller & Nadel, 2013; Mallot, 2024; Newcombe, 2024). This involves the perception of object sizes, shapes, and scales, as well as the relationships between objects and landmarks in the environment (including location, distance, direction, and orientation). Spatial cognition is broadly divided into two categories: large-scale and small-scale (Hegarty et al., 2006; Jansen, 2009; Meneghetti et al., 2022; Newcombe, 2024). *Large-scale spatial cognition* refers to the ability to build spatial representations of environments and use them effectively for navigation and spatial reasoning. Large-scale spatial cognition tasks typically involve egocentric spatial transformations, where the viewer's perspective changes with respect to the environment while the spatial relationships between parts of the environment remain constant (Wang et al., 2014). *Small-scale spatial cognition* refers to the ability to perceive, imagine, and mentally transform objects or shapes in 2D or 3D. This is typically evaluated using paper and pencil tasks that require allocentric spatial transformations of objects and shapes (Wang et al., 2014). While large-scale spatial cognition has been demonstrated in a wide range of animals (Tolman, 1948; Menzel, 1973; Peters, 1974;

O'Keefe & Nadel, 1978; Gillner & Mallot, 1998; Richardson et al., 1999; Geva-Sagiv et al., 2015; Toledo et al., 2020), the study of small-scale spatial cognition is specific to humans.

**Emergent spatial representations.** Several works have shown that spatial representations, a phenomenon similar to spatial cognition, *can* emerge in neural networks (Banino et al., 2018; Cueva & Wei, 2018; Wijmans et al., 2023; Sorscher et al., 2023). These works train a neural network from scratch for path integration or navigation tasks and analyze the model weights to identify spatial representations.

**Spatial reasoning in large language models.** PlanBench (Valmeekam et al., 2024) and CogEval (Momennejad et al., 2023) evaluate LLMs on text-based planning tasks such as navigation, delivery logistics and block stacking to evaluate cognitive mapping and planning. Yamada et al. (2024) evaluate spatial reasoning in LLMs by performing map traversals on different types of graphs and evaluate the model's self-localization ability. EWOK (Ivanova et al., 2024) studies spatial plausibility reasoning in LLMs. In comparison to these benchmarks, SPACE evaluates a broader array of cognitive abilities and implements multimodal presentations of classic animal cognition experiments.

**Benchmarks for large multimodal models.** The recent successes of multimodal models (OpenAI, 2024; Li et al., 2024a; Reid et al., 2024) have been facilitated by large-scale training on text and multimodal corpora (Rana, 2010; Together Computer, 2023; Chen et al., 2023; Laurençon et al., 2023; Gadre et al., 2023), followed by tuning on human preferences (Liu et al., 2023a; Awadalla et al., 2024; Ouyang et al., 2022; Rafailov et al., 2023). The remarkable advances in the capabilities of these models inspired a variety of benchmarks that evaluate their performance. Early multimodal benchmarks consisted of single-task datasets such as visual question answering (Antol et al., 2015; Goyal et al., 2019; Marino et al., 2019) and image captioning (Chen et al., 2015). However, due to the limited scope of early datasets and concerns regarding potential test-data leakage, newer benchmarks use diverse collections of tasks (Fu et al., 2023; Yu et al., 2024; Liu et al., 2023b; Yue et al., 2024; Lu et al., 2024; Ying et al., 2024). While these datasets primarily focus on image understanding, newer datasets that emphasize spatiotemporal reasoning have been proposed for video (Li et al., 2024b; Fu et al., 2024a; Majumdar et al., 2024).

Recent studies highlight a number of shortcomings of frontier multimodal models (Moskvichev et al., 2023; Tong et al., 2024; Chen et al., 2024a; Fu et al., 2024b). One such shortcoming is that models may not perceive the image in detail, often missing fine-grained details or ignoring the image entirely (Chen et al., 2024b; Guan et al., 2024; Tong et al., 2024). HallusionBench proposes a new dataset of image pairs, where tiny edits are made from one image to another that change the answer to the question (Guan et al., 2024). MMVP identifies issues with CLIP-based pretraining of visual encoders, which make current models blind to certain visual patterns, and proposes a benchmark of CLIP-blind image pairs where the same question has opposite answers (Tong et al., 2024). MMStar shows that many questions in multimodal benchmarks can be answered correctly without the image and proposes a new split of existing benchmarks that addresses this issue (Chen et al., 2024b).

Another shortcoming of existing models is their lack of spatial perception and reasoning (Chen et al., 2024a; Cheng et al., 2024). SpatialVLM proposes a VQA dataset that requires answering questions about relative spatial arrangements and metric relationships (Chen et al., 2024a). SpatialRGPT further includes region-level understanding (Cheng et al., 2024). MOCHI evaluates the ability of vision models to identify rotated versions of procedurally-generated objects (Bonnen et al., 2025). 'Perception test' aims to overcome shortcomings of standard video datasets by creating a diagnostic dataset where participants record videos while following complex scripts depicting interesting events (Patraucean et al., 2023). It evaluates fundamental perceptual skills (memory, abstraction, intuitive physics, and semantics) and various types of reasoning.

Another line of work considers skill acquisition (the ability to learn a skill and apply it to new scenarios). Prior work has studied this using visual analogical reasoning (Chollet, 2019; Moskvichev et al., 2023; Yiu et al., 2024). The ARC dataset contains samples consisting of a few examples of abstract grids and their transformations and one or more test inputs (Chollet, 2019). The objective is to understand the transformation performed using the examples and apply it to test inputs. The transformations have been further organized into specific concepts with varying degrees of difficulty in the ConceptARC dataset (Moskvichev et al., 2023). Inspired by ARC and developmental psy-

chology, the KiVA dataset studies visual analogies in the context of visually realistic 3D shapes with concepts like transformations in color, size, rotations, reflections, and counting (Yiu et al., 2024).

# 3 SPACE: A BENCHMARK FOR SPATIAL PERCEPTION AND COGNITION EVALUATION

We develop a benchmark for evaluating the spatial cognition of frontier models. The benchmark comprises large-scale and small-scale tasks and is designed for compatibility with both text-only and multimodal models.

## 3.1 LARGE-SCALE SPATIAL COGNITION

In large-scale spatial cognition tasks, we evaluate the ability of models to build spatial representations of their surrounding environment, and whether they can use these representations to reason about and navigate in the environment. There are two stages to these tasks. First, we familiarize the model with an environment by showing a video walkthrough.[1] The model must build a mental representation of the environment that captures the locations of start, goal and landmark locations, and their spatial relationships. After the model is familiarized with the environment, we evaluate the model's spatial representation using five tasks derived from the cognitive science literature (Meneghetti et al., 2022). See Figure 2(top) and Figure 3 for an overview.

1. **Direction estimation.** The goal is to determine the directions to other landmarks from a given landmark. The participant is asked to pretend that they are facing a landmark A, and then asked to estimate the direction (in degrees) to another landmark B. This is known as a pointing trial in the cognitive science literature (Allen et al., 1996; Hegarty et al., 2006; Pazzaglia & Taylor, 2007; Weisberg et al., 2014; Meneghetti et al., 2016). We formulate this as a multiple-choice QA task with four options for the direction (only one correct option).

2. **Distance estimation.** The goal is to determine the straight-line distances from one landmark to all other landmarks (Allen et al., 1996; Hegarty et al., 2006). The participant is asked to pretend that they are facing a landmark A, and then asked to estimate the Euclidean distance to all the other landmarks. We pose this as a multiple-choice QA with four options for the list of distances to each landmark. Since current models are not good at estimating metric measurements (Chen et al., 2024a; Cheng et al., 2024), we generate incorrect options such that the ratios of distances between landmarks are not preserved, making it easier to identify the correct option.

3. **Map sketching.** The goal is to draw a map of the environment that contains the start, goal and landmark positions (Allen et al., 1996; Hegarty et al., 2006; Pazzaglia & Taylor, 2007; Weisberg et al., 2014; Meneghetti et al., 2016; 2021). We formulate this as multiple-choice QA with four options for the map sketches. The correct option preserves the true spatial relationships between the different map elements, while the incorrect options skew the spatial relationships randomly.

4. **Route retracing.** The goal is to retrace the route shown in the video from the start to the goal (Allen et al., 1996; Pazzaglia & Taylor, 2007; Meneghetti et al., 2016; 2021). This task evaluates the model's ability to remember landmarks seen in the route and the actions required along the route to reach the goal. We formulate this as an interactive task where the model receives the current observation, decides which action to take, and receives updated observations based on the actions taken. We measure performance using the SPL metric (success weighted by path length), which penalizes the model for taking unnecessary detours (Anderson et al., 2018). (The demonstrated route, which the model must retrace, is always the shortest path to the goal.)

5. **Shortcut discovery.** The goal is to discover a shortcut (i.e., a route never observed before) from the start to the goal after observing a video walkthrough that takes detours to reach the goal (Tolman, 1948; Allen et al., 1996; Pazzaglia & Taylor, 2007; Meneghetti et al., 2016; 2021). The ability to discover shortcuts in familiar environments is a key indicator of cognitive mapping ability (Tolman, 1948). When designing environments and walkthrough paths, we ensured that a novel shortcut exists that the model can exploit. Similar to route retracing, we treat this as an interactive navigation task and measure performance using the SPL metric.

---

[1]For text-only models, the 'video walkthrough' is a sequence of discrete map observations presented as arrays of characters, see Figure 3 for examples.

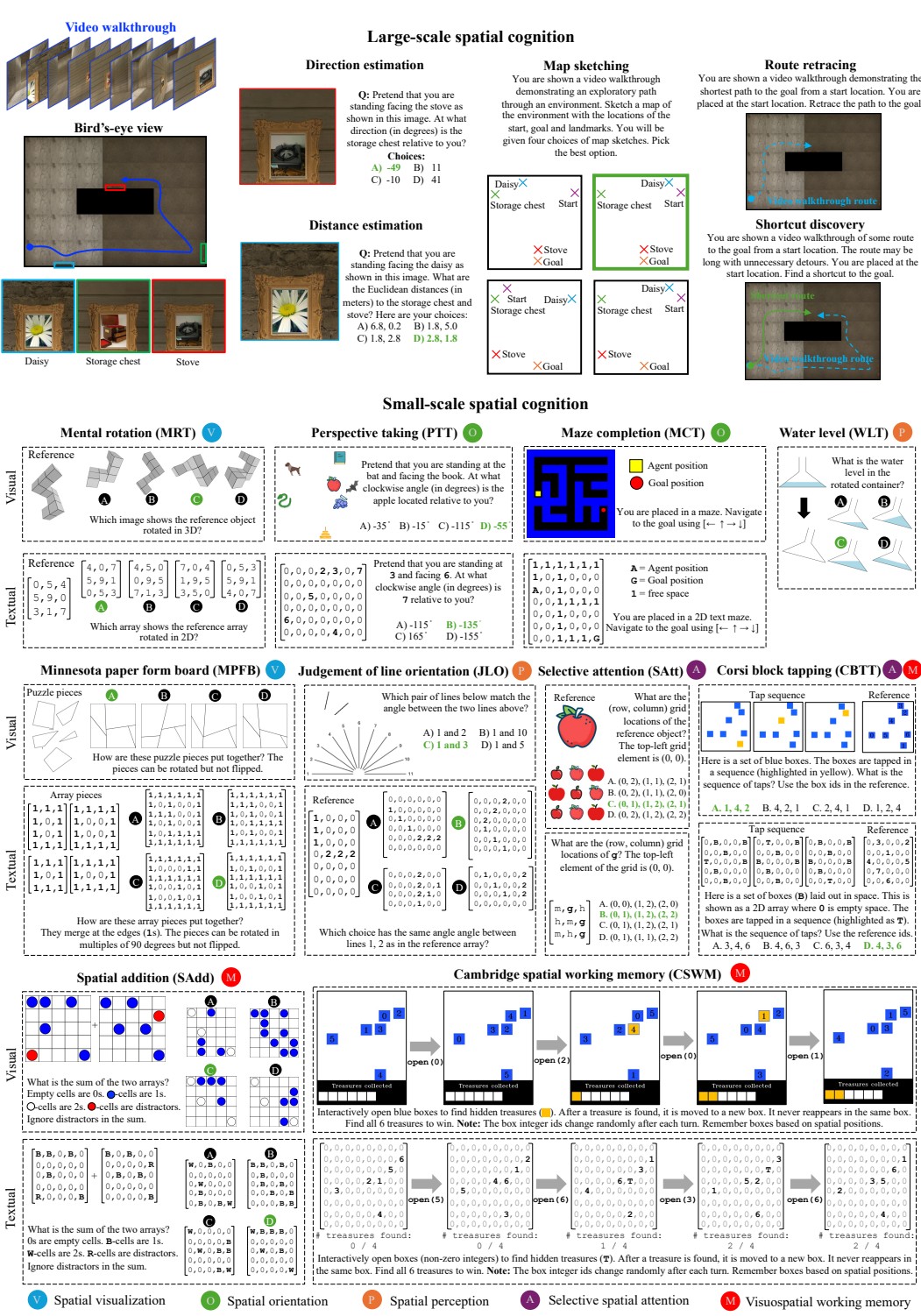

Figure 2: The tasks in SPACE. For all tasks (other than the water level test), we include multimodal as well as purely textual presentations, to support evaluating both large language models (LLMs) and vision-language models (VLMs). For large-scale tasks, we visualize examples from the egocentric image presentation here and visualize alternate presentations in Figure 3. For small-scale tasks, we visualize both visual and textual presentations here. Bolding of characters in the arrays is for illustration purposes only.

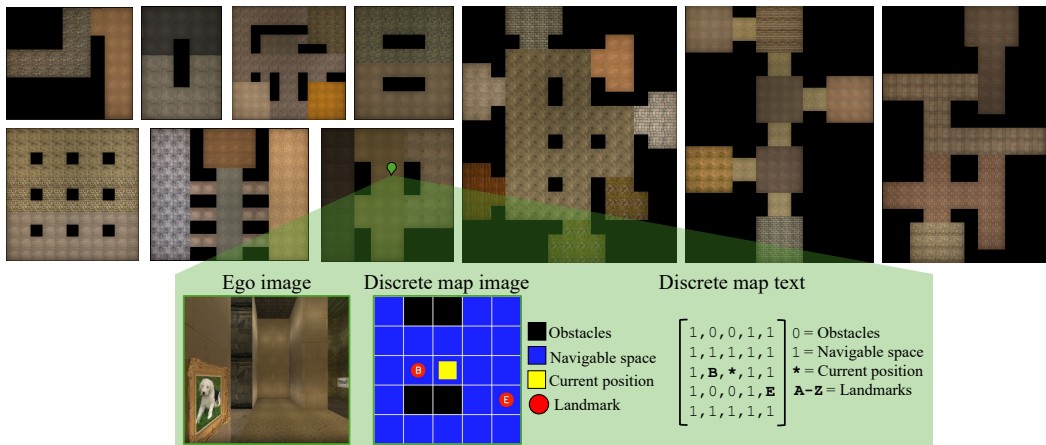

Figure 3: **Large-scale spatial cognition.** We design ten environment layouts based on experimental protocols in cognitive science. The top row shows bird's-eye view renderings of these environments. To evaluate large-scale spatial cognition in frontier models, we implement three observation spaces: egocentric image, discrete map (DM) image, and discrete map (DM) text (see bottom row). **Ego image** shows a first-person view within the environment. **DM image** shows a quantized, allocentric bird's-eye view of the $2.5\text{m} \times 2.5\text{m}$ region centered on the current position. Unlike ego image, DM image enables performing the large-scale tasks in a simplified setting without requiring perspective geometry. **DM text** depicts the DM image using text characters. We evaluate multimodal models using ego image and DM image, and large language models using DM text.

### 3.1.1 IMPLEMENTATION

**3D environment generation.** We create ten environment layouts based on prior work in cognitive science and artificial intelligence (Tolman, 1948; Gillner & Mallot, 1998; Richardson et al., 1999; Banino et al., 2018; Bouchekioua et al., 2021). Figure 3 shows bird's-eye view images of each layout. See the appendix for more details about the environment generation process.

**Observation spaces.** We create multiple observation spaces to support evaluating both text-only and vision+text models. These are egocentric images, discrete map (DM) images, and discrete map (DM) text presentations.

- **Ego image.** The environment is captured using a forward-facing perspective camera placed at the model's location in the environment. This is similar to the setup of an animal navigating through an immersive environment and requires understanding perspective geometry.
- **DM image.** This is a quantized bird's-eye view image of a $2.5\text{m} \times 2.5\text{m}$ area in the environment surrounding the model's location. This is akin to a human using a map to navigate. The current location is always at the center of the DM image. We use a Pacman-like coloring scheme highlighting the obstacles, navigable space, current postion, and landmarks. DM image simplifies the mapping process by removing the need for perspective geometry understanding.
- **DM text.** This is a translation of DM image to an text array. We carefully select the text encoding to ensure compatibility with text tokenizers of popular models and ensure that each element of the array is encoded by the tokenizers of all evaluated models as a distinct token.

See Figure 3 (bottom) for examples of these presentations. The first two observation spaces are used for models that support visual inputs, while the last observation space is used for text-only models. See the appendix for additional illustrations of these tasks and dataset statistics.

### 3.2 SMALL-SCALE SPATIAL COGNITION

In small-scale spatial cognition tasks, we evaluate the models' ability to perceive, imagine, and mentally transform objects or shapes in two and three dimensions. We build on the body of work on visuospatial abilities, which are evaluated in humans via paper-and-pencil tasks (Allen et al., 1996; Weisberg et al., 2014; Meneghetti et al., 2022). These abilities may be used to explain individual differences between participants in large-scale spatial cognition (Meneghetti et al., 2022). We define ten small-scale tasks to evaluate abilities such as spatial perception, spatial visualization, spatial orientation, selective attention, and visuospatial working memory. See Figure 2(bottom) for illustrations of each task. We summarize each task below and provide additional details in the appendix.

6. **Mental rotation test (MRT).** This is a test of spatial visualization, i.e., the ability to mentally manipulate 2D or 3D stimuli (Vandenberg & Kuse, 1978). In the visual presentation, a reference 3D shape from Shepard & Metzler (1971) is provided along with four choices. The correct choice is a rotated version of the reference, and the remaining choices are rotated versions of an alternate shape. The goal is to identify the correct choice from the distractors. The text-only version of this task uses 2D character arrays, akin to the card rotations test from French et al. (1963).

7. **Perspective taking test (PTT).** This is a test of spatial orientation, i.e., the ability to imagine being in a different position in space and seeing the surroundings from a new perspective (Kozhevnikov & Hegarty, 2001). We place $N$ randomly-sampled objects (like apples, bats, dogs, books, grapes, etc.) at random locations in an image (with no overlap between objects). The objective is to take the perspective of standing next to an object (say, a bat) facing another object (say, a book), and determine the relative orientation of a third object (say, an apple). This is a multiple-choice QA with four options (only one correct option).

8. **Water level test (WLT).** This is a test of spatial perception (Piaget et al., 1957). Originally, it was designed to evaluate children's knowledge about the horizontal nature of the surface of water in a sealed bottle regardless of its orientation. Performance on the water-level test was found to be related to performance on spatial ability tests (Foltz, 1978; Wittig & Allen, 1984). We present the model with an image of a water container partially filled with water and ask it to imagine the position of the water if the container were tilted. We implement this as a four-way multiple-choice QA, where each choice is an image showing the tilted container with varying water levels. The objective is to select the one choice that shows the correct water level.

9. **Minnesota Paper Form Board test (MPFB).** This is a test of spatial visualization, where the model must perform multi-step manipulations of complex spatial information (Meneghetti et al., 2022). Specifically, we provide the model with pieces of a figure and ask it to identify how the pieces fit together (Likert & Quasha, 1941; 1969). We programmatically segment a square into five pieces, and rotate the pieces randomly to generate the final segments. We generate alternate segmentations of a square as negative choices for a multiple-choice QA presentation.

10. **Judgement of Line Orientation test (JLO).** This is a test of spatial perception (Benton, 1994), where a model must determine the angle between two lines in an image. Our visual presentation shows two lines in an image along with a set of 11 reference lines. The objective is to determine the pair of reference lines that have the same angles between them as the lines in the image. This is presented as a multiple-choice QA with four choices (only one of them correct). Our text-only presentation implements the tasks via lines embedded in 2D integer arrays.

11. **Selective attention task (SAtt).** This is a test of selective spatial attention, i.e., the ability to selectively attend to a particular region of space while ignoring others (Serences & Kastner, 2014; Pahor et al., 2022). In particular, we use the widely used cancellation task, where the goal is to search for and mark out target stimuli embedded amidst distractors (Della Sala et al., 1992; Brickenkamp & Zillmer, 1998; Dalmaijer et al., 2015; Lacroix et al., 2021; Pahor et al., 2022; Kalina & Walgrave, 2004). We design the task as multiple-choice QA with objects as the stimuli for visual evaluation and characters as stimuli for text-only evaluation. The target stimuli and distractors are arranged on a grid. The answer must be selected from one out of four options. The correct option lists the (row, column) pairs that localize the target stimuli in the grid.

12. **Maze completion task (MCT).** This is an interactive game to evaluate spatial orientation, planning, and executive functioning (Lacroix et al., 2021). We programmatically create mazes using Mazelib (Stilley, 2014) and render them using a Pacman-like color scheme for the visual presentation and a character array for the text-only presentation (similar to DM image and DM text in Figure 3). Using the maze rendering, a model must sequentially select an up/down/left/right action to reach the goal and execute a stop action to successfully complete the task. If the model does not reach the goal within 250 actions, it is considered to have failed. We measure the success rate, i.e., the percentage of mazes where the model reaches the goal within the allotted time.

13. **Corsi block-tapping task (CBTT).** This is a test of visuospatial working memory and attention (Corsi, 1972; Claessen et al., 2015). We create a digital Corsi board with $N$ blue-colored blocks that are randomly placed on the board with no overlap ($N \in [5, 8]$). We randomly sample a sequence of $K$ taps, where each block is tapped at most once ($K \in [4, N]$). The taps are digitally rendered on the blocks by highlighting them in yellow when tapped, yielding an sequence of $K$ images. After presenting the $K$ images, we provide a rendering of the board with integer IDs

assigned to each block and ask the model to reproduce the sequence of taps using these IDs. We treat this as multiple-choice QA with four choices of tap sequences, only one of which is correct.

14. **Spatial addition task (SAdd).** This is a test of visuospatial working memory, i.e., the ability to store and manipulate spatial information in memory (Wechsler, 2009). The model is presented with two 2D grids, where each grid location can be empty or contain a blue or red dot. The objective is to add the two grids together by following certain rules. If a grid location has a blue dot in exactly one of grids, the result should be a blue dot. If a grid location has blue dots on both grids, the result should be a white dot. Red dots are distractors and must be ignored. We programmatically generate grid pairs with sizes sampled from $\{3, 5, 7, 9\}$ and pseudo-randomly populate them with blue and red dots. We formulate the task as multiple-choice QA, presenting four grids as possible answers, exactly one of which is correct.

15. **Cambridge spatial working memory test (CSWM).** This is an interactive game that evaluates visuospatial working memory (Sahakian et al., 1988). The model is presented with an image containing $N$ blue colored boxes ($N \in [3, 7]$). A yellow 'treasure' is initially hidden in one of the boxes. The model must sequentially select boxes one at a time to find the hidden treasure. Once the treasure is found, another treasure is placed in one of the remaining boxes. The objective is to locate all the yellow treasures via a process of elimination. We programmatically generate instances of this task by randomly sampling blue boxes, placing them at random locations (without overlap), and placing the treasures in each box in random order. At each step, we assign random integer IDs to each box as a reference for selecting a box. The boxes' integer IDs are randomized in each step, forcing the model to remember boxes based on their spatial positions. When the model finds a treasure, the box containing the treasure becomes yellow. The model must find all the treasures before a time limit $T$ (determined based on $N$) to succeed.

As with large-scale spatial cognition, we also implement purely textual presentations of these tasks to support evaluation of large language models (LLMs). Figure 2 illustrates both the multimodal and the purely textual presentations. The key idea in instantiating the textual presentations is to encode all spatial information via 2D character arrays. We did not identify a natural such encoding for the water level test (WLT) and did not include a text-only presentation for it for this reason. See the appendix for additional illustrations of these tasks. In some tasks, such as MRT, MPFB, and JLO, the text presentations are substantially easier than the corresponding visual presentations. However, the visual and textual presentations match closely for the remaining tasks, enabling us to identify modality-specific limitations of multimodal models by evaluating them on the two presentations.

## 4 EXPERIMENTS

**Baselines.** We evaluate a number of LLMs and VLMs. Using text-only presentations, we evaluate GPT-4v and GPT-4o (OpenAI, 2023; 2024), Claude 3.5 Sonnet (Anthropic, 2024), the Llama3 family (Dubey et al., 2024), Mistral models such as Mixtral 8x7B, Mixtral 8x22B, and Mistral 123B (Jiang et al., 2024; Mistral AI team, 2024a), and two Yi 1.5 models (Young et al., 2024). Using multimodal presentations, we evaluate GPT-4v and GPT-4o (OpenAI, 2023; 2024), Claude 3.5 Sonnet (Anthropic, 2024), LlaVA-NeXT-Interleave (Li et al., 2024a), Pixtral 12B (Mistral AI team, 2024b), and Phi-3.5-vision (Abdin et al., 2024). We use the vLLM inference engine for evaluating the open-source models (Kwon et al., 2023). For each task, we implement a prompt that provides a detailed description of the task and the expected response format (see the appendix). We also list the results of a chance baseline that selects an answer at random. For multiple-choice QA tasks, chance is at $25\%$. For interactive tasks, the chance baseline samples an action at random in each step. We further include human performance for reference for the multiple-choice QA tasks. See the appendix for additional implementation details.

**Large-scale spatial cognition results.** The results are shown in Table 1, grouped by presentation modality (ego image, DM image, DM text). For image-based presentations, we evaluate Claude 3.5 Sonnet, GPT-4v and GPT-4o because they support video understanding (via a succession of images). For DM text, we evaluate both open and closed LLMs. We also list the performance of the chance baseline for calibration, as well as human performance (see the appendix for details). In the text-only modality, Claude 3.5 Sonnet attains the highest average performance. Mistral 123B is the highest-performing open model. All evaluated models struggle with large-scale spatial cognition, falling significantly below human performance on direction estimation, distance estimation,

**Observation space: Ego image**

| Method | Direction estimation | Distance estimation | Map sketching | Route retracing | Shortcut discovery | Average |
|---|---|---|---|---|---|---|
| Human | 82.8 | 83.2 | 96.6 | – | – | – |
| GPT-4o | 32.0 ±4.1 | 36.5 ±5.0 | 33.3 ±4.1 | 6.6 ±3.6 | 6.4 ±1.0 | 23.0 |
| Claude 3.5 Sonnet | 29.0 ±2.9 | 34.4 ±2.9 | 27.5 ±8.3 | 7.4 ±2.8 | 0.0 ±0.0 | 19.6 |
| GPT-4v | 29.7 ±0.3 | 31.9 ±2.7 | 20.0 ±11.8 | 1.6 ±1.2 | 3.9 ±0.9 | 17.4 |
| Chance | 25.0 | 25.0 | 25.0 | 0.0 | 0.0 | 15.0 |

**Observation space: DM image**

| Method | Direction estimation | Distance estimation | Map sketching | Route retracing | Shortcut discovery | Average |
|---|---|---|---|---|---|---|
| Human | 82.9 | 82.5 | 100.0 | – | – | – |
| GPT-4o | 29.5 ±5.5 | 31.9 ±1.0 | 33.3 ±3.3 | 23.6 ±3.1 | 25.9 ±2.0 | 28.8 |
| Claude 3.5 Sonnet | 32.5 ±2.3 | 40.0 ±2.6 | 30.0 ±4.1 | 15.4 ±4.3 | 13.7 ±6.4 | 26.3 |
| GPT-4v | 26.3 ±3.0 | 29.3 ±4.1 | 45.0 ±5.0 | 13.7 ±5.2 | 15.3 ±3.0 | 25.9 |
| Chance | 25.0 | 25.0 | 25.0 | 0.0 | 0.0 | 15.0 |

**Observation space: DM text**

| Method | Direction estimation | Distance estimation | Map sketching | Route retracing | Shortcut discovery | Average |
|---|---|---|---|---|---|---|
| Human | 66.7 | 76.5 | 66.7 | – | – | – |
| Claude 3.5 Sonnet | 29.2 ±4.4 | 40.2 ±3.1 | 51.7 ±5.5 | 26.5 ±2.9 | 20.0 ±3.0 | 33.5 |
| GPT-4o | 28.7 ±4.1 | 33.3 ±1.7 | 46.7 ±4.1 | 27.5 ±3.2 | 26.6 ±0.1 | 32.6 |
| Mistral 123B | 30.5 ±5.1 | 28.9 ±5.7 | 38.3 ±5.5 | 20.3 ±2.8 | 19.9 ±3.0 | 27.6 |
| GPT-4v | 30.7 ±4.1 | 26.5 ±2.7 | 40.8 ±6.0 | 20.6 ±5.8 | 15.4 ±2.0 | 26.8 |
| Llama 3 70B | 27.0 ±2.2 | 30.4 ±1.9 | 35.0 ±8.3 | 13.2 ±9.2 | 5.3 ±4.1 | 22.2 |
| Yi 1.5 34B | 26.2 ±4.7 | 35.7 ±1.4 | 35.0 ±10.7 | 3.2 ±0.2 | 1.1 ±1.6 | 20.2 |
| Mixtral 8x22B | 21.3 ±1.9 | 19.4 ±1.4 | 39.2 ±12.6 | 1.5 ±1.4 | 3.9 ±1.7 | 17.0 |
| Yi 1.5 9B | 10.8 ±1.0 | 20.0 ±1.4 | 35.0 ±5.0 | 5.0 ±2.2 | 1.3 ±1.5 | 14.4 |
| Llama 3 8B | 22.5 ±2.9 | 24.6 ±2.1 | 23.3 ±7.1 | 0.0 ±0.0 | 1.1 ±1.6 | 14.3 |
| Mixtral 8x7B | 15.8 ±2.0 | 16.1 ±1.4 | 30.0 ±8.2 | 1.1 ±1.6 | 1.1 ±1.6 | 12.8 |
| Chance | 25.0 | 25.0 | 25.0 | 0.0 | 0.0 | 15.0 |

Table 1: **Large-scale spatial cognition results.** The three tables show results for different observation spaces. Results below 50% of human performance are gray. Methods are sorted based on their overall performance.

and map sketching, and less than 30% SPL on route retracing and shortcut discovery, even with allocentric presentation. With egocentric multimodal presentation (the closest counterpart to classic experimental protocols in animal cognition), the models are near chance level on all tasks.

Human performance ranges from 80% to 100% accuracy on image-based presentations of the multiple-choice QA tasks. Since perceiving large sequences of text arrays is non-trivial for humans, the performance drops to 65%–80% for the text presentations.

**Small-scale spatial cognition results.** The results are shown in Table 2. With multimodal presentations, we benchmark GPT-4o, GPT-4v, Claude 3.5 Sonnet, and a number of open multimodal models. With purely textual presentations, we benchmark both open and closed models. We also list the performance of the chance baseline for calibration, as well as human performance (see the appendix for details).

Performance of some model classes (e.g., GPT-4o, GPT-4v, Claude 3.5 Sonnet) on purely textual presentations is considerably higher than on multimodal presentations. The best-performing models, Claude 3.5 Sonnet and GPT-4o, achieve 43.8% and 40.1% average accuracies in the multimodal regime and 64.5% and 65.2% average accuracies with purely textual presentations. (Chance is < 25%.) We attribute this in part to the simplified nature of the text-only implementations of tasks like MRT, MPFB, and JLO (e.g., the text-only presentation of mental rotation uses only 2D shapes and constrained 2D rotations) and in part to the relative developmental maturity of large language models (LLMs) versus multimodal models on the remaining tasks.

On tasks that evaluate visuospatial working memory (specifically SAtt, CBTT, SAdd, and CSWM), the strongest LLMs perform well. On selective attention (SAtt), GPT-4o, Claude 3.5 Sonnet, Mistral 123B, and GPT-4v all achieve over 95% accuracy, matching or outperforming the human performance on this task. On the other hand, all models perform poorly on maze completion (MCT), in both presentation modalities. (Note that the models operate with full visibility, as illustrated in Figure 2.) With multimodal presentation, all evaluated models are near chance on perspective taking (PTT) and the Minnesota Paper Form Board test (MPFB). On mental rotation (MRT), the best models are near chance with multimodal presentation, which uses 3D shapes, and only marginally better with purely textual presentation, which uses 2D arrays and constrained rotations.

| Multimodal | | | | | | | | | | | |
|---|---|---|---|---|---|---|---|---|---|---|---|
| Method | MRT | PTT | WLT | MPFB | JLO | SAtt | MCT | CBTT | SAdd | CSWM | Average |
| Human | 78.5 | 80.0 | 94.0 | 84.0 | 82.0 | 95.0 | – | 100.0 | 98.0 | – | – |
| Claude 3.5 Sonnet | 29.9 ±3.8 | 21.8 ±2.9 | 37.0 ±4.6 | 35.5 ±7.0 | 40.5 ±3.8 | **90.5** ±3.5 | 2.2 ±1.8 | 56.5 ±3.8 | 48.0 ±6.2 | 76.7 ±2.5 | 43.8 |
| GPT-4o | 33.3 ±1.9 | 26.5 ±3.6 | 59.0 ±10.8 | 27.0 ±2.2 | 26.5 ±5.9 | 70.2 ±1.8 | 10.4 ±1.0 | 68.0 ±2.0 | 40.5 ±7.1 | 40.0 ±0.0 | 40.1 |
| GPT-4v | 32.3 ±0.3 | 28.0 ±2.0 | 35.0 ±7.7 | 22.5 ±4.1 | 26.5 ±6.8 | 59.8 ±4.4 | 0.7 ±1.0 | 44.5 ±3.0 | 32.0 ±4.5 | 26.7 ±3.4 | 30.8 |
| Pixtral 12B | 28.3 ±3.1 | 23.2 ±4.9 | 43.0 ±7.0 | 30.5 ±7.9 | 24.5 ±7.3 | 36.0 ±3.9 | OOM | 39.5 ±3.0 | 28.5 ±6.1 | OOM | 25.4* |
| Phi-3.5-vision | 24.1 ±1.0 | 27.0 ±3.2 | 22.5 ±7.9 | 26.0 ±0.0 | 21.0 ±4.1 | 44.0 ±4.6 | OOM | 33.0 ±4.6 | 22.0 ±6.8 | OOM | 22.0* |
| Llava interleave 7B | 25.1 ±3.2 | 25.8 ±5.8 | 25.0 ±8.5 | 25.0 ±3.3 | 24.0 ±5.7 | 32.0 ±4.9 | OOM | 25.5 ±5.7 | 27.0 ±4.1 | OOM | 20.9* |
| Chance | 25.0 | 25.0 | 25.0 | 25.0 | 25.0 | 25.0 | 0.0 | 25.0 | 25.0 | 33.8 ±5.4 | 23.4 |
| Text-only | | | | | | | | | | | |
| Method | MRT | PTT | MPFB | JLO | SAtt | MCT | CBTT | SAdd | CSWM | Average | |
| Human | 90.0 | 75.0 | 92.0 | 98.0 | 96.0 | – | 100.0 | 98.0 | – | – | |
| GPT-4o | 41.9 ±6.2 | 55.5 ±3.9 | 50.5 ±9.6 | 66.5 ±4.8 | **98.8** ±0.4 | 21.5 ±3.8 | 82.5 ±1.7 | **93.5** ±3.6 | 76.7 ±2.5 | 65.2 | |
| Claude 3.5 Sonnet | 37.5 ±1.8 | 50.0 ±7.5 | 45.0 ±6.7 | 70.5 ±4.3 | **97.0** ±1.0 | 10.0 ±1.1 | **97.5** ±0.9 | **91.5** ±4.3 | 82.0 ±3.3 | 64.5 | |
| Mistral 123B | 39.4 ±6.2 | 44.8 ±4.0 | 48.5 ±5.2 | 57.0 ±5.4 | **97.5** ±0.5 | 14.8 ±2.8 | 88.5 ±0.9 | **92.5** ±0.9 | 62.0 ±2.8 | 60.5 | |
| GPT-4v | 41.2 ±7.2 | 67.5 ±6.5 | 34.0 ±6.0 | 62.0 ±4.0 | **95.8** ±1.3 | 3.7 ±1.0 | 87.5 ±3.6 | 79.0 ±2.2 | 45.3 ±2.5 | 57.3 | |
| Llama 3 70B | 28.1 ±9.2 | 29.2 ±2.4 | 38.5 ±3.8 | 42.5 ±0.9 | 71.8 ±3.8 | 1.5 ±1.0 | 52.5 ±5.7 | 62.5 ±5.4 | 34.0 ±5.9 | 40.0 | |
| Mixtral 8x22B | 26.9 ±3.2 | 24.5 ±5.2 | 31.0 ±5.9 | 36.0 ±5.1 | 73.5 ±3.6 | 1.5 ±2.1 | 55.0 ±3.3 | 68.0 ±6.8 | 17.3 ±2.5 | 37.0 | |
| Yi 1.5 34B | 20.6 ±6.0 | 28.0 ±2.1 | 34.5 ±4.6 | 33.5 ±3.6 | 58.2 ±4.3 | 0.7 ±1.0 | 35.5 ±3.8 | 41.5 ±0.9 | 24.0 ±0.0 | 30.7 | |
| Yi 1.5 9B | 21.2 ±1.2 | 23.8 ±2.7 | 30.0 ±3.2 | 24.5 ±5.5 | 48.2 ±4.0 | 0.7 ±1.0 | 36.5 ±4.6 | 51.5 ±8.9 | 24.7 ±8.4 | 29.0 | |
| Llama 3 8B | 14.4 ±1.1 | 25.8 ±5.1 | 26.0 ±4.2 | 27.0 ±1.7 | 46.0 ±3.7 | 0.0 ±0.0 | 27.5 ±7.1 | 30.0 ±7.3 | 26.0 ±6.5 | 24.7 | |
| Mixtral 8x7B | 19.4 ±4.5 | 10.5 ±0.9 | 29.5 ±5.7 | 27.5 ±7.5 | 39.0 ±5.4 | 0.0 ±0.0 | 22.5 ±3.8 | 43.5 ±3.3 | 22.7 ±4.1 | 23.8 | |
| Chance | 25.0 | 25.0 | 25.0 | 25.0 | 25.0 | 0.0 | 25.0 | 25.0 | 33.0 ±5.3 | 23.1 | |

Table 2: **Small-scale spatial cognition results.** The two tables show results for multimodal and text-only presentations, respectively. Results below 50% of human performance are gray, results above 90% of human performance are **bold**. Methods are sorted based on their average performance. (*Some multimodal models ran out of memory on MCT and CSWM tasks; their accuracy is taken to be 0 for calculating the average.)

Humans perform well, achieving over 80% accuracy on the majority of the multiple-choice QA tasks with both text-only and multimodal presentations. Humans perform better on the textual presentations of tasks like MRT, MPFB and JLO than their vision counterparts due to the simplified nature of the text-only implementations.

**Ecological compatibility of SPACE with frontier models.** Our results indicate that current frontier models lack spatial cognition. Alternatively, these results could be the result of models not understanding the inputs presented to them (i.e., the inputs are not ecological compatibility). We study this in Appendix A.1 and demonstrate that this is not the case. Models can understand the inputs correctly and perform non-spatial cognition tasks well, yet fail to demonstrate spatial cognition.

## 5 DISCUSSION

We presented SPACE, a benchmark for spatial cognition in frontier models. Our evaluation of contemporary models brings up intriguing questions and opportunities for further investigation. First, our results underscore that frontier models exhibit a fundamentally different form of intelligence from what has been observed (and studied) in humans and animals. No biological intelligence we have encountered has exhibited such advanced skill in some aspects of higher cognition (Trinh et al., 2024) while failing so profoundly in basic spatial cognition. This is particularly intriguing because in biological intelligence, spatial cognition is considered a prerequisite for higher cognition, and breakdowns in spatial cognition are diagnostic of higher-level disorders (Cappa, 2008; Possin, 2010; Verghese et al., 2017; Cammisuli et al., 2024). From a scientific standpoint, the constellation of traits exhibited by frontier models is fascinating and may inspire a new cognitive science (Simon, 2019). As a precautionary stance, we can refrain from drawing analogies based on experience with biological cognition. (E.g., "a model won the Mathematics Olympiad therefore it possesses a comparable cognitive repertoire to a human Olympiad winner and could be expected to have comparable skill in other domains".)

Could deficiencies in spatial cognition be causally linked to some of the puzzling breakdowns exhibited by contemporary frontier models in higher-level tasks? What is the roadmap for bringing spatial cognition in frontier models up to the level of animal cognition (and perhaps beyond)? Is this a prerequisite for attaining some of the more far-reaching aspirations of contemporary artificial intelligence research? Does embodiment play a role, as it has in prior forms of intelligence (Smith & Gasser, 2005; Savva et al., 2019)? Or will artificial cognition continue to develop along a fundamentally different ontogenetic path? We expect further advances to increase the robustness and generality of frontier models, and to continue to broaden our understanding of the nature of intelligence.

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

# A  APPENDIX

## A.1  ECOLOGICAL COMPATIBILITY OF MULTIMODAL INPUTS WITH FRONTIER MODELS

Our results in Section 4 suggest that state-of-the-art frontier models fail in the spatial cognition tasks presented in SPACE. These failures can be attributed to the lack of spatial cognition in these models. Alternatively, these failures could be due to models not comprehending the inputs presented to them (i.e., the inputs are not *ecologically compatible* with the models). To rule out this alternative possibility, we design additional tests unrelated to spatial cognition on the same vision / text inputs used in our benchmark. If models succeed on these tests, we can infer that the inputs are ecologically compatible since the models can understand and perform tasks using these inputs. In each test, we pose a series of multiple-choice questions evaluating a model's fine-grained understanding of the inputs. We now describe these additional tests.

**Test 1: Given discrete map image / text inputs (see Figure 3), answer the following questions:**

Q1. What is the size of the grid (H x W)?
Q2. What is your current (x, y) location?
Q3. What are the (x, y) locations of all navigable cells? Include cells containing landmarks and your current position.
Q4. What are the (x, y) locations of all obstacle cells?
Q5. What are the landmarks visible in the image / array?
Q6. What are the locations of the landmarks visible in the image / array?

**Test 2: Given an ego image (see Figure 3), answer the following questions:**

Q1. What is the name of the landmark visible in the image?
Q2. Is the landmark <name> in the left half of the image?
Q3. Is the landmark <name> in the right half of the image?
Q4. Is the landmark <name> in the central section of the image?

**Test 3: Given two consecutive ego images from a walkthrough (see Figure 4), answer the following question:**

Q1. What is the action taken to go from image 1 to image 2 (move forward, turn left, turn right, wait/do nothing)?

**Test 4: Given a perspective taking image / text array (see Figures 9 and 10), answer the following questions:**

Q1. How many objects / non-zero locations are present in the image / array?
Q2. What objects / non-zero locations are present in the image / array?
Q3. Is <object / location> to the left of <object / location> in the image / array?
Q4. Is <object / location> to the above <object / location> in the image / array?

**Test 5: Given water level test images (see Figure 11), answer the following questions:**

Q1. Is there water in the water container?
Q2. From image 1 to image 2, is the water container rotated to the left, right or not rotated at all?
Q3. From image 1 to image 2, what is the absolute rotation angle of the water container (in degrees)?

**Test 6: Given a grid of icons / characters from selective attention (see Figures 16 and 17), answer the following questions:**

Q1. How many total objects / characters are present in the image / grid (including repetitions)?
Q2. What is the size of the grid of objects / characters (width x height)?
Q3. How many unique objects / characters are present in the grid (ignore repetitions)?

**Results discussion:** We evaluate GPT-4o and GPT-4v on these tests. The results are shown in Tables 3 and 4. Both models largely understand DM image and text inputs (test 1). However, they fall short in calculating the grid size for DM images (Q1). GPT-4o understands egocentric images,

**Multimodal evaluation**

| Model | Q1 | Q2 | Q3 | Test 1 Q4 | Q5 | Q6 | Avg. | Q1 | Q2 | Test 2 Q3 | Q4 | Avg. | Test 3 Q1 |
|-------|------|------|------|------|-------|------|------|-------|------|------|------|------|------|
| GPT-4o | 30.4 | 78.2 | 89.4 | 87.8 | 100.0 | 93.6 | 79.9 | 100.0 | 84.0 | 95.5 | 83.5 | 92.6 | 59.3 |
| GPT-4v | 55.8 | 86.2 | 89.8 | 91.2 | 99.8 | 79.2 | 83.6 | 98.0 | 45.0 | 36.5 | 56.5 | 66.8 | 48.0 |

**Text-only evaluation**

| Model | Q1 | Q2 | Q3 | Test 1 Q4 | Q5 | Q6 | Avg. | Q1 | Q2 | Test 2 Q3 | Q4 | Avg. | Test 3 Q1 |
|-------|-------|-------|------|------|-------|------|------|----|----|----|----|------|----|
| GPT-4o | 100.0 | 100.0 | 77.5 | 85.6 | 100.0 | 82.8 | 90.9 | - | - | - | - | - | - |
| GPT-4v | 100.0 | 100.0 | 96.8 | 91.0 | 100.0 | 77.6 | 94.2 | - | - | - | - | - | - |

Table 3: Measuring ecological compatibility of multimodal inputs with frontier models (part 1)

**Multimodal evaluation**

| Model | Q1 | Q2 | Test 4 Q3 | Q4 | Avg. | Q1 | Q2 | Test 5 Q3 | Avg. | Q1 | Q2 | Test 6 Q3 | Avg. |
|-------|------|------|------|------|------|-------|------|------|------|------|------|------|------|
| GPT-4o | 99.6 | 99.6 | 89.8 | 87.7 | 92.4 | 100.0 | 73.9 | 38.7 | 64.0 | 83.0 | 90.5 | 58.2 | 77.2 |
| GPT-4v | 78.3 | 87.4 | 78.4 | 76.0 | 79.0 | 100.0 | 56.3 | 32.4 | 55.4 | 74.5 | 88.0 | 35.8 | 66.0 |

**Text-only evaluation**

| Model | Q1 | Q2 | Test 4 Q3 | Q4 | Avg. | Q1 | Q2 | Test 5 Q3 | Avg. | Q1 | Q2 | Test 6 Q3 | Avg. |
|-------|------|------|------|------|------|----|----|----|----|-------|-------|------|------|
| GPT-4o | 97.6 | 99.6 | 99.5 | 94.2 | 97.4 | - | - | - | - | 100.0 | 100.0 | 99.5 | 99.8 |
| GPT-4v | 96.8 | 98.1 | 92.2 | 76.8 | 90.8 | - | - | - | - | 99.5 | 100.0 | 94.8 | 98.1 |

Table 4: Measuring ecological compatibility of multimodal inputs with frontier models (part 2)

i.e., recognizes and localizes landmarks in egocentric images (test 2). GPT-4v recognizes landmarks well (Q1), but performs poorly in localization (Q2, Q3 and Q4). Both GPT-4o and GPT-4v perform poorly on action estimation (test 3) and estimation of water container rotations (Q2 and Q3 in test 5). GPT-4o excels in understanding the perspective taking inputs with multimodal and text-only presentations (test 4). GPT-4v also performs well on test 4, but is worse with multimodal inputs when compared to text-only inputs. Finally, both GPT-4o and GPT-4v perform adequately with counting objects (Q1 in test 6) and grid sizes (Q2 in test 6) on selective attention task inputs with multimodal inputs. However, they struggle to calculate the number of unique objects / characters (Q3 in test 6). Both GPT-4o and GPT-4v excel at the text-only presentation of test 6.

Our results indicate that state-of-the-art models can understand multimodal and text-only inputs provided in our benchmark. They perform well in most of the tests, but have specific shortcomings (e.g., localizing landmarks in ego images for GPT-4v, understanding rotations of water containers and counting unique characters / objects in a grid). Importantly, the average results on each test is much higher than the SPACE task counterparts. For example, even though GPT-4o and GPT-4v understand DM text inputs nearly perfectly (test 1), they perform poorly in the DM text versions of the large-scale spatial cognition tasks (see Table 1). Similarly, even though GPT-4o understands the perspective taking inputs nearly perfectly for both text-only and multimodal presentations, it performs poorly on the perspective taking task in SPACE (see Table 2). Therefore, the failure of frontier models on SPACE is most likely due to their lack of spatial cognition, and not because they cannot understand the inputs presented to them.

## A.2 SMALL-SCALE SPATIAL COGNITION: ADDITIONAL DETAILS

We described the small-scale spatial cognition tasks from our benchmark in Section 3.2. Here, we provide additional details about the historical context and motivations behind these tasks.

**Mental rotation test (MRT).** This was introduced by Vandenberg & Kuse (1978) as a test of spatial visualization. The original MRT contained 20 items, where each item consisted of a criterion figure, two correct alternatives, and two distractors (Vandenberg & Kuse, 1978). The criterion figure is a perspective rendering of a 3D criterion shape from Shepard & Metzler (1971). The correct alternatives are rotated versions of the criterion shape, where the rotation is applied in the 2D image space

on the criterion figure, or along the vertical axis in 3D for the criterion shape. The distractors are rotated mirror-images of the criterion shape or renderings of other criterion shapes. The goal was to identify the two correct alternatives from the four choices. We implement a version of MRT with one correct choice and three distractors, and incorporate rotations along multiple axes (Peters et al., 1995).

**Perspective taking test (PTT).** This was introduced by Kozhevnikov & Hegarty (2001) as a test of spatial orientation. An arrangement of objects is shown on a piece of paper. A test participant is asked to take the perspective of standing next to an object (say, object A) facing another (say, object B), and is required to point to a third object (say, object C). This task has been used extensively in subsequent literature (Hegarty & Waller, 2004; Weisberg et al., 2014; Meneghetti et al., 2022). We implement this task by randomly sampling $N$ icons of objects like cars, carrots, chairs, and grapes and place them at random locations in an image (with no overlap between objects). We then randomly sample three of the $N$ objects as A, B, and C.

**Water level test (WLT).** This was introduced by Piaget et al. (1957) as a test of visuospatial perception. Originally, the test was designed to evaluate children's knowledge about the horizontal nature of the surface of water in a sealed bottle regardless of its orientation. Children were presented with bottles partially filled with colored water and asked to imagine the position of the water if it were tilted. Children had to gesture, draw, or use cardboard cutouts to answer the question (Piaget et al., 1957; Foltz, 1978; Wittig & Allen, 1984). Performance on the water-level test was found to be related to performance on spatial ability tests (Foltz, 1978; Wittig & Allen, 1984).

**Judgement of Line Orientation test (JLO).** This was introduced by Benton (1994) as a measure of visuospatial perception. The original implementation contained 30 samples presented in a flip-book style, where two lines are shown at the top of each page. The goal is to determine the angles between the two lines by comparing them to an array of reference lines (i.e., pick two reference lines that have same angle between them as the lines at the top). There have been multiple variations of JLO with subsets of the 30 questions for faster evaluation (Spencer et al., 2013). We recreate the JLO test suite by randomly sampling pairs of lines on a 2D plane with an angle between 0 to 180 degrees (in multiples of 18 degrees) and formulate it as multiple-choice QA.

**Selective attention task (SAtt).** This is designed to evaluate selective spatial attention (Serences & Kastner, 2014; Pahor et al., 2022). In particular, we use the widely used cancellation task, where the goal is to search for and mark out target stimuli embedded amidst distractors (Della Sala et al., 1992; Brickenkamp & Zillmer, 1998; Dalmaijer et al., 2015; Lacroix et al., 2021; Pahor et al., 2022; Kalina & Walgrave, 2004). The stimuli may be characters (Brickenkamp & Zillmer, 1998; Dalmaijer et al., 2015; Pahor et al., 2022; Della Sala et al., 1992; Kalina & Walgrave, 2004), pictures (Lacroix et al., 2021; Pahor et al., 2022), or icons (Lacroix et al., 2021). We implement this task with objects as the stimuli for visual evaluation and characters as stimuli for textual evaluation.

**Maze completion task (MCT).** This task was designed to evaluate spatial orientation, planning, and executive functioning (Lacroix et al., 2021). It was used as a neuropsychological test to assess executive function disorders in children (Marquet-Doléac et al., 2010).

**Corsi block-tapping task (CBTT).** This is designed to assess visuospatial working memory and attention in healthy participants and patients with known or suspected brain damage (Corsi, 1972; Claessen et al., 2015). An examiner demonstrates a sequence of block-tapping movements on a board containing fixed blocks placed in pseudo-random positions. Participants are required to reproduce the same sequence (forward condition) or the inverted sequence (backward condition) of block-tapping movements to succeed. We evaluate frontier models on the forward condition since prior work has not found significant differences between task performance in the forward and backward conditions (Claessen et al., 2015).

**Spatial addition task (SAdd).** This was introduced in the fourth edition of the Wechsler Memory Scale, a suite of neuropsychological tests to evaluate memory function in individuals aged 16 to 90 (Wechsler, 2009). SAdd evaluates visuospatial storage and manipulation in working memory. A test participant is shown a grid with blue and red dots for five seconds. The participant is asked to remember the location of the blue dots and ignore the red dots. The participant is then shown another such grid. The objective is to add the two grids together by following certain rules. If a grid location has a blue dot in exactly one of grids, the result should be blue. If a grid location has blue dots on both grids, the result should be white.

**Cambridge spatial working memory test (CSWM).** This was designed to evaluate spatial working memory in human subjects (Sahakian et al., 1988). Multiple colored boxes are shown on a screen. A yellow 'treasure' is initially hidden in one of the boxes. The participant must select boxes one at a time to open them and search for the treasure. Once the treasure is found, another treasure is placed in one of the remaining boxes. The intention is for the participant to locate all the yellow treasures via a process of elimination.

## A.3 SPACE EXAMPLES

We illustrate examples for each task from our proposed SPACE benchmark.

**Large-scale spatial cognition**

- **Egocentric image observations:** Figures 4
- **DM image observations[2]:** Figures 5, 6

**Small-scale spatial cognition**

- **MRT:** Figures 7 and 8
- **PTT:** Figures 9 and 10
- **WLT:** Figure 11
- **MPFB:** Figures 12 and 13
- **JLO:** Figures 14 and 15
- **MCT:** Figures 22 and 23
- **CBTT:** Figures 18 and 19
- **SAdd:** Figures 20 and 21
- **CSWM:** Figures 24 and 25

## A.4 IMPLEMENTATION DETAILS

We provide additional implementation details about our experimental setup in this section.

**3D environment generation:** We create ten environment layouts based on prior work in cognitive science and artificial intelligence (Tolman, 1948; Gillner & Mallot, 1998; Richardson et al., 1999; Banino et al., 2018; Bouchekioua et al., 2021). Figure 3 shows bird's-eye view images of each layout. We populate each environment with visual landmarks in the form of paintings hanging on the walls, where the painting frames are 3D meshes and the paintings are images from ImageNet (Deng et al., 2009). To create a 3D environment for a given layout, we first randomly sample textures for walls, floors, and ceilings from a database of textures to create the base 3D mesh. Next, we randomly assign ImageNet images and 3D frame meshes to predefined landmark locations in the environment. We create the 3D environment using the Trimesh library and export it in glTF format (Dawson-Haggerty et al., 2019). We simulate the environment using the Habitat simulator (Savva et al., 2019). We create 3 environments per layout, for a total of 30 environments in our benchmark.

**Randomized trails for evaluation:** For multiple-choice QA, we randomize the placement of the correct answer among the four choices such that it appears in each of the four positions once, yielding four trials per question. For each trial, we evaluate the performance over all questions to obtain the average accuracy. For interactive tasks like route retracing, shortcut discovery, MCT and CSWM, we evaluate each model in three independent trials and obtain the corresponding metrics. By performing multiple trials, we can compute means and standard deviations for each model on each task across the trials.

**Human performance:** We obtain human performance on SPACE tasks by evaluating 29 participants (aged 20 - 50). We evenly divide the questions from our benchmark across all participants. For each participant, we provide HTML files containing a subset of questions from each SPACE task and the corresponding choices. The HTML files contain formatted versions of the prompts used to evaluate

---

[2]DM text observations are obtained by simply converting the DM image to text as illustrated in Figure 3.

frontier models. We do not provide any additional instructions or background information about how to solve the tasks. For efficiency, we group all questions corresponding to a single environment in the large-scale spatial cognition tasks. Each participant is assigned to view a video walkthrough from one environment and asked to answer a series of questions about that same environment. This is in line with classical protocols in human cognition (Allen et al., 1996; Hegarty et al., 2006; Pazzaglia & Taylor, 2007; Weisberg et al., 2014; Meneghetti et al., 2016; 2021). We further provide a CSV file where the participant is instructed to enter the answers. We instruct the participants to perform all tasks mentally without any aids like pen and paper. Each participant is estimated to have taken 60 to 90 minutes to answer all the questions. The participants send us their responses and we evaluate them collectively. We denote the collective performance of all participants as the human performance in Tables 1 and 2.

Note that we establish the human baseline only for the multiple-choice QA tasks since it was straightforward to share the test materials with the participants online and obtain their answers. The interactive tasks would require us to meet participants in person to perform evaluations and we were not equipped to do this.

**Image preprocessing:** For most of our experiments, we use square images. We provide the images to models as is without preprocessing. For most models (especially closed-source ones), the processing of the image beyond the input stage is outside our control. We rely on the model creators to correctly process the images. The exact image resolution and aspect ratios are task-dependent and listed in Table 5. For egocentric video inputs in the large-scale spatial cognition tasks, the number of frames varies from 61 to 240. Since GPT-4o, GPT-4v and Claude 3.5 Sonnet APIs did not permit 240+ frames as inputs, we subsample the video frames by a factor of 2 before providing them to the model. For DM video inputs, the number of frames varies from 13 to 72. We provide them as is to the model.

| Task | Image resolutions (W $\times$ H) |
|---|---|
| Large-scale spatial cognition (Ego images) | $512 \times 512$ |
| Large-scale spatial cognition (DM images) | $512 \times 512$ |
| Mental rotation (MRT) | Varies from $595 \times 541$ to $1133 \times 1432$ since we crop the redundant white space around images. |
| Perspective taking (PTT) | $640 \times 480$ |
| Water level (WLT) | Varies from $239 \times 488$ to $787 \times 631$ since we crop the redundant white space around images. |
| Minnesota paper form board (MPFB) | $480 \times 480$ for choice images (i.e., puzzle pieces put together). Varies from $831 \times 578$ to $1211 \times 740$ for the image containing the puzzle pieces. |
| Judgement of line orientation (JLO) | $512 \times 512$ for the input image, $1656 \times 910$ for the legend |
| Selective attention (SAtt) | $512 \times 512$ for $3 \times 3$ grids, $768 \times 768$ for $4 \times 4$ grids, and $1024 \times 1024$ for $5 \times 5$ and $6 \times 6$ grids |
| Maze completion (MCT) | $1100 \times 1100$ for $11 \times 11$ mazes, $2300 \times 2300$ for $23 \times 23$ mazes, and $3100 \times 3100$ for $31 \times 31$ mazes |
| Corsi block-tapping (CBTT) | $1024 \times 1024$ |
| Spatial addition (SAdd) | $300 \times 300$ for $3 \times 3$ grids, $500 \times 500$ for $5 \times 5$ grids, $700 \times 700$ for $7 \times 7$ grids, and $900 \times 900$ for $9 \times 9$ grids |
| Cambridge spatial working memory (CSWM) | $1024 \times 1024$ |

Table 5: Image resolutions and aspect ratios for images and videos in SPACE.

**Prompting frontier models for SPACE tasks:** We evaluate frontier models on each of the SPACE tasks using zero-shot prompting. For each task, we design a prompt that provides a detailed description of the task and the expected response format. Below, we provide the prompt templates for each of the SPACE tasks. While the prompts have been formatted for visual display in LaTeX, the content remains the same. We have replaced images and arrays (in some cases) with placeholders for brevity.

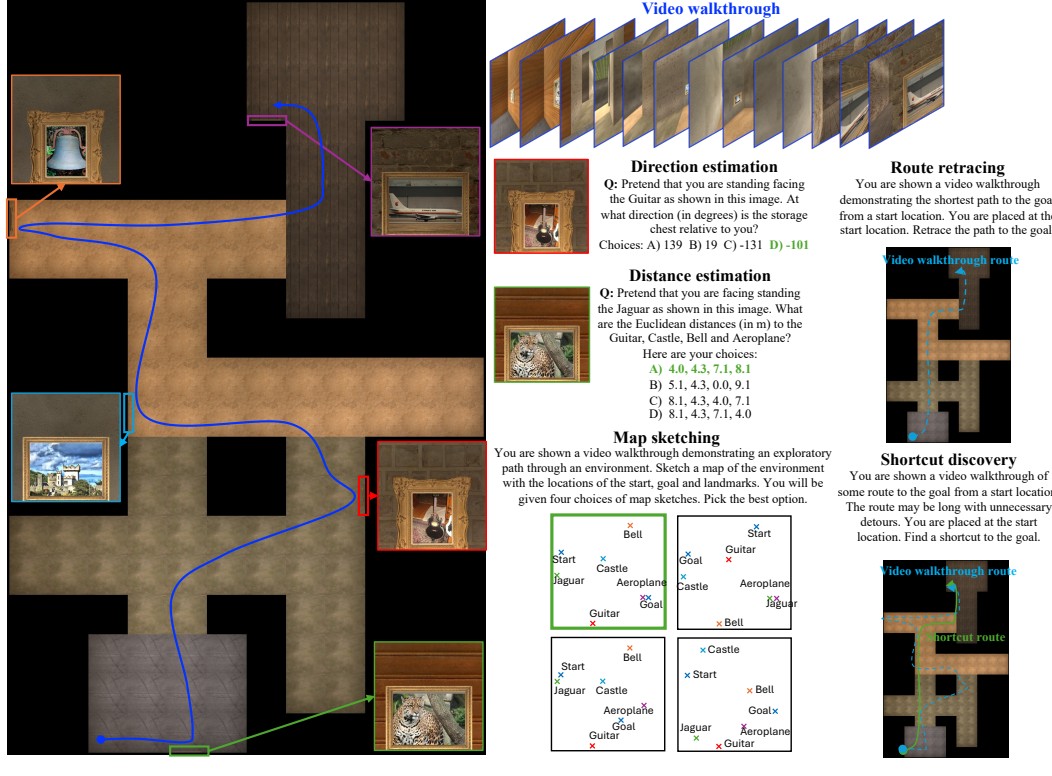

Figure 4: **Large-scale spatial cognition with ego image observations**

**Large-scale spatial cognition**

- **Direction estimation:** Prompts 1, 2 and 3
- **Distance estimation:** Prompts 4, 5 and 6
- **Map sketching:** Prompts 4, 5 and 6
- **Route retracing:** Prompts 10, 11, 12 and 13
- **Shortcut discovery:** Prompts 14, 15, 16 and 17

**Small-scale spatial cognition**

- **Mental rotation test:** Prompts 18 and 19
- **Perspective taking test:** Prompts 20 and 21
- **Water level test:** Prompt 22
- **Minnesota Paper Form Board test:** Prompts 23 and 24
- **Judgement of Line Orientation test:** Prompts 25 and 26
- **Selective attention task:** Prompts 27 and 28
- **Maze completion task:** Prompts 29 and 30
- **Corsi block-tapping task:** Prompts 32 and 31
- **Spatial addition task:** Prompts 33 and 34
- **Cambridge spatial working memory test:** Prompts 35 and 36

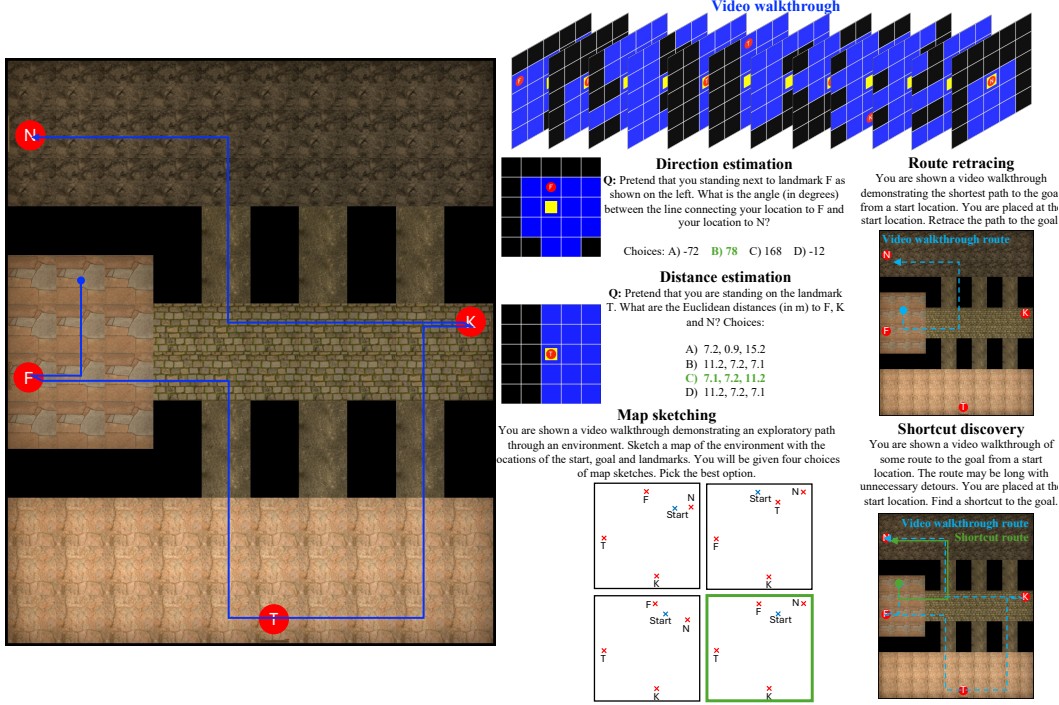

Figure 5: **Large-scale spatial cognition with DM image observations.** Please note that the top-down visualization on the left needs to be rotated by 90° clockwise to get the DM images.

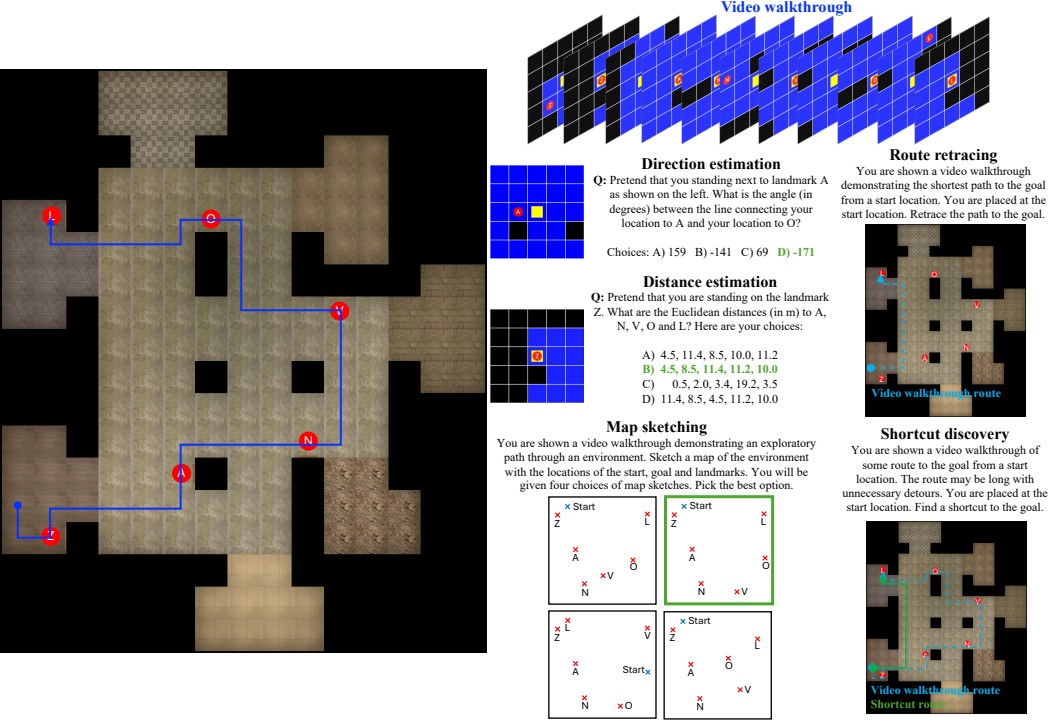

Figure 6: **Large-scale spatial cognition with DM image observations.** Please note that the top-down visualization on the left needs to be rotated by 90° clockwise to get the DM images.

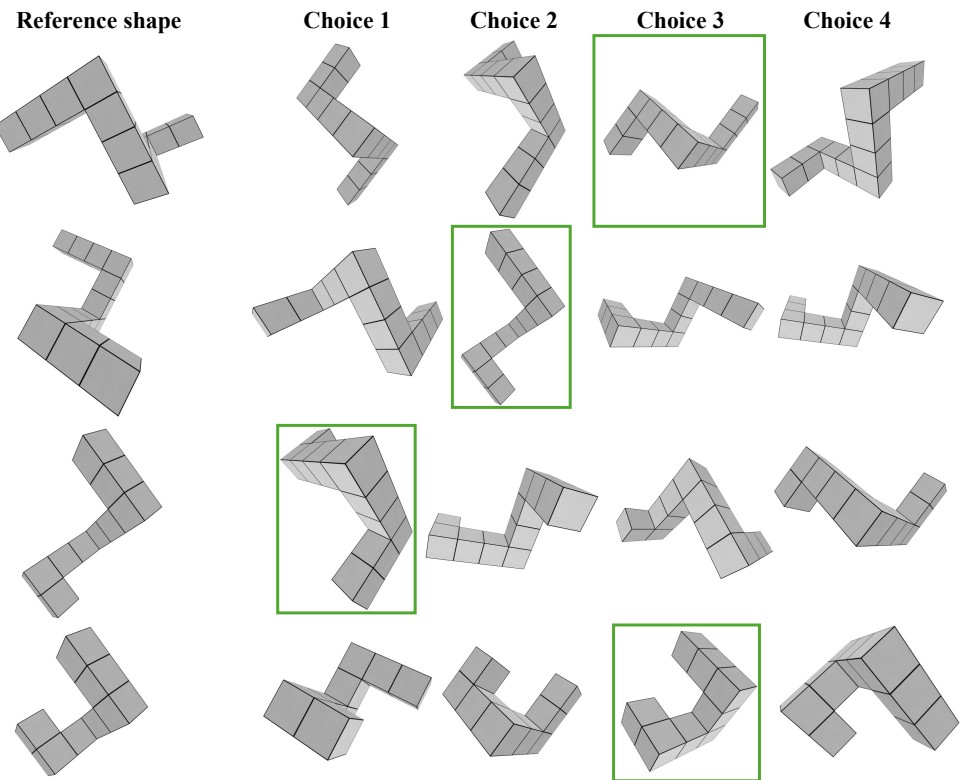

Figure 7: **Mental rotation (MRT) with visual inputs:** Which choice image shows the reference shape rotated in 3D?

| Reference array | Choice 1 | Choice 2 | Choice 3 | Choice 4 |
|---|---|---|---|---|
| 0,0,**8,1,8**,0,0 | 0,0,**8,1,8**,0,0 | **7**,0,**1**,0,**4**,0,0 | 0,0,0,**9**,0,0,0 | 0,0,0,**9**,0,0,0 |
| 0,0,**2**,0,**1**,0,0 | 0,0,**1**,0,**2**,0,0 | 0,**2**,0,0,0,0,0 | **2**,0,0,0,0,0,0 | 0,0,0,0,0,0,**2** |
| 0,0,**7**,0,**7**,0,**4** | **4**,0,**7**,0,**7**,0,0 | 0,0,**2**,0,**7,1,8** | **3,5**,0,**9,7,2,8** | **8,2,7,9**,0,**5,3** |
| **9**,0,**9,5**,0,0,0 | 0,0,0,**5,9**,0,**9** | 0,**1,7,5**,0,0,**1** | 0,**1,7,5**,0,0,**1** | **1**,0,0,**5,7,1**,0 |
| 0,0,0,**7,2**,0,**1** | **1**,0,**2,7**,0,0,0 | **3,5**,0,**9,7,2,8** | 0,0,**2**,0,**7,1,8** | **8,1,7**,0,**2**,0,0 |
| 0,0,**5,1**,0,**2**,0 | 0,**2**,0,**1,5**,0,0 | **2**,0,0,0,0,0,0 | 0,**2**,0,0,0,0,0 | 0,0,0,0,0,**2**,0 |
| 0,**2,3**,0,0,0,**7** | **7**,0,0,0,**3,2**,0 | 0,0,0,**9**,0,0,0 | **7**,0,**1**,0,**4**,0,0 | 0,0,**4**,0,**1**,0,**7** |

| | | | | |
|---|---|---|---|---|
| **view**,none,**back** | **view**,**used**,**view** | **back**,none,**view** | **view**,**been**,none | **view**,**used**,**view** |
| **used**,**year**,none | **been**,**year**,none | none,**year**,**used** | **used**,**year**,none | none,**year**,**been** |
| **view**,**been**,none | none,none,**back** | none,**been**,**view** | **view**,none,**back** | **back**,none,none |

| | | | | |
|---|---|---|---|---|
| 0,0,0,0,**9,1,9**,0,0 | 0,**5**,0,**3**,0,0,0,0,**6** | 0,0,**3,9,2,4,7**,0,0 | 0,0,**7,4,2,9,3**,0,0 | 0,0,**9,1,9**,0,0,0,0 |
| 0,0,0,**8,2**,0,**3**,0,0 | 0,**4**,0,0,0,0,**9,4**,0 | 0,0,0,0,0,0,0,**4,5** | **5,4**,0,0,0,0,0,0,0 | 0,0,**3**,0,**2,8**,0,0,0 |
| **3**,0,0,0,0,**1**,0,0,**3** | **7**,0,0,0,0,0,**2,2**,0 | 0,0,0,**9,2,1**,0,0,0 | 0,0,0,**1,2,9**,0,0,0 | **3**,0,0,0,**1**,0,0,0,**3** |
| **9**,0,**9,1**,0,0,0,0,0 | **4**,0,**1**,0,0,0,0,**9**,0 | 0,**8**,0,**1**,0,0,0,0,**3** | **3**,0,0,0,0,**1**,0,**8**,0 | 0,0,0,0,0,0,**1,9**,0,**9** |
| **2**,0,**2**,0,**3**,0,0,0,0 | **2**,0,**2**,0,**3**,0,0,0,0 | **9,2,1**,0,**3**,0,0,0,0 | 0,0,0,0,**3**,0,**1,2,9** | 0,0,0,0,**3**,0,**2**,0,**2** |
| **4**,0,**1**,0,0,0,0,0,**9** | **9**,0,**9,1**,0,0,0,0,0 | **1**,0,0,0,0,0,0,**9**,0 | 0,**9**,0,0,0,0,0,0,**1** | **9**,0,0,0,0,0,**1**,0,**4** |
| **7**,0,0,0,0,0,**2,2**,0 | **3**,0,0,0,**1**,0,0,0,**3** | **9,3**,0,0,0,0,**2,4**,0 | 0,**4,2**,0,0,0,0,**3,9** | 0,**2,2**,0,0,0,0,0,**7** |
| 0,**4**,0,0,0,**9,4**,0,0 | 0,0,0,**8,2**,0,**3**,0,0 | 0,0,0,0,0,0,**9,2**,0,0 | 0,0,**2,9**,0,0,0,0,0 | 0,0,**4,9**,0,0,0,**4**,0 |
| 0,**5**,0,**3**,0,0,0,0,**6** | 0,0,0,0,**9,1,9**,0,0 | 0,0,**3**,0,0,0,0,0,**6** | **6**,0,0,0,0,0,**3**,0,0 | **6**,0,0,0,0,**3**,0,**5**,0 |

| | | | | |
|---|---|---|---|---|
| 0,0,0,**3,9**,0,**8** | 0,0,0,0,0,0,0 | **8**,0,**9,3**,0,0,0 | 0,**5**,0,0,**2,5,8** | 0,0,0,0,0,0,0 |
| 0,0,**3**,0,**8**,0,**5** | 0,0,**4**,0,0,0,0 | **5**,0,**8**,0,**3**,0,0 | **6,1,1,4**,0,0,0 | 0,0,0,0,**4**,0,0 |
| 0,**4,1**,0,0,0,**2** | 0,**3,1**,0,0,0,**1** | **2**,0,0,0,**1,4**,0 | 0,0,0,**3**,0,**8,9** | **1**,0,0,0,**1,3**,0 |
| 0,0,0,0,**3,4**,0 | **3**,0,0,0,**9,4,8** | 0,**4,3**,0,0,0,0 | **8,4,9**,0,0,0,**3** | **8,4,9**,0,0,0,**3** |
| 0,0,0,**9**,0,**1**,0 | **9,8**,0,**3**,0,0,0 | 0,**1**,0,**9**,0,0,0 | **1**,0,0,0,0,**1,3**,0 | 0,0,0,**3**,0,**8,9** |
| 0,0,0,**4**,0,**1,5** | 0,0,0,**4,1,1,6** | **5,1**,0,**4**,0,0,0 | 0,0,0,0,**4**,0,0 | **6,1,1,4**,0,0,0 |
| 0,0,**1,8**,0,**6**,0 | **8,5,2**,0,0,**5**,0 | 0,**6**,0,**8,1**,0,0 | 0,0,0,0,0,0,0 | 0,**5**,0,0,**2,5,8** |

Figure 8: **Mental rotation (MRT) with text inputs:** Which choice array shows the reference array rotated in 2D?

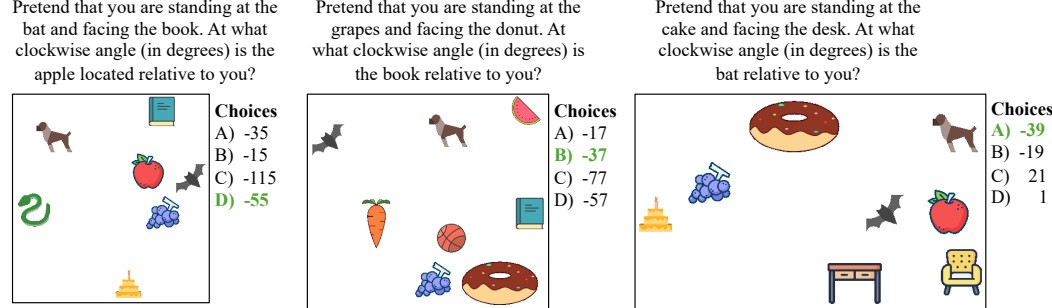

Figure 9: **Perspective taking (PTT) with visual inputs**.

Pretend that you are standing at 6 and facing 9. At what clockwise angle (in degrees) is 1 relative to you?

```
0,0,0,8,0,0,0,0
0,0,4,7,0,0,0,0
0,0,0,0,0,0,0,0
0,0,6,2,3,0,0,0
9,0,0,0,0,0,0,0
0,0,0,0,0,1,0,0
0,0,0,0,0,0,0,5
0,0,0,0,0,0,0,0
```

**Choices**
A) -119
B) -59
C) -99
D) -159

Pretend that you are standing at 7 and facing 2. At what clockwise angle (in degrees) is 1 relative to you?

```
0,3,0,0
1,6,0,0
0,7,2,0
0,4,8,0
```

**Choices**
A) -115
B) 165
C) -135
D) -75

Pretend that you are standing at 4 and facing 9. At what clockwise angle (in degrees) is 6 relative to you?

```
0,0,0,0,0,0,0
0,0,0,0,0,0,0
0,0,0,1,0,6,0
7,0,0,0,0,0,0
2,0,5,0,0,4,8
0,3,0,9,0,0,0
0,0,0,0,0,0,0
```

**Choices**
A) 136
B) 116
C) 76
D) 156

Pretend that you are standing at 9 and facing 4. At what clockwise angle (in degrees) is 2 relative to you?

```
0,9,0,5
0,3,0,0
0,0,8,0
4,7,2,0
```

**Choices**
A) -36
B) -96
C) -76
D) 4

Figure 10: **Perspective taking (PTT) with text inputs**. *The array colors are only for illustration purposes.*

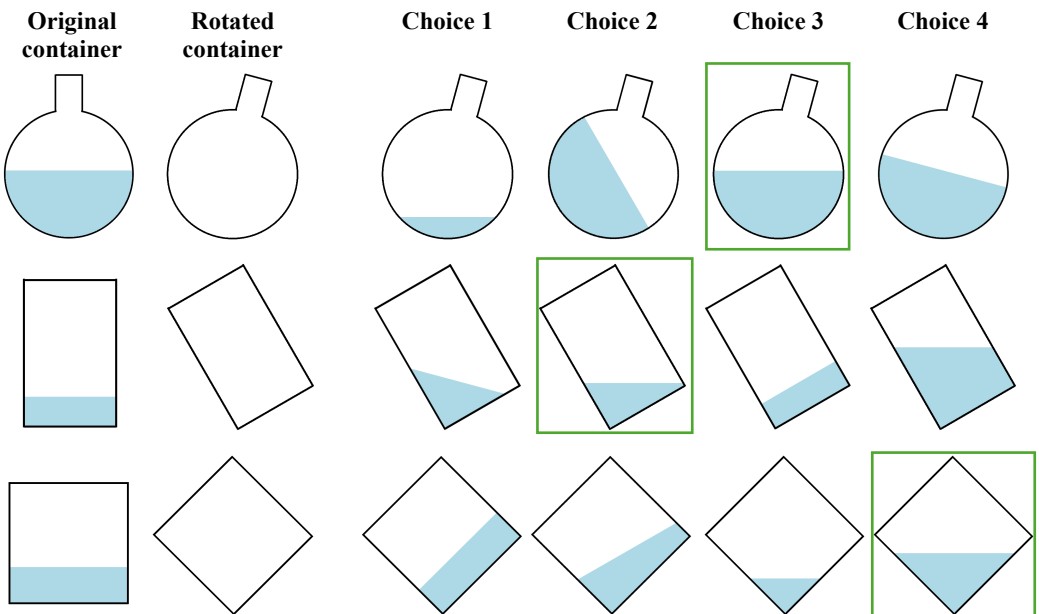

Figure 11: **Water level (WLT) with vision inputs:** Given a water container filled with water, predict the water level in the rotated container.

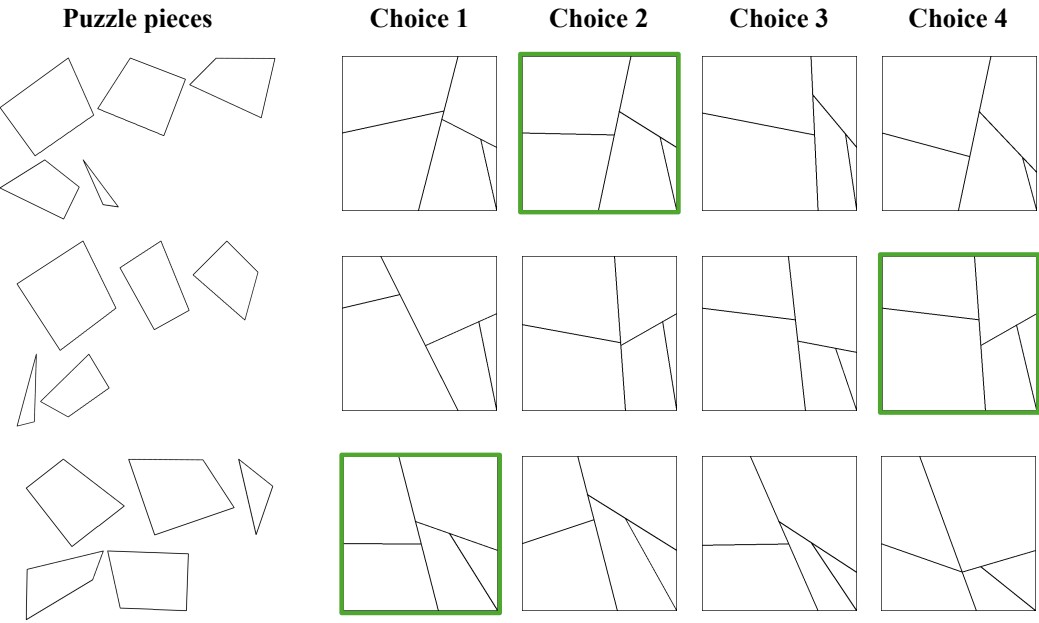

**Puzzle pieces** | **Choice 1** | **Choice 2** | **Choice 3** | **Choice 4**

Figure 12: **Minnesota Paper Form Board (MPFB) with visual inputs:** Which one of the four choices shows what it would be like when the puzzle pieces are put together? The puzzle pieces can be rotated but not flipped.

**Array pieces** | **Choice 1** | **Choice 2** | **Choice 3** | **Choice 4**

```
1,1,1,1   1,1,1,1,1,1
1,0,0,1   1,0,0,0,0,1
1,0,0,1   1,0,0,0,0,1
1,0,0,1   1,0,0,0,0,1
1,0,0,1   1,0,0,0,0,1
1,0,0,1   1,0,0,0,0,1
1,1,1,1   1,1,1,1,1,1

1,1,1,1,1,1   1,1,1,1
1,0,0,0,0,1   1,0,0,1
1,0,0,0,0,1   1,0,0,1
1,1,1,1,1,1   1,1,1,1
```

Choice 1:
```
1,1,1,1,1,1,1,1,1,1
1,0,0,0,0,0,1,0,0,1
1,0,0,0,0,0,1,0,0,1
1,1,1,1,1,1,1,1,1,1
1,0,0,1,0,0,0,0,0,1
1,0,0,1,0,0,0,0,0,1
1,0,0,1,0,0,0,0,0,1
1,0,0,1,0,0,0,0,0,1
1,0,0,1,0,0,0,0,0,1
1,1,1,1,1,1,1,1,1,1
```

Choice 2:
```
1,1,1,1,1,1,1,1,1,1
1,0,0,0,0,0,1,0,0,1
1,0,0,0,0,0,1,0,0,1
1,0,0,0,0,0,1,0,0,1
1,0,0,0,0,0,1,0,0,1
1,0,0,0,0,0,1,0,0,1
1,0,0,0,0,0,1,0,0,1
1,1,1,1,1,1,1,1,1,1
1,0,0,1,0,0,0,0,0,1
1,1,1,1,1,1,1,1,1,1
```

Choice 3:
```
1,1,1,1,1,1,1,1,1,1
1,0,0,0,0,0,1,0,0,1
1,0,0,0,0,0,1,0,0,1
1,1,1,1,1,1,1,1,1,1
1,0,0,1,0,0,0,0,0,1
1,0,0,1,0,0,0,0,0,1
1,0,0,1,0,0,0,0,0,1
1,0,0,1,0,0,0,0,0,1
1,0,0,1,0,0,0,0,0,1
1,1,1,1,1,1,1,1,1,1
```

Choice 4:
```
1,1,1,1,1,1,1,1,1,1
1,0,0,0,0,0,1,0,1
1,0,0,0,0,0,1,0,1
1,1,1,1,1,1,1,1,1,1
1,0,0,1,0,0,0,0,0,1
1,0,0,1,0,0,0,0,0,1
1,0,0,1,0,0,0,0,0,1
1,0,0,1,0,0,0,0,0,1
1,0,0,1,0,0,0,0,0,1
1,1,1,1,1,1,1,1,1,1
```

```
1,1,1,1   1,1,1,1,1,1
1,0,0,1   1,0,0,0,0,1
1,0,0,1   1,0,0,0,0,1
1,0,0,1   1,0,0,0,0,1
1,0,0,1   1,0,0,0,0,1
1,1,1,1   1,0,0,0,0,1
          1,1,1,1,1,1
1,1,1,1,1
1,0,0,0,1   1,1,1,1,1
1,0,0,0,1   1,0,0,0,1
1,0,0,0,1   1,0,0,0,1
1,0,0,0,1   1,0,0,0,1
1,1,1,1,1   1,1,1,1,1
```

Figure 13: **Minnesota Paper Form Board (MPFB) with text inputs:** Which one of the four choices shows what it would be like when the array pieces are put together? The pieces merge at the edges (1s). They can be rotated in multiples of 90 degrees but not flipped. *The array colors are purely for illustration purposes.*

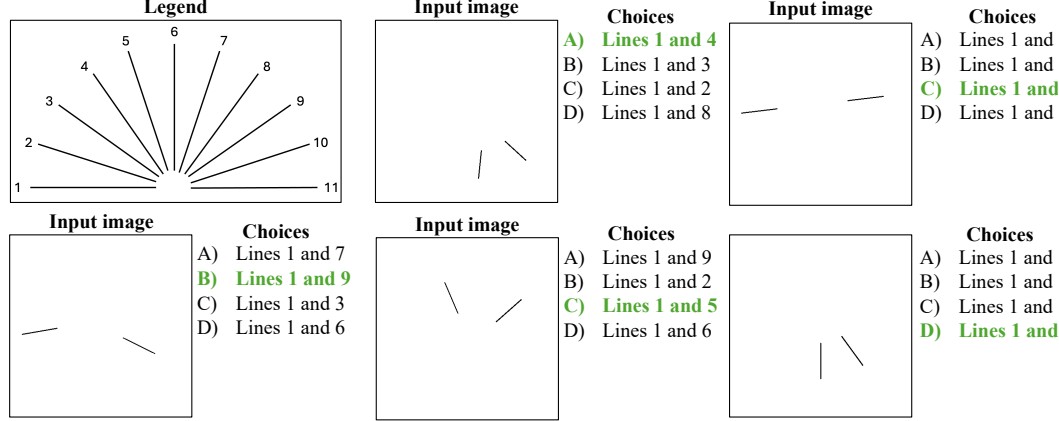

Figure 14: **Judgement of line orientation (JLO) with visual inputs:** Which pair of lines from the legend have a matching angle to the lines in the input image?

Figure 15: **Judgement of line orientation (JLO) with text inputs:** Which choice has an angle between lines 1 and 2 that matches the angle from the input array? *The array colors are purely for illustration purposes.*

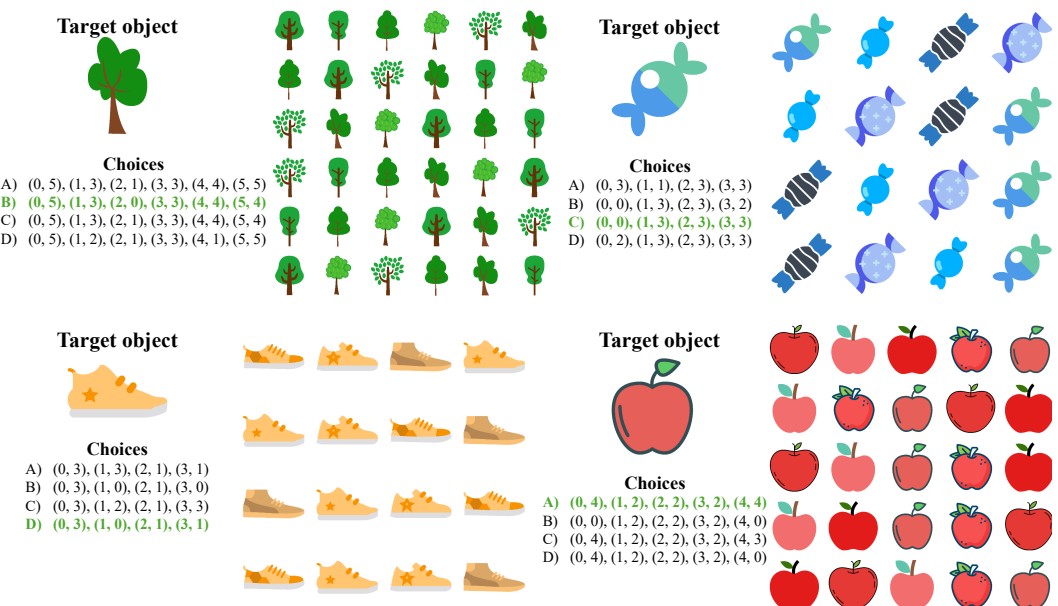

**Target object**

**Choices**
A)  (0, 5), (1, 3), (2, 1), (3, 3), (4, 4), (5, 5)
B)  **(0, 5), (1, 3), (2, 0), (3, 3), (4, 4), (5, 4)**
C)  (0, 5), (1, 3), (2, 1), (3, 3), (4, 4), (5, 4)
D)  (0, 5), (1, 2), (2, 1), (3, 3), (4, 1), (5, 5)

**Target object**

**Choices**
A)  (0, 3), (1, 1), (2, 3), (3, 3)
B)  (0, 0), (1, 3), (2, 3), (3, 2)
C)  **(0, 0), (1, 3), (2, 3), (3, 3)**
D)  (0, 2), (1, 3), (2, 3), (3, 3)

**Target object**

**Choices**
A)  (0, 3), (1, 3), (2, 1), (3, 1)
B)  (0, 3), (1, 0), (2, 1), (3, 0)
C)  (0, 3), (1, 2), (2, 1), (3, 3)
D)  **(0, 3), (1, 0), (2, 1), (3, 1)**

**Target object**

**Choices**
A)  **(0, 4), (1, 2), (2, 2), (3, 2), (4, 4)**
B)  (0, 0), (1, 2), (2, 2), (3, 2), (4, 0)
C)  (0, 4), (1, 2), (2, 2), (3, 2), (4, 3)
D)  (0, 4), (1, 2), (2, 2), (3, 2), (4, 0)

Figure 16: **Selective attention (SAtt) with visual inputs:** What are the (row, column) grid locations of the reference object? The top-left element of the grid is (0, 0).

**Target character:** b

```
l,k,h,s,p,z,x,b,n
b,p,x,s,z,n,h,l,k
n,p,x,s,k,h,l,z,b
k,b,s,x,p,l,z,h,n
x,l,s,z,p,b,k,n,h
p,k,h,n,l,b,x,s,z
n,h,l,z,k,x,p,s,b
x,b,n,l,p,h,k,z,s
h,x,n,k,p,l,b,z,s
```

**Choices**
A)  (0, 7), (1, 0), (2, 8), (3, 1), (4, 5), (5, 5), (6, 8), (7, 1), (8, 2)
B)  **(0, 7), (1, 0), (2, 8), (3, 1), (4, 5), (5, 5), (6, 8), (7, 1), (8, 6)**
C)  (0, 3), (1, 1), (2, 0), (3, 1), (4, 5), (5, 6), (6, 3), (7, 5), (8, 0)
D)  (0, 7), (1, 0), (2, 5), (3, 1), (4, 5), (5, 5), (6, 8), (7, 1), (8, 6)

**Target character:** 5

```
7,0,6,5,4
0,7,6,5,4
6,7,5,4,0
7,4,5,6,0
0,6,7,4,5
```

**Choices**
A)  (0, 3), (1, 3), (2, 2), (3, 2), (4, 2)
B)  (0, 2), (1, 2), (2, 2), (3, 0), (4, 1)
C)  (0, 3), (1, 3), (2, 1), (3, 2), (4, 4)
D)  **(0, 3), (1, 3), (2, 2), (3, 2), (4, 4)**

**Target character:** 9

```
9,3,5,2,4,7,6
5,4,2,9,7,6,3
5,7,6,9,2,4,3
3,4,7,2,5,6,9
3,6,7,4,2,9,5
2,6,9,4,7,5,3
3,5,2,4,6,9,7
```

**Choices**
A)  (0, 3), (1, 3), (2, 3), (3, 6), (4, 5), (5, 3), (6, 2)
B)  **(0, 0), (1, 3), (2, 3), (3, 6), (4, 5), (5, 2), (6, 5)**
C)  (0, 0), (1, 3), (2, 3), (3, 6), (4, 5), (5, 2), (6, 3)
D)  (0, 0), (1, 1), (2, 3), (3, 6), (4, 5), (5, 2), (6, 5)

**Target character:** k

```
l,c,k,d,z
l,d,z,c,k
d,c,l,z,k
c,z,k,d,l
d,l,k,c,z
```

**Choices**
A)  **(0, 2), (1, 4), (2, 4), (3, 2), (4, 2)**
B)  (0, 1), (1, 1), (2, 4), (3, 2), (4, 1)
C)  (0, 2), (1, 4), (2, 4), (3, 2), (4, 4)
D)  (0, 2), (1, 4), (2, 4), (3, 1), (4, 2)

Figure 17: **Selective attention (SAtt) with text inputs:** What are the (row, column) grid locations of the target character? The top-left element of the grid is (0, 0).

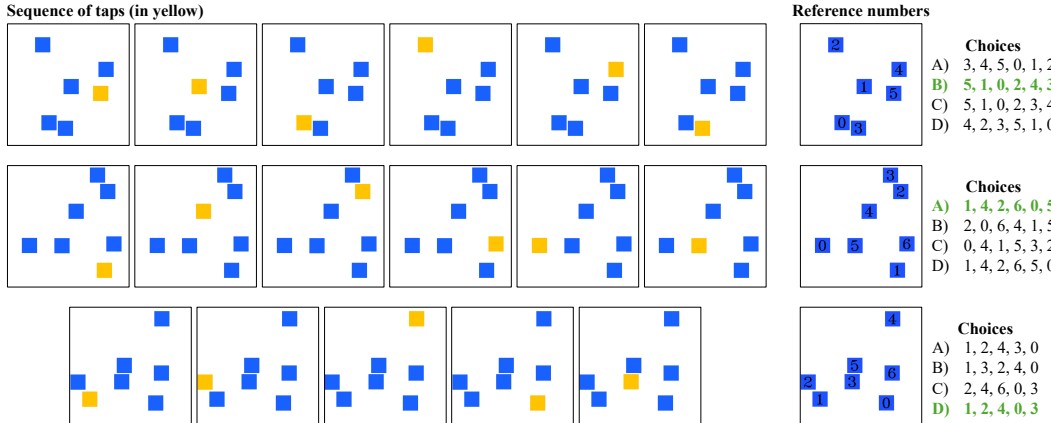

Figure 18: **Corsi block tapping (CBTT) with visual inputs:** Each row shows a set of blue boxes. The boxes are tapped in a sequence (highlighted in yellow). What is the sequence of taps? Use the box ids in the reference.

**Sequence of taps ("T")**

```
0,0,0,0,0,0   0,0,0,0,0,0   0,0,0,0,0,0   0,0,0,0,0,0   0,0,0,0,0,0   0,0,0,0,0,0
0,0,0,0,0,0   0,0,0,0,0,0   0,0,0,0,0,0   0,0,0,0,0,0   0,0,0,0,0,0   0,0,0,0,0,0
0,0,B,0,0,B,0 0,0,B,0,0,B,0 0,0,T,0,0,B,0 0,0,B,0,0,B,0 0,0,B,0,0,B,0 0,0,6,0,0,1,0
0,0,0,0,0,B,0 0,0,0,0,0,B,0 0,0,0,0,0,B,0 0,0,0,0,0,T,0 0,0,0,0,0,B,0 0,0,0,0,0,5,0
0,0,0,0,0,0,B 0,0,0,0,0,0,T 0,0,0,0,0,0,B 0,0,0,0,0,0,B 0,0,0,0,0,0,B 0,0,0,0,0,0,2
0,0,0,0,0,0,0 0,0,0,0,0,0,0 0,0,0,0,0,0,0 0,0,0,0,0,0,0 0,0,0,0,0,0,0 0,0,0,0,0,0,0
B,T,0,0,0,0,0 B,B,0,0,0,0,0 B,B,0,0,0,0,0 B,B,0,0,0,0,0 T,B,0,0,0,0,0 3,4,0,0,0,0,0
```

Reference numbers

Choices
A) 2, 3, 6, 5, 4
B) 4, 2, 6, 3, 5
C) 1, 3, 4, 5, 2
D) **4, 2, 6, 5, 3**

```
0,0,0,0,0,0   0,0,0,0,0,0   0,0,0,0,0,0   0,0,0,0,0,0   0,0,0,0,0,0   0,0,0,0,0,0
0,B,0,0,0,0   0,B,0,0,0,0   0,B,0,0,0,0   0,B,0,0,0,0   0,B,0,0,0,0   0,3,0,0,0,0
0,0,0,0,0,B,0 0,0,0,0,0,B,0 0,0,0,0,0,T,0 0,0,0,0,0,B,0 0,0,0,0,0,0,0 0,0,0,0,0,4,0
0,B,0,0,0,0,B 0,B,0,0,0,0,B 0,B,0,0,0,0,B 0,B,0,0,0,0,T 0,T,0,0,0,0,B 0,2,0,0,0,0,1
0,0,0,0,0,0,0 0,0,0,0,0,0,0 0,0,0,0,0,0,0 0,0,0,0,0,0,0 0,0,0,0,0,0,0 0,0,0,0,0,0,0
T,0,0,B,0,0,0 B,0,0,T,0,0,0 B,0,0,B,0,0,0 B,0,0,B,0,0,0 B,0,0,B,0,0,0 5,0,0,6,0,0,0
0,0,0,0,0,0,0 0,0,0,0,0,0,0 0,0,0,0,0,0,0 0,0,0,0,0,0,0 0,0,0,0,0,0,0 0,0,0,0,0,0,0
```

Choices
A) 2, 1, 5, 4, 6
B) **5, 6, 4, 1, 2**
C) 5, 6, 4, 2, 1
D) 2, 4, 1, 3, 5

```
            0,B,0,0,0   0,T,0,0,0   0,B,0,0,0   0,B,0,0,0   0,1,0,0,0
            B,0,0,0,0   B,0,0,0,0   B,0,0,0,0   B,0,0,0,0   5,0,0,0,0
            0,0,0,B,B   0,0,0,B,B   0,0,0,T,B   0,0,0,B,T   0,0,0,7,6
            0,T,B,0,0   0,B,B,0,0   0,B,B,0,0   0,B,B,0,0   0,2,4,0,0
            0,0,0,0,B   0,0,0,0,B   0,0,0,0,B   0,0,0,0,B   0,0,0,0,3
```

Choices
A) **2, 1, 7, 6**
B) 2, 7, 6, 1
C) 2, 1, 6, 7
D) 1, 3, 7, 6

Figure 19: **Corsi block tapping (CBTT) with text inputs:** Each row shows a set of boxes (**B**) laid out in space. This is shown as a 2D array where **0** is empty space. The boxes are tapped in a sequence (highlighted as **T**). What is the sequence of taps? Use the box ids in the reference. *The array colors are purely for illustration purposes.*

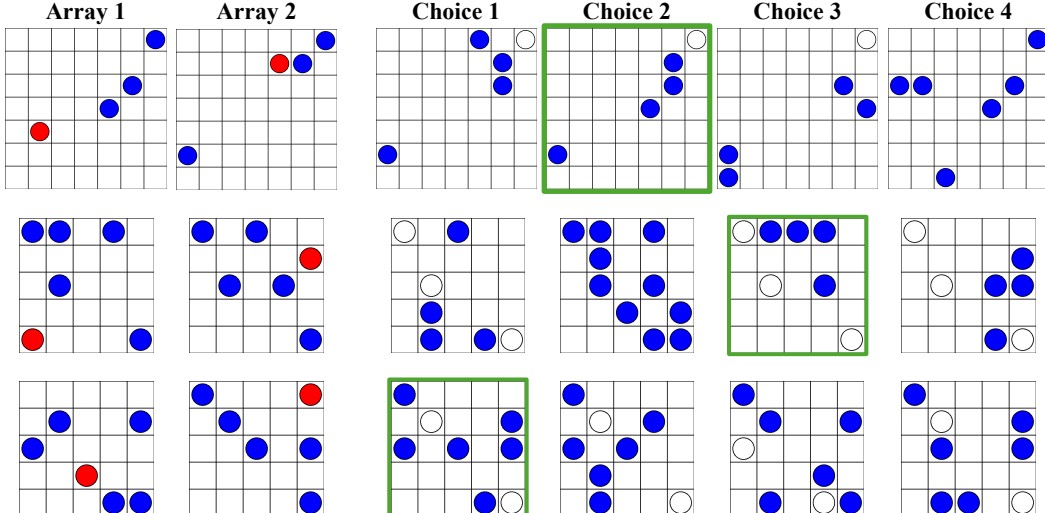

Figure 20: **Spatial addition (SAdd) with visual inputs:** What is the sum of the two arrays? Ignore red circles. Blue circles represent 1, white circles represent 2 and empty spaces represent 0.

| Array 1 | Array 2 | Choice 1 | Choice 2 | Choice 3 | Choice 4 |
|---|---|---|---|---|---|
| E,**B**,E,**R**,E | E,**B**,E,E,E | E,**W**,E,E,E | E,**B**,E,E,E | E,**W**,E,E,E | E,**W**,E,E,E |
| E,E,E,E,E | E,**B**,E,E,E | E,**B**,E,E,E | E,E,E,E,E | E,E,E,E,**B** | E,**B**,E,E,E |
| E,E,E,E,E | E,E,E,E,E | E,E,E,E,E | E,E,E,E,E | E,E,E,**B**,E | E,E,E,E,E |
| E,E,**B**,E,E | E,**R**,E,E,E | E,E,E,**B**,E | E,E,**W**,E,E | E,E,E,E,E | E,E,**B**,E,E |
| E,E,E,E,E | E,E,E,E,E | E,E,E,E,E | E,E,E,E,**B** | E,E,E,E,E | E,E,E,E,E |
| | | | | | |
| **B**,E,E | **B**,E,E | **W**,E,E | **B**,E,E | **B**,E,E | **B**,E,E |
| E,E,**R** | E,E,E | E,E,E | E,E,**B** | E,E,E | E,**B**,E |
| E,E,E | E,E,**R** | E,E,E | E,E,E | **B**,E,E | E,E,E |
| | | | | | |
| **R**,E,E,**B**,E,E,E,**B** | E,E,E,**B**,E,E,E,**B**,E | E,E,E,**W**,E,E,E,**B**,E | E,E,E,**W**,E,E,E,**B**,**B** | **B**,E,E,**W**,E,**B**,E,**B**,**B** | E,**B**,E,**B**,E,**B**,E,E,**B** |
| E,E,E,E,E,E,E,**B**,E | E,E,E,E,E,E,E,E,E | E,E,E,E,E,E,E,**B**,E | E,E,E,E,E,E,E,**B**,E | E,E,E,E,E,E,E,**B**,E | E,E,**B**,E,E,E,E,**B**,E |
| E,E,E,E,**B**,E,E,E,E | E,E,E,**B**,**B**,E,E,E,E | E,E,E,**B**,**W**,**B**,E,E,E | E,E,E,E,**B**,**W**,E,E,E,E | E,E,E,**B**,**W**,E,E,E,E | E,E,E,E,**B**,E,E,E,E |
| E,E,E,E,**B**,E,E,**B**,E | E,E,**B**,E,**B**,E,E,**B**,E | E,E,**B**,E,**W**,E,E,**W**,E | E,E,**B**,E,**W**,E,E,**W**,E | E,E,**B**,E,**W**,E,E,**W**,E | **B**,E,E,E,E,**B**,E,E,**B**,E |
| E,**B**,E,E,E,E,E,E,E | E,E,E,E,E,E,E,E,E | E,**B**,E,E,E,E,E,E,E | E,**B**,E,E,E,E,E,E,E | E,**B**,E,E,E,E,E,E,E | E,**W**,E,**B**,E,E,E,E,E |
| E,E,E,E,E,E,E,E,E | E,E,E,E,E,**R**,E,E,E | E,E,E,E,E,E,E,E,E | E,E,E,E,E,E,E,E,E | E,E,E,E,E,E,E,E,E | E,E,E,E,E,E,E,E,E |
| E,E,E,**B**,E,E,E,E,**B** | E,E,E,E,E,E,**B**,E,E | E,E,E,**B**,E,E,E,**B**,**B** | E,E,E,**B**,E,E,**B**,E,**B** | E,E,E,**B**,E,E,**B**,E,E | E,E,E,**B**,E,E,E,E,**B** |
| E,E,E,E,E,E,E,E,E | E,E,E,E,E,**B**,E,E,E | E,E,E,E,E,E,**B**,E,E | E,E,E,E,**B**,E,E,E,E | E,E,E,E,E,E,E,E,E | E,**B**,E,E,E,E,E,E,E |
| E,E,E,E,E,E,E,E,E | E,E,E,E,E,E,E,E,E | E,E,E,E,E,E,E,E,E | E,E,E,E,E,E,E,E,E | E,E,E,E,E,E,E,E,E | E,E,E,E,**B**,E,E,**B**,E |

Figure 21: **Spatial addition (SAdd) with text inputs:** What is the sum of the two arrays? Ignore **R**. **E** is 0, **B** is 1 and **W** is 2. *The array colors are purely for illustration purposes.*

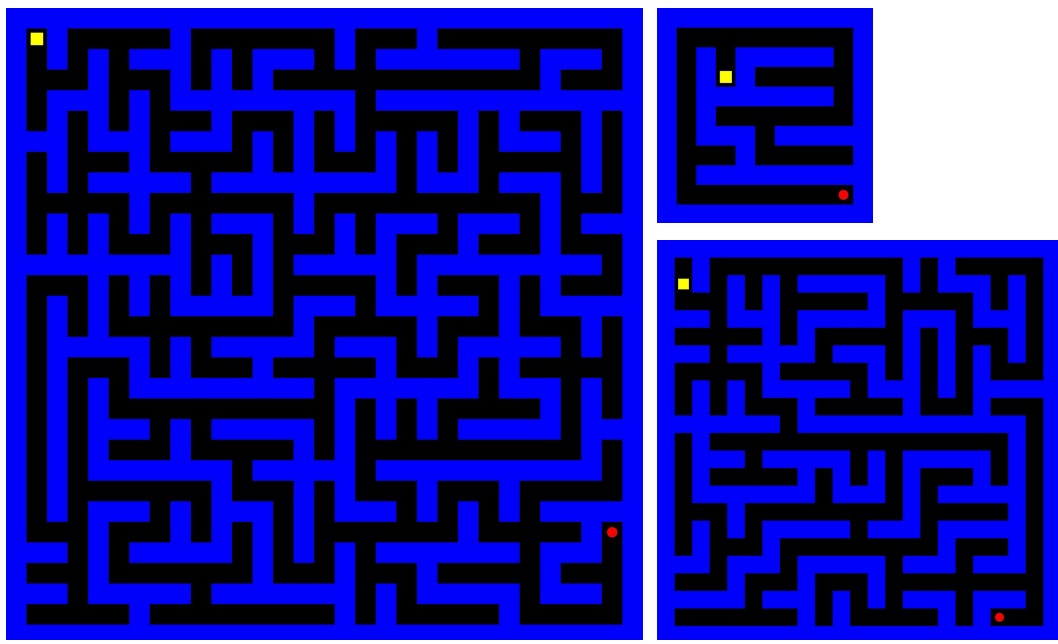

Figure 22: **Maze completion (MCT) with visual inputs:** We illustrate examples of mazes used for the MCT task. We programmatically generate mazes of different sizes using Mazelib (Stilley, 2014). Blue cells are obstacles. Black cells are navigable space. The yellow square represents the current location. The red circle represents the goal location.

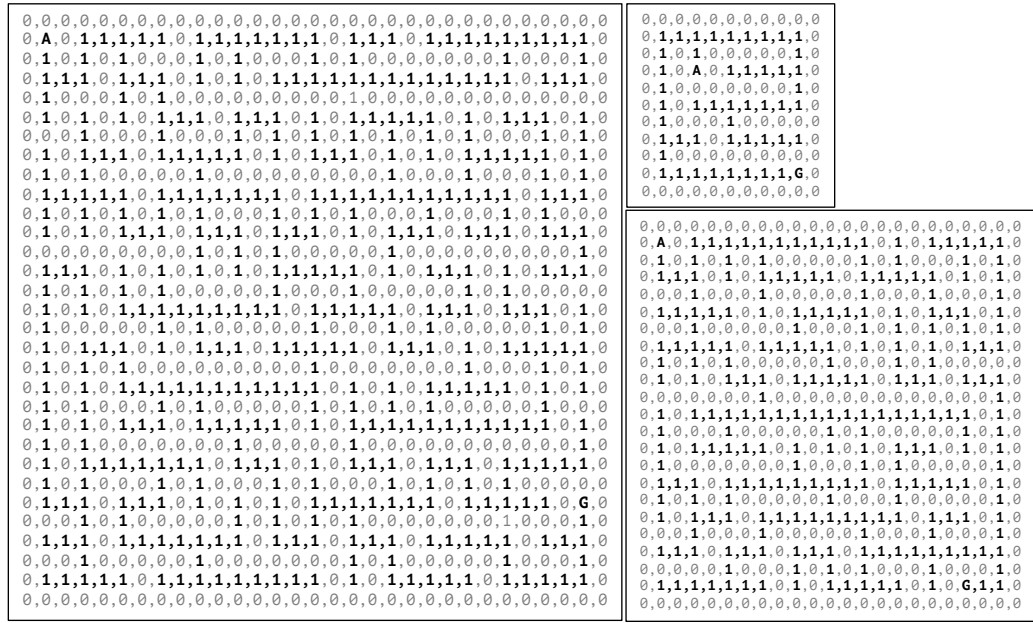

Figure 23: **Maze completion (MCT) with text inputs:** We illustrate examples of mazes used for the MCT task. We programmatically generate mazes of different sizes using Mazelib (Stilley, 2014). **0**s are obstacles. **1**s are navigable space. **A** represents the current location. **G** represents the goal location. *The array colors are purely for illustration purposes.*

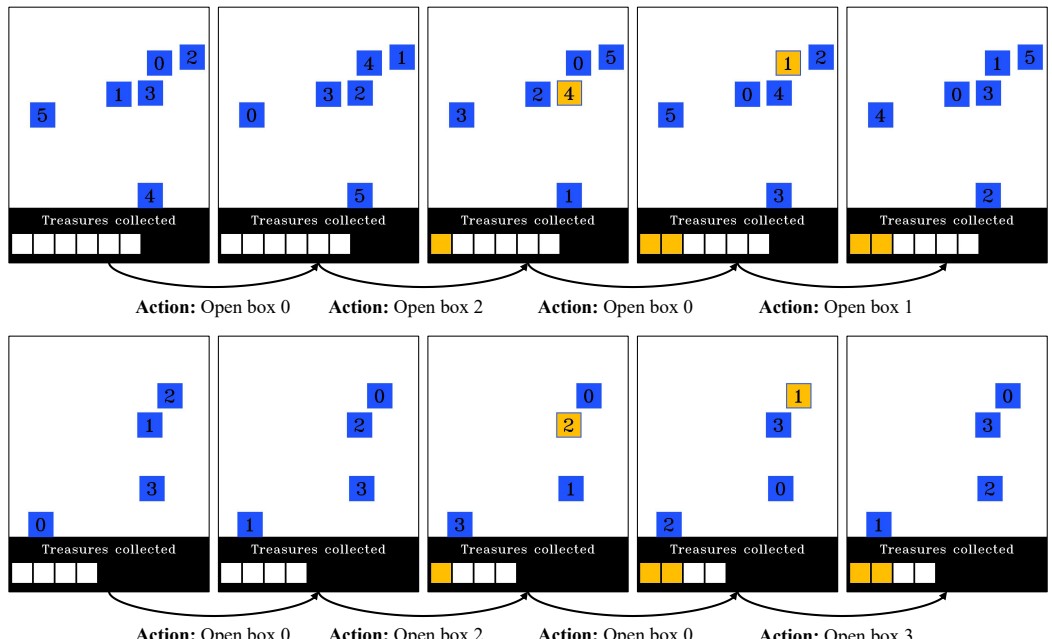

Figure 24: **Cambridge spatial working memory (CSWM) with visual inputs:** We illustrate two game plays of the CSWM task in the two rows. In each row, we show the initial observation followed by actions taken and the resulting observations. Note how the box identities change after each step. This is intended to force models to remember boxes by their spatial locations instead of their integer identities. As treasures get collected, they are populated in the "Treasures collected" section of the game screen. When a treasure is collected, a new treasure is placed in one of the boxes where the treasure never appeared before.

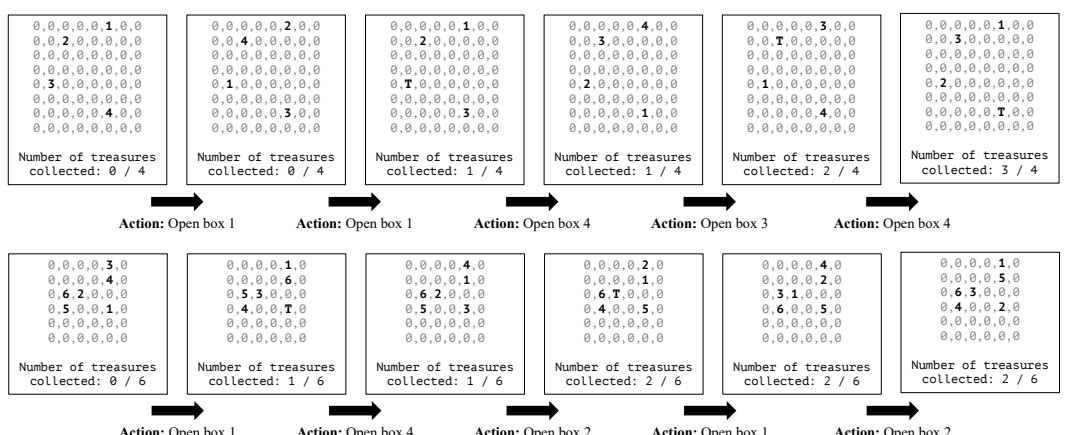

Figure 25: **Cambridge spatial working memory (CSWM) with text inputs:** We illustrate two game plays of the CSWM task in the two rows. In each row, we show the initial observation (provided as text arrays) followed by actions taken and the resulting observations. The boxes in each array are the non-zero elements. Note how the box identities change after each step. This is intended to force models to remember boxes by their spatial locations instead of their integer identities. As treasures get collected, the "Number of treasures collected" gets incremented. When a treasure is collected, a new treasure is placed in one of the boxes where the treasure never appeared before. *The array colors are purely for illustration purposes.*

**Prompt 1: Direction estimation (Ego image)**

**USER:** You are a sentient AI system capable of visually understanding the physical world, performing spatial reasoning, remembering landmarks in the world and navigating in it. Here is a video taken in a physical enviornment as you were navigating in it. Understand the environment you are navigating in, build a map of the environment to keep track of your position as well as locations of landmarks in the environment. Landmarks are paintings of objects hung on the walls.

```
<IMAGE 1>
<IMAGE 2>
.
.
.
```

Pretend that you are standing facing the painting of a Shopping_Cart. See image below.

```
<IMAGE OF Shopping_Cart>
```

At what direction (in angles from -180 to 180 degrees) is the painting of a Lawn_Mower relative to you? See image below.

```
<IMAGE OF Lawn_Mower>
```

Here are your choices: 1) -39  2) 111  3) 81  4) -69

Think step by step. Then answer your question in the following json format.

```
{
    "answer": <fill in one of 1/2/3/4 integer choice value>
}
```

**Prompt 2: Direction estimation (DM image)**

**USER:** You are playing a game in a 2D world. Each image shows the immediate surroundings around you, with you at the center of the image (in yellow). The black cells are obstacles, i.e., you cannot move over them. The blue cells are navigable spaces that you can move over. Some blue cells have landmarks in them (red circles with a text label). These are important to remember. Here is a video taken in the 2D world as you were navigating in it. Understand the 2D world you are navigating in, build a map of the world to keep track of your position as well as locations of landmarks in the world.

```
<IMAGE 1>
<IMAGE 2>
.
.
.
```

You must now answer a question based on your understanding of the 2D world. Pretend that you are standing next to the landmark A. See image below.

```
<IMAGE OF A>
```

What is the angle between the line connecting your current location to the landmark A and the line connecting your current location to the landmark C? Angles range from -180 to 180 degrees. Here are your choices: 1) 156 2) 96 3) 66 4) -24 Think step by step. Then answer your question in the following json format.

```
{
    "answer": <fill in one of 1/2/3/4 integer choice value>
}
```

**Large-scale spatial cognition**

| | | Direction est. | Distance est. | Map sketching | Route retracing | Shortcut discovery |
|---|---|---|---|---|---|---|
| Ego image | # questions | 150 | 135 | 30 | 30 | 30 |
| | # videos | 30 | 30 | 30 | 30 | 30 |
| DM image | # questions | 150 | 135 | 30 | 30 | 30 |
| | # videos | 30 | 30 | 30 | 30 | 30 |
| DM text | # questions | 150 | 135 | 30 | 30 | 30 |

**Small-scale spatial cognition**

| | | MRT | PTT | WLT | MPFB | JLO | SAtt | MCT | CBTT | SAdd | CSWM |
|---|---|---|---|---|---|---|---|---|---|---|---|
| Visual | # questions | 172 | 100 | 50 | 50 | 50 | 100 | 45 | 50 | 50 | 50 |
| | # images | 139 | 20 | 300 | 250 | 51 | 200 | - | 297 | 300 | - |
| Textual | # questions | 40 | 100 | - | 50 | 50 | 100 | 45 | 50 | 50 | 50 |

Table 6: **SPACE benchmark statistics:** We show the number of questions, images, and videos for each SPACE task. For large-scale spatial cognition tasks, we have one video per environment. We generate questions and navigation tasks based on these videos. Some small-scale spatial cognition tasks have multiple images for the same question (e.g., MPFB, WLT, SAtt and CBTT), while other tasks have multiple questions for the same image (e.g., PTT, MRT). For interactive tasks like MCT, CSWM, route retracing and shortcut discovery, images are rendered conditioned on the actions taken by the agent.

**Prompt 3: Direction estimation (DM text)**

**USER:** You are playing a game in a 2D text world. The console of the game is represented as a comma-separated text array. Obstacles are represented using 0, i.e., you cannot move over them. Navigable spaces that you can move over are represented using 1. Some navigable spaces have landmarks represented as an ascii character (A - Z). These are also navigable spaces and are just labeled with an ascii character. These landmarks are important to remember. You will always be located at the center of the array with your position highlighted using the "*" character. Here is a sequence of console screen recordings taken as you were navigating in the 2D text world. Understand the world you are navigating in, build a map of the world to keep track of your position as well as locations of landmarks in the world.

```
========== Start of console screen recordings ==========
Screen at time = 0

0,0,0,0,0
0,0,0,0,0
0,1,*,1,1
0,C,1,1,1
0,0,1,1,0

Screen at time = 1

0,0,0,0,0
0,0,0,0,0
0,0,*,1,1
0,0,C,1,1
0,0,0,1,1

.
.
.
========== End of console screen recordings ==========
```

You must now answer a question based on your understanding of the 2D text world. Pretend that you are standing next to the landmark B as shown below.

```
1,1,1,1,1
1,1,1,1,1
1,B,*,1,1
1,0,1,1,0
1,1,1,1,1
```

What is the angle between the line connecting your current location to the landmark B and the line connecting your current location to the landmark C? Angles range from -180 to 180 degrees. Note that you may not see landmark C in your immediate vicinity. You must use spatial knowledge from the sequence of screen recordings to locate your current position and both landmarks to answer this question.

Here are your choices: 1) -127 2) 53 3) 83 4) -97

Think step by step. Then answer your question in the following json format.

```
```
{
    "answer": <fill in one of 1/2/3/4 integer choice value>
}
```
```

**Prompt 4: Distance estimation (Ego image)**

**USER:** You are a sentient AI system capable of visually understanding the physical world, performing spatial reasoning, remembering landmarks in the world and navigating in it. Here is a video taken in a physical enviornment as you were navigating in it. Understand the environment you are navigating in, build a map of the environment to keep track of your position as well as locations of landmarks in the environment. Landmarks are paintings of objects hung on the walls.

```
<IMAGE 1>
<IMAGE 2>
.
.
.
```

Pretend that you are standing facing the painting of a Shopping_Cart. See image below.

```
<IMAGE OF Shopping_Cart>
```

What are the euclidean distances (in meters) to each of the following landmarks from your current position?

Landmark: painting of a Guitar

```
<IMAGE OF Guitar>
```

Landmark: painting of a Horse_Cart

```
<IMAGE OF Horse_Cart>
```

Landmark: painting of a Lawn_Mower

```
<IMAGE OF Lawn\_Mower>
```

Landmark: painting of a Hammer

```
<IMAGE OF Hammer>
```

Landmark: painting of a Soccer_Ball

```
<IMAGE OF Soccer\_Ball>
```

Here are your choices:
1) 2.2, 4.0, 5.7, 5.3, 5.0
2) 5.7, 4.0, 2.2, 5.3, 5.0
3) 1.7, 2.0, 3.3, 2.2, 1.0
4) 4.0, 5.0, 2.2, 5.7, 5.3

Think step by step. Then answer your question in the following json format.

```
{
    "answer": <fill in one of 1/2/3/4 integer choice value>
}
```

**Prompt 5: Distance estimation (DM image)**

**USER:** You are playing a game in a 2D world. Each image shows the immediate surroundings around you, with you at the center of the image (in yellow). The black cells are obstacles, i.e., you cannot move over them. The blue cells are navigable spaces that you can move over. Some blue cells have landmarks in them (red circles with a text label). These are important to remember. Here is a video taken in the 2D world as you were navigating in it. Understand the 2D world you are navigating in, build a map of the world to keep track of your position as well as locations of landmarks in the world.

```
<IMAGE 1>
<IMAGE 2>
.
.
.
```

You must now answer a question based on your understanding of the 2D world. Pretend that you are standing on the landmark C. What are the euclidean distances (in meters) from landmark C to each of the following landmarks: B, A, N, O, Y? Assume that each grid square (white borders) is 1m x 1m in size. Here are your choices: 1) 4.5, 8.5, 11.4, 11.2, 10.0 2) 11.4, 8.5, 4.5, 11.2, 10.0 3) 10.0, 8.5, 4.5, 11.4, 11.2 4) 12.5, 6.0, 3.4, -3.5, 7.2

Think step by step. Then answer your question in the following json format.

```
{
    "answer": <fill in one of 1/2/3/4 integer choice value>
}
```

**Prompt 6: Distance estimation (DM text)**

---

**USER:** You are playing a game in a 2D text world. The console of the game is represented as a comma-separated text array. Obstacles are represented using 0, i.e., you cannot move over them. Navigable spaces that you can move over are represented using 1. Some navigable spaces have landmarks represented as an ascii character (A - Z). These are also navigable spaces and are just labeled with an ascii character. These landmarks are important to remember. You will always be located at the center of the array with your position highlighted using the "*" character. Here is a sequence of console screen recordings taken as you were navigating in the 2D text world. Understand the world you are navigating in, build a map of the world to keep track of your position as well as locations of landmarks in the world.

```
========== Start of console screen recordings ==========
Screen at time = 0

0,0,0,0,0
0,0,0,0,0
0,1,*,1,1
0,C,1,1,1
0,0,1,1,0

Screen at time = 1

0,0,0,0,0
0,0,0,0,0
0,0,*,1,1
0,0,C,1,1
0,0,0,1,1

.
.
.
========== End of console screen recordings ==========
```

You must now answer a question based on your understanding of the 2D text world. Pretend that you are standing on the landmark C. What are the euclidean distances (in meters) from landmark C to each of the following landmarks: B, A, N, O, Y? Assume that each array location is 1m x 1m in size.

Here are your choices: 1) 4.5, 8.5, 11.4, 11.2, 10.0 2) 10.0, 19.2, 16.5, 12.5, 11.4 3) 11.4, 8.5, 4.5, 11.2, 10.0 4) 11.2, 11.4, 8.5, 4.5, 10.0

Think step by step. Then answer your question in the following json format.

```
{
    "answer": <fill in one of 1/2/3/4 integer choice value>
}
```

---

**Prompt 7: Map sketching (Ego image)**

**USER:** You are a sentient AI system capable of visually understanding the physical world, performing spatial reasoning, remembering landmarks in the world and navigating in it. Here is a video taken in a physical enviornment as you were navigating in it. Understand the environment you are navigating in, build a map of the environment to keep track of your position as well as locations of landmarks in the environment. Landmarks are paintings of objects hung on the walls.

```
<IMAGE 1>
<IMAGE 2>
.
.
.
```

You must sketch a map of the environment with the locations of the start, goal and landmark locations. To refresh your memory, here are the landmarks present in the environment.

Landmark: Soccer_Ball

```
<IMAGE OF Soccer_Ball>
```

Landmark: Shopping_Cart

```
<IMAGE OF Shopping_Cart>
```

Landmark: Lawn_Mower

```
<IMAGE OF Lawn_Mower>
```

Landmark: Horse_Cart

```
<IMAGE OF Horse_Cart>
```

Landmark: Guitar

```
<IMAGE OF Guitar>
```

Landmark: Hammer

```
<IMAGE OF Hammer>
```

Follow these map conventions. Your initial heading direction in the video must be along the Y axis (upward). Which of these map sketches best capture the structure of the environment?

Choice 1

```
<SKETCH IMAGE OF Choice 1>
```

Choice 2

```
<SKETCH IMAGE OF Choice 2>
```

Choice 3

```
<SKETCH IMAGE OF Choice 3>
```

Choice 4

```
<SKETCH IMAGE OF Choice 4>
```

Think step by step. Then answer your question in the following json format.

```
{
    "answer": <fill in one of 1/2/3/4 integer choice value>
}
```

**Prompt 8: Map sketching (DM image)**

**USER:** You are playing a game in a 2D world. Each image shows the immediate surroundings around you, with you at the center of the image (in yellow). The black cells are obstacles, i.e., you cannot move over them. The blue cells are navigable spaces that you can move over. Some blue cells have landmarks in them (red circles with a text label). These are important to remember. Here is a video taken in the 2D world as you were navigating in it. Understand the 2D world you are navigating in, build a map of the world to keep track of your position as well as locations of landmarks in the world.

```
<IMAGE 1>
<IMAGE 2>
.
.
.
```

You must now sketch a map of the environment with the locations of the start and landmark locations. Use your understanding of the 2D world. Which of these map sketches best capture the true structure of the 2D world?

Choice 1

```
<SKETCH IMAGE OF Choice 1>
```

Choice 2

```
<SKETCH IMAGE OF Choice 2>
```

Choice 3

```
<SKETCH IMAGE OF Choice 3>
```

Choice 4

```
<SKETCH IMAGE OF Choice 4>
```

Think step by step. Then answer your question in the following json format.

```
{
    "answer": <fill in one of 1/2/3/4 integer choice value>
}
```

**Prompt 9: Map sketching (DM text)**

**USER:** You are playing a game in a 2D text world. The console screen of the game is represented as a comma-separated text array. Obstacles are represented using 0, i.e., you cannot move over them. Navigable spaces that you can move over are represented using 1. Some navigable spaces have landmarks represented as an ascii character (A - Z). These are also navigable spaces and are just labeled with an ascii character. These landmarks are important to remember. You will always be located at the center of the array with your position highlighted using the "*" character. Here is a sequence of console screen recordings taken as you were navigating in the 2D text world. Understand the world you are navigating in, build a map of the world to keep track of your position as well as locations of landmarks in the world.

```
========== Start of console screen recordings ==========
Screen at time = 0

0,0,0,0,0
0,0,0,0,0
0,1,*,1,1
0,C,1,1,1
0,0,1,1,0

Screen at time = 1

0,0,0,0,0
0,0,0,0,0
0,0,*,1,1
0,0,C,1,1
0,0,0,1,1

.
.
.
========== End of console screen recordings ==========
```

You must now sketch a map of the environment with the locations of the start and landmark locations. Use your understanding of the 2D text world.

Which of these map sketches best capture the true structure of the 2D world? The map sketches are 2D arrays with markers highlighting the landmarks (ascii characters from A to Z) and the start location (marked as *). Locations in the map sketch with 0s are empty and can be ignored.

Choice 1

```
<TEXT ARRAY OF Choice 1>
```

Choice 2

```
<TEXT ARRAY OF Choice 2>
```

Choice 3

```
<TEXT ARRAY OF Choice 3>
```

Choice 4

```
<TEXT ARRAY OF Choice 4>
```

Think step by step. Then answer your question in the following json format.

```
{
    "answer": <fill in one of 1/2/3/4 integer choice value>
}
```

**Prompt 10: Route retracing (Ego image)**

**SYSTEM:** You are a sentient living creature capable of navigating in environments, building internal spatial representations of environments, and finding goals in them. You will be shown a video of the shortest route from the initial position to the goal. You must look at the video and understand the environment structure and the route taken. Then, you will be placed in the environment at the same initial position. You must navigate from the initial position to the goal using the same route shown in the video, as quickly as possible. Below, you will find sections highlighting more details about the task. You can refer to these for more information.

OBSERVATIONS:
The images are recorded from a perspective viewpoint (i.e., egocentric or first-person). This means that you are likely to see objects from different angles, resulting in a skewed appearance of the underlying 3D objects. It is important for you to look past this skew in the appearance and percive the true shape of the object in 3D.

GOAL:
You will be provided an object goal using a text description and an image of the object. You must find the goal object in the environment by repeating the path shown in the video walkthrough. Once you find it, move close to the location of the goal and re-orient yourself to face the object.

ACTIONS:
You have four actions available.
move_forward: move forward by 0.25m along the current heading direction. It does not change the heading angle.
turn_left: decrease your heading angle by 30 degrees. It does not change the (x, y) position.
turn_right: increase your heading angle by 30 degrees. It does not change the (x, y) position.
stop: ends the current task. Issue this action only if you think you have reached the goal. If you haven't reached the goal, this action will result in a navigation failure that cannot be recovered from.

STUCK IN PLACE BEHAVIOR:
Avoid getting stuck in one place, i.e., do not alternate between left and right turns without going anywhere. You must try and move around consistently without being stuck in one place.

STOP CRITERIA:
Before executing stop, you must ensure that you've "reached" the goal correctly. To reach a goal, you have to move close enough to the wall where you see the goal, and see the object clearly in your observation in front of you.

RESPONSE FORMAT:
Respond in the following format:
Reasoning: text explanation string in one or two short sentences — provide all your explanations and inner thoughts here - avoid verbosity and be concise
Intent: state your intent in one short sentence, i.e., what you are trying to achieve
Then provide the final action to take in a json formatted string.
```
{
    "action": <action name -- must be one of
    move_forward, turn_left, turn_right, stop>
}
```

**USER:** Here are the sequence of frames from the walkthrough video demonstrating the route you need to take. Analyze the walkthrough to understand the movements and the maze structure. Take a note of all the details needed to help you repeat this route when navigating next. Think step by step.

```
<IMAGE 1>
<IMAGE 2>
.
.
.
```

**ASSISTANT:** ...

**USER:** Now, you must navigate to the goal. Here is the goal description and the image: Painting of a Soccer_Ball

```
<IMAGE OF Soccer_Ball>
```

**USER:** Here is the current observation.

```
<CURRENT IMAGE OBSERVATION>
```

**ASSISTANT:** ...

**USER:** Here is the current observation.

```
<CURRENT IMAGE OBSERVATION>

.
.
.
```

**Prompt 11: Route retracing (DM image)**

**SYSTEM:** You are playing a game in a 2D world. You will be shown a video of the shortest route from an initial position to a goal. You must look at the video and understand the 2D world structure and the route taken. Then, you will be placed in the 2D world at the same initial position. You must navigate from the initial position to the goal using the same route shown in the video, as quickly as possible. Below, you will find sections highlighting more details about the 2D world and the task. You can refer to these for more information.

2D WORLD:
The world consists of the following.
* black cells: these are obstacles, i.e., you cannot move over them
* blue cells: these are navigable spaces, i.e., you can move over them
Some blue cells contain landmarks, which are red circles filled with a text character. These are important as they will allow you to better understand the world and locate yourself. Your position will be marked using a yellow square.

OBSERVATIONS:
The images are recorded from a birds-eye view of the 2D world. The images capture a local neighborhood surrounding your current position in the world, i.e., you will always remain at the center of the image while the world changes around you.

GOAL:
You will be asked to navigate to a goal landmark. You must find the goal in the 2D world by repeating the path shown in the video. Once you find it, move to the location of the goal till you are standing on the landmark and then execute a stop action.

ACTIONS:
You have four actions available.
up: move up by one unit cell
down: move down by one unit cell
left: move left by one unit cell
right: move right by one unit cell
stop: ends the current task. Issue this action only if you think you have reached the goal. If you haven't reached the goal, this action will result in a navigation failure that cannot be recovered from.

STOP CRITERIA:
Before executing stop, you must ensure that you've "reached" the goal correctly. To reach a goal, you have to move to the cell containing the goal landmark. Then execute the stop action.

RESPONSE FORMAT:
Respond in the following format:
Reasoning: text explanation string in one or two short sentences — provide all your explanations and inner thoughts here - avoid verbosity and be concise
Intent: state your intent in one short sentence, i.e., what you are trying to achieve
Then provide the final action to take in a json formatted string.

```
{
    "action": <action name -- must be one of up, down, left, right>
}
```

**USER:** Here is sequence of video frames recorded in the 2D world. This demonstrates the route you need to repeat. Analyze the video to understand the movements and the world structure. Take a note of all the details needed to help you repeat this route when navigating next. Think step by step.

<IMAGE 1>
<IMAGE 2>
.
.
.

**ASSISTANT:** ...

**USER:** Now, you must navigate to the goal based on your knowledge of the 2D world you obtained from the video. Here is the goal description: landmark Y

**USER:** Here is the local view of your surroundings in the 2D world. You are at the center of this view.

<CURRENT IMAGE OBSERVATION>

**ASSISTANT:** ...

**USER:** Here is the local view of your surroundings in the 2D world. You are at the center of this view.

<CURRENT IMAGE OBSERVATION>

.
.
.

**Prompt 12: Route retracing (DM text) — part 1**

**SYSTEM:** You are playing a game in a 2D text world. The console screen of the game is represented as a comma-separated text array. You will be shown a sequence of console screen recordings that demonstrate the shortest route from an initial position to a goal. You must look at the sequence and understand the 2D text world structure and the route taken. Then, you will be placed in the 2D text world at the same initial position. You must navigate from the initial position to the goal using the same route shown in the screen recording sequence, as quickly as possible. Below, you will find sections highlighting more details about the 2D text world and the task. You can refer to these for more information.

2D TEXT WORLD:
The console of the game is represented as a comma-separated text array. Obstacles are represented using 0, i.e., you cannot move over them. Navigable spaces that you can move over are represented using 1. Some navigable spaces have landmarks represented as an ascii character (A - Z). These are also navigable spaces and are just labeled with an ascii character. These landmarks are important to remember. You will always be located at the center of the array with your position highlighted using the "*" character.

OBSERVATIONS:
The images are recorded from a birds-eye view of the 2D world. The images capture a local neighborhood surrounding your current position in the world, i.e., you will always remain at the center of the image while the world changes around you.

GOAL:
You will be asked to navigate to a goal landmark. You must find the goal in the 2D text world by repeating the path shown in the console screen recording sequence. Once you find it, move to the location of the goal till you are standing on the landmark and then execute a stop action.

ACTIONS:
You have four actions available.
up: move up by one unit cell
down: move down by one unit cell
left: move left by one unit cell
right: move right by one unit cell
stop: ends the current task. Issue this action only if you think you have reached the goal. If you haven't reached the goal, this action will result in a navigation failure that cannot be recovered from.

STOP CRITERIA:
Before executing stop, you must ensure that you've "reached" the goal correctly. To reach a goal, you have to move to the cell containing the goal landmark. Then execute the stop action.

RESPONSE FORMAT:
Respond in the following format:
Reasoning: text explanation string in one or two short sentences — provide all your explanations and inner thoughts here - avoid verbosity and be concise Intent: state your intent in one short sentence, i.e., what you are trying to achieve Then provide the final action to take in a json formatted string.
```
{
    "action": <action name -- must be one of up, down, left, right>
}
```

**USER:** Here is the sequence of console screen recordings taken in the 2D text world. This demonstrates the route you need to repeat. Analyze the sequence to understand the movements and the world structure. Take a note of all the details needed to help you repeat this route when navigating next. Think step by step.

```
### Console screen recorded at time = 0

```
0,0,0,0,0
0,0,0,0,0
0,1,*,1,1
0,C,1,1,1
0,0,1,1,0

```

### Console screen recorded at time = 1

```
0,0,0,0,0
0,1,1,1,1
0,C,*,1,1
0,0,1,1,0
0,1,1,1,1

```

.
.
.

**Prompt 13: Route retracing (DM text) — part 2**

**ASSISTANT:** ...

**USER:** Now, you must navigate to the goal based on your knowledge of the 2D text world you obtained from the sequence of console screen recordings. Here is the goal description: landmark Y

**USER:** Here is a birds-eye view of the 5x5 area surrounding your current position. You are located at the center of this view. Your position is denoted by "*".

```
0,0,0,0,0
0,0,0,0,0
0,1,*,1,1
0,C,1,1,1
0,0,1,1,0
```

The landmarks visible in your local context are: C. Note that the landmark locations are also navigable spaces, i.e., you can move over them. Your objective is to reach landmark: Y

**ASSISTANT:** ...

**USER:** Here is a birds-eye view of the 5x5 area surrounding your current position. You are located at the center of this view. Your position is denoted by "*".

```
0,0,0,0,0
0,1,1,1,1
0,C,*,1,1
0,0,1,1,0
0,1,1,1,1
```

The landmarks visible in your local context are: C. Note that the landmark locations are also navigable spaces, i.e., you can move over them. Your objective is to reach landmark: Y

**ASSISTANT:** ...

.
.
.

**Prompt 14: Shortcut discovery (Ego image)**

**SYSTEM:** You are a sentient living creature capable of navigating in environments, building internal spatial representations of environments, and finding goals in them. You will be shown a video of some route from the initial position to the goal. You must look at the video and understand the environment structure, and remember the locations of the start and the goal. The video may show a long-winded route from the start to the goal with unnecessary detours. Based on the environment structure, you must identify a faster route to the goal. Then, you will be placed in the environment at the same initial position. You must navigate to the goal using your identified shortest route as quickly as possible. Below, you will find sections highlighting more details about the task. You can refer to these for more information.

OBSERVATIONS:
The images are recorded from a perspective viewpoint (i.e., egocentric or first-person). This means that you are likely to see objects from different angles, resulting in a skewed appearance of the underlying 3D objects. It is important for you to look past this skew in the appearance and perceive the true shape of the object in 3D.

GOAL:
You will be provided an object goal using a text description and an image of the object. You must find the goal object in the environment by identifying the shortest route based on your experience from the video. Once you find the goal, move close to its location and re-orient yourself to face the object.

ACTIONS:
You have four actions available.
move_forward: move forward by 0.25m along the current heading direction. It does not change the heading angle.
turn_left: decrease your heading angle by 30 degrees. It does not change the (x, y) position.
turn_right: increase your heading angle by 30 degrees. It does not change the (x, y) position.
stop: ends the current task. Issue this action only if you think you have reached the goal. If you haven't reached the goal, this action will result in a navigation failure that cannot be recovered from.

STUCK IN PLACE BEHAVIOR:
Avoid getting stuck in one place, i.e., do not alternate between left and right turns without going anywhere. You must try and move around consistently without being stuck in one place.

STOP CRITERIA:
Before executing stop, you must ensure that you've "reached" the goal correctly. To reach a goal, you have to move the robot close enough to the wall where you see the goal, and see the object clearly in your observation in front of you.

RESPONSE FORMAT:
Respond in the following format:
Reasoning: text explanation string in one or two short sentences — provide all your explanations and inner thoughts here - avoid verbosity and be concise
Intent: state your intent in one short sentence, i.e., what you are trying to achieve
Then provide the final action to take in a json formatted string.
```
{
    "action": <action name -- must be one of move_forward, turn_left, turn_right, stop>
}
```

**USER:** Here are the sequence of frames from the walkthrough video demonstrating a suboptimal route from the start to some goal location. Analyze the walkthrough to understand the movements and the environment structure. Keep track of the start and goal locations, and the current location in the environment as you watch the walkthrough. Then plan a shortcut route that takes you to the goal while avoiding unnecessary detours. Think step by step.

```
<IMAGE 1>
<IMAGE 2>
.
.
.
```

**ASSISTANT:** ...

**USER:** Now, you must navigate to the goal. Here is the goal description and the image: Painting of a Soccer_Ball

```
<IMAGE OF Soccer_Ball>
```

**USER:** Here is the current observation.

```
<CURRENT IMAGE OBSERVATION>
```

**ASSISTANT:** ...

**USER:** Here is the current observation.

```
<CURRENT IMAGE OBSERVATION>
```

**ASSISTANT:** ...
```
.
.
.
```

**Prompt 15: Shortcut discovery (DM image)**

**SYSTEM:** You are playing a game in a 2D world. You will be shown a video of some route from an initial position to a goal. You must look at the video and understand the 2D world structure and remember the locations of the start and the goal. The video may show a long-winded route from the start to the goal with unnecessary detours. Based on the world structure, you must identify a faster route to the goal. Then, you will be placed in the 2D world at the same initial position. You must navigate from the initial position to the goal using your identified shortest route as quickly as possible. Below, you will find sections highlighting more details about the 2D world and the task. You can refer to these for more information.

2D WORLD:
The world consists of the following.
* black cells: these are obstacles, i.e., you cannot move over them
* blue cells: these are navigable spaces, i.e., you can move over them
Some blue cells contain landmarks, which are red circles filled with a text character. These are important as they will allow you to better understand the world and locate yourself. Your position will be marked using a yellow square.

OBSERVATIONS:
The images are recorded from a birds-eye view of the 2D world. The images capture a local neighborhood surrounding your current position in the world, i.e., you will always remain at the center of the image while the world changes around you.

GOAL:
You will be asked to navigate to a goal landmark. You must find the goal in the 2D world by identifying the shortest path based your your experience from the video. Once you find it, move to the location of the goal till you are standing on the landmark and then execute a stop action.

ACTIONS:
You have four actions available.
up: move up by one unit cell
down: move down by one unit cell
left: move left by one unit cell
right: move right by one unit cell
stop: ends the current task. Issue this action only if you think you have reached the goal. If you haven't reached the goal, this action will result in a navigation failure that cannot be recovered from.

STOP CRITERIA:
Before executing stop, you must ensure that you've "reached" the goal correctly. To reach a goal, you have to move to the cell containing the goal landmark. Then execute the stop action.

RESPONSE FORMAT:
Respond in the following format:
Reasoning: text explanation string in one or two short sentences — provide all your explanations and inner thoughts here - avoid verbosity and be concise
Intent: state your intent in one short sentence, i.e., what you are trying to achieve
Then provide the final action to take in a json formatted string.
```
{
    "action": <action name -- must be one of up, down, left, right>
}
```

**USER:** Here is the sequence of video frames recorded in the 2D world. This demonstrates a suboptimal route from the start to some goal location. Analyze the video to understand the movements and the world structure. Keep track of the start and goal locations, and the current location in the world as you watch the video. Then plan a shortcut route that takes you to the goal while avoiding any unnecessary detours. Think step by step.

```
<IMAGE 1>
<IMAGE 2>
.
.
.
```

**ASSISTANT:** ...

**USER:** Now, you must navigate to the goal based on your knowledge of the 2D world you obtained from the video. Here is the goal description: landmark Y

**USER:** Here is the local view of your surroundings in the 2D world. You are at the center of this view.

```
<CURRENT IMAGE OBSERVATION>
```

**ASSISTANT:** ...

**USER:** Here is the local view of your surroundings in the 2D world. You are at the center of this view.

```
<CURRENT IMAGE OBSERVATION>
```

**ASSISTANT:** ...

```
.
.
.
```

**Prompt 16: Shortcut discovery (DM text) — part 1**

**SYSTEM:** You are playing a game in a text 2D world. The console screen of the game is represented as a comma-separated text array. You will be shown a sequence of console screen recordings that demonstrates a route from an initial position to a goal. You must look at the sequence and understand the 2D text world structure and remember the locations of the start and the goal. The recordings may show a long-winded route from the start to the goal with unnecessary detours. Based on the world structure, you must identify a faster route to the goal. Then, you will be placed in the 2D text world at the same initial position. You must navigate from the initial position to the goal using your identified shortest route as quickly as possible. Below, you will find sections highlighting more details about the 2D text world and the task. You can refer to these for more information.

2D TEXT WORLD:
The console of the game is represented as a comma-separated text array. Obstacles are represented using 0, i.e., you cannot move over them. Navigable spaces that you can move over are represented using 1. Some navigable spaces have landmarks represented as an ascii character (A - Z). These are also navigable spaces and are just labeled with an ascii character. These landmarks are important to remember. You will always be located at the center of the array with your position highlighted using the "*" character.

OBSERVATIONS:
The images are recorded from a birds-eye view of the 2D world. The images capture a local neighborhood surrounding your current position in the world, i.e., you will always remain at the center of the image while the world changes around you.

GOAL:
You will be asked to navigate to a goal landmark. You must find the goal in the 2D text world by identifying the shortest path based your your experience from the screen recording sequence. Once you find it, move to the location of the goal till you are standing on the landmark and then execute a stop action.

ACTIONS:
You have four actions available.
up: move up by one unit cell
down: move down by one unit cell
left: move left by one unit cell
right: move right by one unit cell
stop: ends the current task. Issue this action only if you think you have reached the goal. If you haven't reached the goal, this action will result in a navigation failure that cannot be recovered from.

STOP CRITERIA:
Before executing stop, you must ensure that you've "reached" the goal correctly. To reach a goal, you have to move to the cell containing the goal landmark. Then execute the stop action.

RESPONSE FORMAT:
Respond in the following format:
Reasoning: text explanation string in one or two short sentences — provide all your explanations and inner thoughts here - avoid verbosity and be concise
Intent: state your intent in one short sentence, i.e., what you are trying to achieve
Then provide the final action to take in a json formatted string.

```
{
    "action": <action name -- must be one of up, down, left, right>
}
```

**USER:** Here is the sequence of console screen recordings taken in the 2D text world. This demonstrates a suboptimal route from the start to some goal location. Analyze the sequence to understand the movements and the world structure. Keep track of the start and goal locations, and the current location in the world as you study the sequence. Then plan a shortcut route that takes you to the goal while avoiding any unnecessary detours. Think step by step.

### Console screen recorded at time = 0

```
0,0,0,0,0
0,0,0,0,0
0,1,*,1,1
0,C,1,1,1
0,0,1,1,0
```

### Console screen recorded at time = 1

```
0,0,0,0,0
0,0,0,0,0
0,0,*,1,1
0,0,C,1,1
0,0,0,1,1
```

.
.
.

**Prompt 17: Shortcut discovery (DM text) — part 2**

---

**ASSISTANT:** ...

**USER:** Now, you must navigate to the goal based on your knowledge of the 2D text world you obtained from the sequence of console screen recordings. Here is the goal description: landmark Y

**USER:** Here is a birds-eye view of the 5x5 area surrounding your current position. You are located at the center of this view. Your position is denoted by ”*”.

```
0,0,0,0,0
0,0,0,0,0
0,1,*,1,1
0,C,1,1,1
0,0,1,1,0
```

The landmarks visible in your local context are: C. Note that the landmark locations are also navigable spaces, i.e., you can move over them. Your objective is to reach landmark: Y

**ASSISTANT:** ...

**USER:** Here is a birds-eye view of the 5x5 area surrounding your current position. You are located at the center of this view. Your position is denoted by ”*”.

```
0,0,0,0,0
0,0,0,0,0
1,1,*,1,0
C,1,1,1,0
0,1,1,0,0
```

The landmarks visible in your local context are: C. Note that the landmark locations are also navigable spaces, i.e., you can move over them. Your objective is to reach landmark: Y

**ASSISTANT:** ...

---

**Prompt 18: Mental rotation (vision)**

---

**USER:** Here is an image of a three-dimensional shape.

`<IMAGE OF REFERENCE 3D SHAPE>`

Which of these images show the same object rotated in 3D?

Choice 1

`<CHOICE 1 IMAGE>`

Choice 2

`<CHOICE 2 IMAGE>`

Choice 3

`<CHOICE 3 IMAGE>`

Choice 4

`<CHOICE 4 IMAGE>`

Think step by step. Then answer your question in the following json format.

```
{
    "answer": <fill in one of 1/2/3/4 integer value>
}
```

---

**Prompt 19: Mental rotation (text)**

**USER:** Here is a two-dimensional array.

```
over,none,page
such,none,free
site,none,list
```

Which of these options show the same array rotated in 2D? Note: It must only be rotated, not mirrored.

Choice 1:

```
page,free,list
none,none,none
over,such,site
```

Choice 2:

```
list,free,page
none,none,none
site,such,over
```

Choice 3:

```
page,none,over
free,none,such
list,none,site
```

Choice 4:

```
over,such,site
none,none,none
page,free,list
```

Think step by step. Then answer your question in the following json format.

```
{
    "answer": <fill in one of 1/2/3/4 integer value>
}
```

**Prompt 20: Perspective taking (vision)**

**USER:** Here is an image of various objects (animate and inanimate) on a two-dimensional plane.

```
<IMAGE OF OBJECTS>
```

Pretend that you are standing at the centroid of guitar and facing the centroid of bat. Visualize the world around you. At what angle (from -180 to 180 degrees) is snake located relative to you? Clockwise rotations are positive and anti-clockwise rotations are negative.
Here are your options: 1) 45  2) 85  3) 105  4) 5
Think step by step. Then answer your question in the following json format.

```
{
    "answer": <fill in one of 1/2/3/4 integer value>
}
```

**Prompt 21: Perspective taking (text)**

**USER:** Here is an array of numbers representing the birds-eye view of a two-dimensional plane.

```
0,7,0,9
8,0,0,0
0,0,0,5
0,0,0,3
```

Empty locations are indicated using 0. Important locations are indicated with a number 1 - 9. Pretend that you are standing at the location 8 and facing the location 9. Visualize the world around you. At what angle (from -180 to 180 degrees) is the location 3 relative to you? Clockwise rotations are positive and anti-clockwise rotations are negative.
Here are your options: 1) 52  2) 32  3) 72  4) 112
Think step by step. Then answer your question in the following json format.

```
{
    "answer": <fill in one of 1/2/3/4 integer value>
}
```

**Prompt 22: Water level (vision)**

USER: Here is a container filled with water.

```
<IMAGE OF FILLED WATER CONTAINER>
```

What will be the water level when it is rotated as shown here?

```
<IMAGE OF ROTATED EMPTY WATER CONTAINER>
```

Here are your choices. Which of these match the expected water level in the rotated container?

Choice 1

```
<IMAGE OF CHOICE 1>
```

Choice 2

```
<IMAGE OF CHOICE 2>
```

Choice 3

```
<IMAGE OF CHOICE 3>
```

Choice 4

```
<IMAGE OF CHOICE 4>
```

Think step by step. Then answer your question in the following json format.

```
```
{
    "answer": <fill in one of 1/2/3/4 integer value>
}
```
```

**Prompt 23: Minnesota paper form board (vision)**

USER: This image shows the different pieces of a puzzle.

```
<IMAGE OF PUZZLE PIECES>
```

These pieces are put together by an oracle. Which one of these four options shows what it would look like when the pieces are put together? Pay close attention to not just the final fitted shape, but also the individual pieces contained within the shape.

Choice 1

```
<IMAGE OF CHOICE 1>
```

Choice 2

```
<IMAGE OF CHOICE 2>
```

Choice 3

```
<IMAGE OF CHOICE 3>
```

Choice 4

```
<IMAGE OF CHOICE 4>
```

Think step by step. Then answer your question in the following json format.

```
```
{
    "answer": <fill in one of 1/2/3/4 integer value>
}
```
```

**Prompt 24: Minnesota paper form board (text)**

**USER:** You are playing putting together a text jigsaw puzzle. Here are the pieces, where 0 represents the interiors of the puzzle piece and 1 represents the boundary.

```
1,1,1,1,1
1,0,0,0,1
1,0,0,0,1
1,0,0,0,1
1,0,0,0,1
1,0,0,0,1
1,1,1,1,1

1,1,1,1,1,1
1,0,0,0,0,1
1,0,0,0,0,1
1,0,0,0,0,1
1,1,1,1,1,1

1,1,1,1,1
1,0,0,0,1
1,0,0,0,1
1,1,1,1,1

1,1,1,1,1
1,0,0,0,1
1,0,0,0,1
1,1,1,1,1
```

These pieces are now put together to solve the puzzle by an oracle. Which of these four options shows what it would look like when the pieces are put together?

Choice 1:

```
<CHOICE 1 ARRAY>
```

Choice 2:

```
<CHOICE 2 ARRAY>
```

Choice 3:

```
<CHOICE 3 ARRAY>
```

Choice 4:

```
<CHOICE 4 ARRAY>
```

Pay close attention to not just the final fitted shape, but also the individual pieces contained within the shape. Also, note that the puzzle pieces may need to be rotated before fitting them together.Think step by step. Then answer your question in the following json format.

```
{
    "answer": <fill in one of 1/2/3/4 integer value>
}
```

**Prompt 25: Judgement of line orientation (vision)**

**USER:** Here is an image showing two lines. Your goal is to measure the angle between the two lines.

```
<IMAGE OF LINES>
```

Here is a legend showing a set of reference lines numbered from 1 to 11.

```
<IMAGE OF LEGEND>
```

Which of the following reference line pairs match the angle between the original lines shown in the image?
1) Lines 1 and 9
2) Lines 1 and 7
3) Lines 1 and 10
4) Lines 1 and 3
Think step by step. Then answer your question in the following json format.

```
{
    "answer": <fill in one of 1/2/3/4 integer value>
}
```

**Prompt 26: Judgement of line orientation (text)**

**USER:** Here is a reference array.

```
0,0,0,0,0,0,0,0,0,0,0,0,0,0,0,0,0,0,0,0
0,0,0,0,0,0,0,0,0,0,0,0,0,0,0,0,0,0,0,0
0,0,0,0,0,0,0,0,0,0,0,0,0,0,0,0,0,0,0,0
0,0,0,0,0,0,0,0,0,0,0,0,0,0,0,0,0,0,0,0
0,0,0,0,0,0,0,0,0,0,0,0,0,0,0,0,0,0,0,0
0,0,0,0,0,0,0,0,0,0,0,0,0,0,0,0,0,0,0,0
0,0,0,0,0,0,0,0,0,0,0,0,0,0,0,0,0,0,0,0
0,0,0,0,0,0,0,0,0,0,0,0,0,0,0,0,0,0,0,0
0,0,0,0,0,0,0,0,0,0,0,0,0,0,0,0,0,0,0,0
2,2,2,2,2,0,0,0,0,0,0,1,1,1,1,1,1,1,1
0,0,0,0,0,0,0,0,0,0,0,0,0,0,0,0,0,0,0,0
0,0,0,0,0,0,0,0,0,0,0,0,0,0,0,0,0,0,0,0
0,0,0,0,0,0,0,0,0,0,0,0,0,0,0,0,0,0,0,0
0,0,0,0,0,0,0,0,0,0,0,0,0,0,0,0,0,0,0,0
0,0,0,0,0,0,0,0,0,0,0,0,0,0,0,0,0,0,0,0
0,0,0,0,0,0,0,0,0,0,0,0,0,0,0,0,0,0,0,0
0,0,0,0,0,0,0,0,0,0,0,0,0,0,0,0,0,0,0,0
0,0,0,0,0,0,0,0,0,0,0,0,0,0,0,0,0,0,0,0
0,0,0,0,0,0,0,0,0,0,0,0,0,0,0,0,0,0,0,0
0,0,0,0,0,0,0,0,0,0,0,0,0,0,0,0,0,0,0,0
```

0s mean empty space, ignore them. There are two lines made out of 1s and 2s, respectively. Your goal is to measure the angle between the two lines. Specifically, here are four choices of arrays with two lines per array. Which one of the choices has an angle between the two lines that matches the angle between lines in the reference array?

Choice 1:

```
<ARRAY OF CHOICE 1>
```

Choice 2:

```
<ARRAY OF CHOICE 2>
```

Choice 3:

```
<ARRAY OF CHOICE 3>
```

Choice 4:

```
<ARRAY OF CHOICE 4>
```

Think step by step. Then answer your question in the following json format.

```
{
    "answer": <fill in one of 1/2/3/4 integer value>
}
```

**Prompt 27: Selective attention (vision)**

**USER:** Here is an image of a apple. Let us call this the target.

```
<IMAGE OF TARGET>
```

Here is a grid of apple images. This contains multiple instances of apple, but not all of them are the target object. The grid is indexed from top-left to bottom-right, starting from row, column = (0, 0).

```
<IMAGE OF GRID>
```

Which of these options represent the true locations of the target object in the grid? Locations are represented as (row, column).

Choice 1. (0, 3), (1, 2), (2, 2), (3, 3)
Choice 2. (0, 1), (1, 0), (2, 0), (3, 3)
Choice 3. (0, 3), (1, 2), (2, 2), (3, 1)
Choice 4. (0, 3), (1, 2), (2, 2), (3, 1)
Think step by step. Then answer your question in the following json format.

```
{
    "answer": <fill in one of 1/2/3/4 integer value>
}
```

**Prompt 28: Selective attention (text)**

**USER:** Here is a grid of numbers / letters. The grid is indexed from top-left to bottom-right, starting from row, column = (0, 0).

```
f,t,o,e,r
r,e,o,t,f
f,r,o,t,e
f,e,o,r,t
o,t,r,e,f
```

Your goal is to find all occurrences of 'e' in the grid. Which of these options represent the true locations of the where 'e' occurs in the grid? Locations are represented as (row, column).

Choice 1. (0, 3), (1, 1), (2, 4), (3, 1), (4, 3)
Choice 2. (0, 3), (1, 1), (2, 4), (3, 1), (4, 4)
Choice 3. (0, 1), (1, 1), (2, 0), (3, 2), (4, 1)
Choice 4. (0, 2), (1, 1), (2, 4), (3, 1), (4, 3)

Think step by step. Then answer your question in the following json format.

```
{
    "answer": <fill in one of 1/2/3/4 integer value>
}
```

**Prompt 29: Maze completion (vision)**

**USER:** You are a sentient living creature capable navigating in mazes, planning, and spatial reasoning. You are playing a Pacman-style maze game. You start at some random position in the maze. You must escape the maze as quickly as possible to reach the goal. You are given the game screen that shows the following:
* maze structure - blue is obstacle space, black is navigable space. You can only move on black spaces. You cannot move through blue spaces.
* your current position - yellow square
* goal position - red circle

Below the screen, a status message might appear indicating that you collided into a wall after your previous action.

Actions available: You can take five possible actions.
* left - move left from your current position by one step
* right - move right from your current position by one step
* up - move up from your current position by one step
* down - move down from your current position by one step
* stop - issue this action only after you have reached the goal position. If you execute it prematurely, you will fail. If you do not execute it after reaching the goal, you will again fail.

Response format: Respond in the following format.

```
<text explanation string – explain your reasoning concisely>
<next, provide a json formatted output with the next action>
```
```
{
    "action": "<action>"
}
```

**ASSISTANT:** ...
**USER:** Here is the current state of the maze.

```
<IMAGE OF MAZE>
```

Think step-by-step about how to reach the goal. What action do you take next?

**ASSISTANT:** ...
**USER:** Here is the current state of the maze.

```
<IMAGE OF MAZE>
```

Think step-by-step about how to reach the goal. What action do you take next?

.
.
.

**Prompt 30: Maze completion (text)**

**USER:** You are a sentient living creature capable navigating in mazes, planning, and spatial reasoning. You are playing a text-based maze game. You start at some random position in the maze. You must escape the maze as quickly as possible to reach the goal. You are given a 2D array representing the maze, which contains the following:
* maze structure - 0 is obstacle space, 1 is navigable space. You can only move on 1s (i.e., navigable spaces). You cannot move through 0s (i.e., obstacles).
* your current position - marked as A
* goal position - marked as G

Goal and current positions are always navigable spaces.

Actions available: You can take five possible actions.
* left - move left from your current position by one step
* right - move right from your current position by one step
* up - move up from your current position by one step
* down - move down from your current position by one step
* stop - issue this action only after you have reached the goal position. If you execute it prematurely, you will fail. If you do not execute it after reaching the goal, you will again fail.

Response format: Respond in the following format.

```
<Think step-by-step about what action to take next. Be concise.>
<next, provide a json formatted output with the next action>
```
```
{
    "action": "<action>"
}
```

**ASSISTANT:** ...

**USER:** Here is the current view of the maze.

```
0,0,0,0,0,0,0,0,0,0,0,0,0,0,0,0,0,0,0,0,0
0,A,0,1,1,1,1,1,1,1,1,1,1,0,1,0,1,1,1,1,0
0,1,0,1,0,1,0,1,0,0,0,0,1,0,1,0,0,0,1,0,1,0
0,1,1,1,0,1,0,1,1,1,1,1,0,1,1,1,1,1,0,1,0,1,0
0,0,0,1,0,0,0,1,0,0,0,0,1,0,0,0,1,0,0,0,1,0
0,1,1,1,1,1,0,1,0,1,1,1,1,1,0,1,0,1,1,1,0,1,0
0,0,0,1,0,0,0,0,1,0,0,0,1,0,1,0,1,0,1,0,1,0
0,1,1,1,1,1,0,1,1,1,1,1,0,1,0,1,0,1,0,1,1,1,0
0,1,0,1,0,1,0,0,0,0,0,1,0,0,0,1,0,1,0,0,0,0,0
0,1,0,1,0,1,1,0,1,1,1,1,1,0,1,1,1,0,1,0,1,1,1,0
0,0,0,0,0,0,0,1,0,0,0,0,0,0,0,0,0,0,0,1,0
0,1,0,1,1,1,1,1,1,1,1,1,1,1,1,1,1,1,1,0,1,0
0,1,0,0,0,1,0,0,0,0,0,1,0,1,0,0,0,0,0,1,0,1,0
0,1,0,1,1,1,1,1,0,1,0,1,0,1,0,1,1,0,1,0,1,0
0,1,0,0,0,0,0,0,0,1,0,0,0,1,0,1,0,0,0,0,0,1,0
0,1,1,1,0,1,1,1,1,1,1,1,1,1,1,0,1,1,1,1,0,1,0
0,1,0,1,0,1,0,0,0,0,0,1,0,0,0,1,0,0,0,0,0,1,0
0,1,0,1,1,1,0,1,1,1,1,1,1,1,0,1,1,1,0,1,0
0,0,0,1,0,0,0,1,0,0,0,0,0,1,0,0,0,1,0,0,0,0,1,0
0,1,1,1,0,1,1,1,0,1,1,1,0,1,1,1,1,1,1,1,0
0,0,0,0,0,1,0,0,0,1,0,1,0,1,0,0,0,1,0,0,0,1,0
0,1,1,1,1,1,1,1,0,1,0,1,0,1,1,1,1,1,0,1,0,G,1,1,0
0,0,0,0,0,0,0,0,0,0,0,0,0,0,0,0,0,0,0,0,0,0
```

0 represents obstacles. 1 represents free spaces. G is the goal. A is your current position in the maze. Your current location in the maze is row, column = (1, 1). The goal location is row, column = (21, 19). Think step-by-step about how to reach the goal. What action do you take next?

**ASSISTANT:** ...

.
.
.

**Prompt 31: Corsi block tapping (text)**

**USER:** You are playing the Corsi board tapping game. The board is represented as a two dimensional array. The array contains empty locations (marked as 0) and 7 box locations (marked as B). You will be shown an ordered sequence of 6 arrays, representing a sequence of taps on the board. At each step of the sequence, one of the boxes is tapped, and it the tap is highlighted by marking the box as a T instead of B. You must remember the sequence of taps by remembering which exact boxes were tapped. Here is the sequence of arrays.

```
0,0,0,0,0,B,0
0,0,B,0,0,0,0
0,0,0,0,0,0,0
0,B,0,0,0,0,0
B,0,0,T,0,0,B
0,0,0,0,0,0,0
0,0,B,0,0,0,0

0,0,0,0,0,B,0
0,0,B,0,0,0,0
0,0,0,0,0,0,0
0,B,0,0,0,0,0
B,0,0,B,0,0,T
0,0,0,0,0,0,0
0,0,B,0,0,0,0

0,0,0,0,0,T,0
0,0,B,0,0,0,0
0,0,0,0,0,0,0
0,B,0,0,0,0,0
B,0,0,B,0,0,B
0,0,0,0,0,0,0
0,0,B,0,0,0,0

0,0,0,0,0,B,0
0,0,B,0,0,0,0
0,0,0,0,0,0,0
0,B,0,0,0,0,0
B,0,0,B,0,0,B
0,0,0,0,0,0,0
0,0,T,0,0,0,0

0,0,0,0,0,B,0
0,0,T,0,0,0,0
0,0,0,0,0,0,0
0,B,0,0,0,0,0
B,0,0,B,0,0,B
0,0,0,0,0,0,0
0,0,B,0,0,0,0

0,0,0,0,0,B,0
0,0,B,0,0,0,0
0,0,0,0,0,0,0
0,B,0,0,0,0,0
T,0,0,B,0,0,B
0,0,0,0,0,0,0
0,0,B,0,0,0,0
```

You must now identify the sequence of taps. Here is the corsi board with numbers 1 - 7 assigned on each box. Use this numbering as a reference to answer the question.

```
0,0,0,0,0,3,0
0,0,5,0,0,0,0
0,0,0,0,0,0,0
0,7,0,0,0,0,0
6,0,0,1,0,0,2
0,0,0,0,0,0,0
0,0,4,0,0,0,0
```

What is the sequence of boxes that were tapped? Here are your choices:
(1) 1, 2, 3, 4, 5, 6
(2) 1, 2, 3, 4, 6, 5
(3) 6, 5, 3, 4, 7, 1
(4) 2, 1, 5, 4, 3, 6

Think step by step. Then answer your question in the following json format.

```
{
    "answer": <fill in one of 1/2/3/4 integer value>
}
```

**Prompt 32: Corsi block tapping (vision)**

> **USER:** You are playing the Corsi board tapping game. You will be shown a video with 5 boxes (blue squares). These boxes will be tapped one at a time in a specific sequence. A tap on a box will be shown by highlighting the box in yellow. You must remember the sequence of taps by remembering which exact boxes were tapped. Here is the video.
>
> ```
> <IMAGE 1>
> <IMAGE 2>
> <IMAGE 3>
> ```
>
> You must now identify the sequence of taps. Here is an image of the corsi board with numbers on each box. Use this image as a reference to answer the question.
>
> ```
> <IMAGE WITH NUMBERS ON BOXES>
> ```
>
> Here are your choices:
> (1) 1, 4, 0
> (2) 1, 0, 4
> (3) 0, 1, 4
> (4) 1, 0, 2
>
> Think step by step. Then answer your question in the following json format.
>
> ```
> {
>     "answer": <fill in one of 1/2/3/4 integer value>
> }
> ```

**Prompt 33: Spatial addition (vision)**

> **USER:** You are playing the array addition game. You have to add two arrays by following certain rules. Each array location can be empty (i.e., fully white) or filled with colored circles. Empty locations represent zeros. The colors of the circles mean specific things.
> * Blue circle is a one
> * Red circle is a distrction and must be ignored (i.e., it does not contribute to the array addition)
> * White circle is a two
>
> Array addition works as follows:
> * sum of zeros must be a zero (i.e., an empty array cell)
> * sum of one and zero (or zero and one) must be one (i.e., a blue circle)
> * sum of one and one must be two (i.e., a white circle)
>
> Here is the first array.
>
> ```
> <IMAGE OF ARRAY 1>
> ```
>
> Here is the second array.
>
> ```
> <IMAGE OF ARRAY 2>
> ```
>
> What is the sum of the two arrays? Pick from one of these four choices.
>
> Choice 1
>
> ```
> <IMAGE OF CHOICE 1>
> ```
>
> Choice 2
>
> ```
> <IMAGE OF CHOICE 2>
> ```
>
> Choice 3
>
> ```
> <IMAGE OF CHOICE 3>
> ```
>
> Choice 4
>
> ```
> <IMAGE OF CHOICE 4>
> ```
>
> Think step by step. Then answer your question in the following json format.
>
> ```
> {
>     "answer": <fill in one of 1/2/3/4 integer value>
> }
> ```

**Prompt 34: Spatial addition (text)**

**USER:** You are playing the array addition game. You have to add two arrays by following certain rules. Each array location can be filled with E, B, R or W. E represent a zero. B represents a one. W represents a two. R is a distraction and must be ignored (i.e., it does not contribute to the array addition).

Array addition works as follows:
* sum of zeros must be a zero (i.e., E)
* sum of one and zero (or zero and one) must be one (i.e., B)
* sum of one and one must be two (i.e., W)

Here is the first array.

```
E,E,R,E,E,E
E,E,E,E,B,E
E,E,E,E,E,E
E,E,E,E,E,E
B,E,B,E,E,E
E,E,B,E,E,B,E
E,E,E,E,E,E
```

Here is the second array.

```
E,E,E,E,E,E
E,E,E,E,B,E
E,E,E,E,E,E
E,R,E,E,E,E
E,E,E,E,B,E
E,E,B,E,E,B,E
B,E,E,E,E,E
```

What is the sum of the two arrays? Pick from one of these four choices.

Choice 1:

```
E,E,E,E,E,E
E,B,E,E,B,B
E,E,E,E,E,B
E,E,E,E,E,E
B,E,B,E,B,E
E,E,B,E,E,B,E
E,E,E,E,B,E
```

Choice 2:

```
E,E,E,E,E,E
E,E,E,E,W,E
E,E,E,E,E,E
E,E,E,E,E,E
B,E,B,E,B,E
E,E,W,E,E,W,E
B,E,E,E,E,E
```

Choice 3:

```
E,E,E,E,E,E
B,E,E,E,B,E
E,E,E,E,E,E
E,E,E,E,E,E
B,E,W,B,E,E
B,E,B,E,B,B
E,E,E,E,E,E
```

Choice 4:

```
E,E,E,E,E,B
E,E,E,E,B,E
B,E,E,B,E,E
E,E,E,E,E,E
B,B,B,E,E,E
E,E,B,E,E,W,E
E,E,E,E,E,E
```

Think step by step. Then answer your question in the following json format.

```
{
    "answer": <fill in one of 1/2/3/4 integer value>
}
```

**Prompt 35: Cambridge spatial working memory (vision)**

**USER:** You are playing the Cambridge Spatial Working Memory game. You will be shown a screen with blue boxes. A treasure is hidden in one of the blue boxes. You must identify the box containing the treasure, which is shown as an yellow square. Once you find a treasure, it will be collected and placed in the "Treasures collected" section shown below the image. A new treasure will be hidden in one of the other boxes where the treasure did not appear before. You must again find the new treasure. This process is repeated till you find all treasures placed in each of the blue boxes once. Note: The treasure will never appear in a box where it had already been placed. Each turn, there are randomly selected numbers associated with each box. These numbers are meant to aid you with communication, i.e., specify what box you want to open in that turn. However, these numbers will change after every turn. So do NOT associate boxes with numbers over the long term. The number identity of a box can change any time. Therefore, you must remember the boxes based on their spatial positions and not the numbers.

RESPONSE FORMAT:
Think step-by-step about where the treasure might be based on your past actions. After that, indicate the box you want to open in the following json format:

```
{
    "action": <box integer index>
}
```

**ASSISTANT:** ...

**USER:** Here is the current state of the game. You must find the next treasure. Note that the numbers of the boxes have changed, but the box locations are fixed. Decide which box you want to open next, and then use the number associated with the box as the action.

```
<IMAGE OF SCREEN>
```

**ASSISTANT:** ...

**Prompt 36: Cambridge spatial working memory (text)**

**USER:** You are playing the Cambridge Spatial Working Memory game. You will be shown an array with integers. 0 represents empty locations. Locations numbered 1 - 9 represent boxes. A treasure is hidden in one of the boxes. You must identify the box containing the treasure. Once you find a treasure, the location will be momentarily shown as a "T" indicating that the treasure was found. The treasure is then collected and a new treasure will be hidden in one of the other boxes where the treasure did not appear before. You must then find the new treasure. This process is repeated till you find all treasures placed in each of the boxes once. Note: The treasure will never appear in a box where it had already been placed.

While the boxes are represented using integers from 1 - 9, the true identity of the box is its location (row, column) in the array. The box location is always fixed (i.e., the boxes will not move and the number of boxes will not change). However, each turn, the integer id associated with the box will change randomly. These integer ids are meant to aid you with communication, i.e., specify what box you want to open in that turn. However, these numbers will change after every turn. So do NOT associate boxes with numbers over the long term. The number identity of a box can change any time. Therefore, you must remember the boxes based on their spatial positions and not the numbers.

RESPONSE FORMAT:
Think step-by-step about where the treasure might be based on your past actions. After that, indicate the box you want to open in the following json format:

```
{
    "action": <box integer index>
}
```

**ASSISTANT:** ...

**USER:** Here is the current view of the board. You must find the next treasure. Note that the numbers of the boxes have changed, but the box locations are fixed. Decide which box location you want to open next. Then provide the number associated with the box as the action.

```
0,0,0,0,0,0,0
0,0,0,0,0,3,0
0,0,0,0,0,0,0
0,0,0,0,0,0,1
0,0,0,0,0,0,0
0,0,0,0,0,0,0
0,0,0,2,0,0,0
```

Number of treasures collected: 0 / 3

**ASSISTANT:** ...

