# OpenReview forum: "Does Spatial Cognition Emerge in Frontier Models?"
_ICLR.cc/2025/Conference — ICLR 2025 Poster_

### Official Review · Reviewer_HHQw · 2024-10-27

**Soundness:** 4
**Presentation:** 3
**Contribution:** 3
**Rating:** 6
**Confidence:** 4

**Summary:**

The paper a comprehensive benchmark SPACE that systematically evaluates spatial cognition in large language models (LLMs) and vision language models (VLMs). SPACE divides into large-scale spatial cognition tasks and small-scale spatial cognition tasks. Large-scale spatial cognition is at the environment level and the goal is to test whether LLMs and VLMs can build spatial representations to reason about and navigate in the environment. Small-scale spatial cognition tasks test models' ability to perceive, imagine, and mentally transform objects or shapes in 2D and 3D scenarios. The experimental results on frontier models (e.g. LLMs, VLMs) show that these frontier models has much lower capability in spatial cognition when compared to humans, and suggest frontier models that have acquired advanced skills may have a fundamentally different form of intelligence from humans and animals because of the failure in basic spatial cognition.

**Strengths:**

**Originality**
- While there's an increasing interest in spatial cognition for frontier models, there are not enough datasets to evaluate them systematically. The proposed benchmark SPACE is novel in terms of its comprehensiveness and diversity in tasks.

**Quality**
- SPACE is a very comprehensive and diverse benchmark. The large-scale spatial cognition tasks cover 5 tasks. The small-scale spatial cognition tasks cover 10 tasks. All 15 tasks are based on prior studies in cognitive science, which ensures plausibility in psychology and cognitive science domains. The input of the tests can be pure language or multimodal, which makes the benchmark suitable for evaluation on LLMs and VLMs. All experiments have human performance as the reference, which makes it easier to compare frontier models with humans.

**Clarity**
- The paper is well-written and easy to follow. Figures are helpful for understanding each task. The gray coloring of the results indicating they are below 50% of human performance makes the results very interpretable.

**Significance**
- This paper is valuable to the research community of large language models since it reveals the shortcoming of basic spatial cognition for these frontier models. Additionally, it could make the research community in both artificial intelligence and cognitive science rethink about the difference in intelligence between humans and these models. More specifically, it may indicate only casual language modeling will not lead to the human intelligence without sufficient embodiment in the training process. On the other hand, the results show that while these frontier models are surpassing humans on many tasks, but still lack basic cognition on simple tasks. These findings are insightful as researchers can study (1) How do models skip basic cognition to reach higher cognition? (2) Does skipping basic cognition make the frontier model unreliable and untrustworthy?

**Weaknesses:**

**Lack of experiments to demonstrate prompts can be understood by frontier models**
- Most results are around chance level accuracy. There are two possible conclusions:
1. Models have weak spatial cognition.
2. Frontier models have better spatial cognition internally, but it doesn't show externally because of the unfamiliarization of the multimodal and text-only prompt formats.

It would be more rigorous if there are some experiments show that these formats of prompting can be understood by these frontier models (e.g. achieving high accuracy in some easier tasks with the same format of prompting).

**Unclear use of Chain-of-Thought (CoT) techniques in prompting**
- Chain-of-Thought (CoT) prompting techniques [1] are known for improving language model in solving tasks as it allows LLMs to have some reasoning steps before arriving at the final answer. CoT techniques are not mentioned in the main paper, except "Think step by step" prompts in the appendix, but it is unclear which tasks use the CoT, which doesn't. If some tasks use CoT, it should be listed separately to the results table as a comparison to investigate whether reasoning steps strengthen the frontier models.

*References*

[1] Wei, Jason, et al. "Chain-of-thought prompting elicits reasoning in large language models." Advances in neural information processing systems 35 (2022): 24824-24837.

**Questions:**

**Questions**
- The Discussion Section mentions that spatial cognition is considered to be a prerequisite for higher cognition and frontier models can acquire tasks requiring higher cognition like olympiad geometry problems, but lack basic spatial cognition as indicated in the results. Is there any study in psychology or cognitive science show that basic spatial cognition is related to solving geometry problems?

- Trinh et al [1] trained the neural language model for geometry theorem proving from scratch using their large-scale synthetic data to guide the symbolic deduction engine. Therefore it seems not very clear that whether their model can perform better on spatial cognition tasks with language-heavy prompting because their model may be specialized to theroem proving while lacking language understanding and generation skills when comparing to state-of-the-art language models.

*References*

[1] Trinh, Trieu H., et al. "Solving olympiad geometry without human demonstrations." Nature 625.7995 (2024): 476-482.

---

> ### Author Response · Authors · 2024-11-24
> **Response from authors to reviewer HHQw**
>
> We thank the reviewer for the positive comments and the valuable feedback.
>
> **Do frontier models fail due to the lack of familiarity with prompts or because they have weak spatial cognition?**
> Our experimental evidence indicates that the failures are most likely due to weak spatial cognition. Please see the common response.
>
> **Unclear use of Chain-of-Thought (CoT) techniques in prompting**
> As illustrated in the Appendix and as noted by the reviewer, we add “Think step by step” uniformly for all task prompts. This is a form of zero-shot chain-of-thought as proposed in Kojima et al., NeurIPS 2022. Since we benchmark on 15 tasks, we have not found a way to provide more detailed chain-of-thought prompts in a task-agnostic way. It is possible to guide the model to solve specific tasks through step-by-step instructions, but this requires providing human prior knowledge about the task solving techniques and may not evaluate the cognitive competency of the models that we’re interested in.
>
> Kojima, Takeshi, et al. "Large language models are zero-shot reasoners." Advances in neural information processing systems 35 (2022): 22199-22213.
>
> **Is there any study in psychology or cognitive science show that basic spatial cognition is related to solving geometry problems?**
> Yes, there are several studies looking into the importance of spatial abilities for success in geometry and more broadly, in STEM).  We have cited a couple of representative studies in the paper (L041) and have provided a larger list of studies below:
>
> Shea, Daniel L., David Lubinski, and Camilla P. Benbow. "Importance of assessing spatial ability in intellectually talented young adolescents: A 20-year longitudinal study." Journal of Educational Psychology 93.3 (2001): 604.
> Kozhevnikov, Maria, Michael A. Motes, and Mary Hegarty. "Spatial visualization in physics problem solving." Cognitive science 31.4 (2007): 549-579.
> Wai, Jonathan, David Lubinski, and Camilla P. Benbow. "Spatial ability for STEM domains: Aligning over 50 years of cumulative psychological knowledge solidifies its importance." Journal of educational Psychology 101.4 (2009): 817.
> Newcombe, Nora S. "Picture this: Increasing math and science learning by improving spatial thinking." American educator 34.2 (2010): 29.
> Young, Christopher J., Susan C. Levine, and Kelly S. Mix. "The connection between spatial and mathematical ability across development." Frontiers in psychology 9 (2018): 755.
>
> **Does AlphaGeometry, a model trained for geometric theorem proving, perform better on spatial cognition tasks?**
> This would be an interesting study for future work. If a more general-purpose LLM/VLM had similar capabilities as AlphaGeometry, we would be very curious to see if it performs better on spatial cognition tasks.

---

> ### Comment · Reviewer_HHQw · 2024-11-26
>
> Thanks for addressing my concerns. I increased the soundness and maintained the rating which inclined to acceptance of this paper.

---

### Official Review · Reviewer_61bg · 2024-11-02

**Soundness:** 3
**Presentation:** 3
**Contribution:** 3
**Rating:** 8
**Confidence:** 4

**Summary:**

This work develops SPACE, a benchmark for systematic evaluation of spatial cognition in foundation models, vision-language or language only. These are a large variety of tasks designed to test spatial cognition taken from various places within the cognitive science literature. They separated the tasks into two categories - "large-scale" which are testing the development of a robust cognitive map of an entire environment and "small-scale" which are testing targeted spatial relationships between specific objects in various contexts (mental rotation, putting pieces of a board together, spatial addition, etc). The large-scale tasks have an egocentric visual representation of the task, an allocentric "birds eye view" visual representation, and an allocentric text-only representation. The small-scale tasks have a vision vs. text version of the tasks. They evaluate various popular closed models (e.g. GPT4v) as well as various open-source models (Llama3), both multimodal and language-only models. Performance was generally above chance but far below human performance.

**Strengths:**

* Very timely topic.

* It is a quite impressive set of tasks that really engaged with the literature on spatial reasoning. The translation of spatial reasoning tasks to text only is an especially nice contribution, because it allows for the evaluation on text-only models. I think the benchmark will be immediately valuable to the community.

* Comprehensive evaluation. Lots of models were used, which really gives us an idea as to how the current landscape of models performs at tasks like these.

**Weaknesses:**

1. The authors took the best first step in evaluating a capability that humans have (spatial cognition), which is to directly take tasks that have been used in cognitive science to test this capability in humans. However, these are *tests that were designed for humans specifically*, not Large (vision)-Language Models. And I think this would definitely have implications on the results. For example, vision encoders like CLIP are trained on a lot of ImageNet-style object image data and so a model using CLIP as a visual encoder may be less used to, for example, the egocentric observations or BEV images used in the large-scale studies. I don't know if the specific composition of GPT4v's or GPT4o's image training data is public knowledge, but I think the larger point still stands. This doesn't mean what the authors did is incorrect - it is a good first step and valuable contribution towards understanding spatial capabilities in these models. However, I think more work will need to be done in the future to *ensure these kinds of tasks are presented to the model in a way that is ecological.* This is really common in animal behavioral paradigms. Often when animal researchers are testing a capability known to exist in humans, the original paradigms are modified so that the task makes enough sense to the animal and is actually testing the desired capability. For me to raise the score above threshold, *I would like the authors to engage with this point a little more than what's currently written in the paper.*

2. I think this is a more minor weakness. There are a very large number of tasks and each of them are described independently. However, there isn't much discussion on what features are shared across the different tasks. I think the paper would be a lot more stronger if the authors were able to pick out specific features of spatial cognition that are salient across many of these tasks and identify which tasks test which features. If the paper was structured more like: "we identified 4 features of spatial cognition common to many tasks in the literature: A, B, C, and D. Here are some tasks from the literature. The mental rotation task tests A and B, perspective taking tests B and C, etc." Currently the shear number of tasks (while impressive) is overwhelming, and looking at performance across all of them is harder to interpret. I think this would make evaluation on SPACE more digestible, because you'd be able to diagnose which aspects of spatial cognition your model succeeds at and which aspects your model fails at.

**Questions:**

* Related to weakness 1, have you tried egocentric text representations? For a given location and direction of the environment, you could have a way to parse the egocentric image observation into a textual description (e.g. "there is a landmark with a picture of a dog on the left wall and a deadend wall directly in front of you. On both your left and right sides, there are potential paths leading elsewhere"). This kind of representation may be more ecological to some of these models. This could apply to the small-scale tasks (corsi block tapping,  spatial addition, cambridge spatial working memory) with grid inputs as well (e.g. "Row 1 column 2 has a blue, row 2 column 5 has a red,...") Even if it's a small change in how the input is represented, a change in performance would be very informative.

---

> ### Author Response · Authors · 2024-11-24
> **Response from authors to reviewer 61bg**
>
> We thank the reviewer for the positive comments and valuable feedback.
>
> **How do we ensure the tasks are presented to the model in a way that is ecological?**
> Please see the common response.
>
> **Can you structure the paper to organize tasks based on common spatial cognition features?**
> All large-scale experiments (except route retracing) are currently evaluating cognitive mapping and planning (Tolman, 1948), i.e., the ability to build spatial representations of the environment and use them to reason about and navigate in environments (Section 3.1). Route retracing evaluates the ability to build route maps, i.e., the ability to memorize a route and follow the route in future trials (narrow strip maps from Tolman, 1948).
>
> The small-scale spatial cognition experiments evaluate the following cognitive abilities: spatial perception, spatial visualization, spatial orientation, selective spatial attention and visuospatial working memory (Section 3.2). **Spatial perception** is the ability to perceive spatial information in visual inputs (e.g., horizontal nature of water surface in the water level test and angles between lines in the judgement of line orientation test). **Spatial visualization** is the ability to perform multi-step mental manipulations of 2D or 3D stimuli (mental rotation test and minnesota paper form board test). **Spatial orientation** is the ability to imagine being in a different position in space and seeing the surroundings from the new perspective (perspective taking test and maze completion task). **Selective spatial attention** is the ability to selectively attend to a particular region of space while ignoring others (selective attention task and corsi block-tapping test). **Visuospatial working memory** is the ability to store visual information in memory and manipulate them to solve tasks (corsi block-tapping test, spatial addition test and cambridge spatial working memory test). We have updated Figure 3 to reflect this organization.
>
> Tolman, Edward C. "Cognitive maps in rats and men." Psychological review 55.4 (1948): 189.
>
> **Have you tried egocentric text representations?** Yes, we tried similar experiments during the early stages of our work. To allow LLMs to perform visual navigation tasks, we tried using GPT-4v to describe images and use the descriptions as inputs for an LLM to navigate. However, we abandoned this approach because there was significant perceptual aliasing due to the fine-grained nature of the action space in large-scale experiments. For example, turning left/right by 30 degrees or moving forward by 0.25m does not change the textual description in most cases.
>
> We also tried a version of the text-only experiments where we described the elements of the array instead of providing a comma-separated 2D array. However, we did not notice significant performance differences between the two approaches. Moreover, the text-based descriptions significantly increased the context length of the LLM, the models to exceed their maximum context length. For larger arrays, this could also cause performance degradations associated with very long contexts for LLMs (Liu et al., 2024).
>
> Liu, Nelson F., et al. "Lost in the middle: How language models use long contexts." Transactions of the Association for Computational Linguistics 12 (2024): 157-173.

---

> > ### Comment · Reviewer_61bg · 2024-11-25
> >
> > Thanks for incorporating my feedback. I raised my score.

---

### Official Review · Reviewer_Mbyv · 2024-11-03

**Soundness:** 2
**Presentation:** 2
**Contribution:** 2
**Rating:** 8
**Confidence:** 4

**Summary:**

A benchmark is introduced for evaluating spatial cognition in LLMs and VLMs, comprising both large-scale spatial reasoning tasks (e.g., navigation) and small-scale spatial reasoning tasks (e.g., mental rotation). The results indicate that both VLMs and LLMs perform well below human participants on these tasks, indicating weak spatial reasoning abilities in these models.

**Strengths:**

- The proposed benchmarks are very extensive, covering different types of spatial cognition with many tasks and input modalities.
- The paper is well written and many details are provided on the evaluation methodology, including detailed prompts and model settings.
- The topic investigated in this paper is an important one, with many implications for real-world use of foundation models.

**Weaknesses:**

Although the benchmark appears to be well designed, there are a number of previous works that address very similar questions, and the results are not particularly surprising. Both Valmeekam et al. (2023) and Momennejad et al. (2023) have shown that LLMs are very poor at navigation and planning tasks (i.e., large-scale spatial cognition), and studies from both Yamada et al. (2024) and Ivanova et al. (2024) and have already shown that LLMs have a limited ability to reason about spatial relations. These works are not cited, and there is no discussion of how the proposed benchmark goes beyond previous studies of spatial reasoning in these systems.

Valmeekam, K., Marquez, M., Olmo, A., Sreedharan, S., & Kambhampati, S. (2024). Planbench: An extensible benchmark for evaluating large language models on planning and reasoning about change. Advances in Neural Information Processing Systems, 36.

Momennejad, I., Hasanbeig, H., Vieira Frujeri, F., Sharma, H., Jojic, N., Palangi, H., ... & Larson, J. (2024). Evaluating cognitive maps and planning in large language models with CogEval. Advances in Neural Information Processing Systems, 36.

Yamada, Y., Bao, Y., Lampinen, A. K., Kasai, J., & Yildirim, I. (2023). Evaluating spatial understanding of large language models. arXiv preprint arXiv:2310.14540.

Ivanova, A. A., Sathe, A., Lipkin, B., Kumar, U., Radkani, S., Clark, T. H., ... & Andreas, J. (2024). Elements of World Knowledge (EWOK): A cognition-inspired framework for evaluating basic world knowledge in language models. arXiv preprint arXiv:2405.09605.

**Questions:**

Previous work on spatial reasoning in LLMs should be cited, and the paper should address how the proposed benchmark goes beyond the already existing benchmarks, and what new information we have learned about spatial reasoning in LLMs.

---

> ### Author Response · Authors · 2024-11-24
> **Response from authors to reviewer Mbyv**
>
> We thank the reviewer for their valuable feedback and for sharing these references.
>
> **How does the proposed benchmark go beyond already existing benchmarks like Valmeekam et al., 2024, Momennejad et al., 2024, Yamada et al., 2023 and Ivanova et al., 2024?**
>
> Thank you for the references! They are relevant and we have incorporated them into our paper in the “Spatial reasoning in large language models” section in related works. There are two high-level differences that distinguish SPACE from all of these papers.
>
> * Prior work evaluates cognitive mapping and planning abilities, which are related to the large-scale spatial cognition tasks,  maze completion task and perspective taking tasks from SPACE.  SPACE evaluates spatial cognition, which entails a broader umbrella of cognitive abilities than prior work (L097 - L111). We additionally evaluate the following abilities (L291 - L298)
>     * spatial perception (water level test, judgement of line orientation test)
>     * spatial visualization (mental rotation test, minnesota paper form board)
>     * selective spatial attention (selective attention task, corsi-board tapping task)
>     * visuospatial working memory (corsi-board tapping task, spatial addition task, cambridge spatial working memory task).
> * SPACE aims to mimic classical animal cognition experiments. These are primarily visual in nature. Accordingly, SPACE provides multimodal task presentations that are compatible with VLMs. However, we also provide text-only translations of the multimodal tasks to evaluate LLMs that do not support the vision modality. Additionally, the parallel multimodal and text-only presentations of tasks allows us to compare the text and vision capabilities of VLMs on the same task (L474 - L480).
>
> These are value-adds in SPACE when compared to the papers mentioned above. Here is a more detailed discussion of the similarities and differences between the tasks in SPACE and prior work.
>
> * PlanBench (Valmeekam et al.) and CogEval (Momennejad et al.) evaluate LLMs on text-based planning tasks such as navigation, delivery logistics planning and block stacking. The navigation tasks from CogEval are related to SPACE’s large-scale spatial cognition tasks like route retracing and novel shortcuts. CogEval samples graph structures with varying levels of connectivity, where each node is a room (potentially containing an object) and edges represent the connectivity between rooms. This simulates high-level planning and navigation and is unlike what animals / humans deal with when performing the same tasks in the animal cognition literature. We replicate fine-grained navigation with egocentric and birds-eye view observation spaces akin to classical cognitive science experiments.
> * Yamada et al. presents spatial reasoning tasks based on map traversals. Specifically, they create maps/graphs of certain shapes (squares, triangles, circles, etc.) and simulate a traversal through the graph, where each node contains a unique object.  After the traversal, the model is expected to identify the object at the current location (i.e., self-localization). This task is related, but complementary, to SPACE’s large-scale spatial cognition tasks. SPACE focuses on perspective-taking (i.e., if you are at X, where is Y?) and map sketching abilities. Yamada et al. design abstract graph domains intended to evaluate text models and present the inputs in a narrative story form. In contrast, we mimic observation spaces and environments that were designed for animal cognition, and evaluate both text and multimodal models.
> * EWOK (Ivanova et al.) study spatial plausibility reasoning (e.g., given the context “The piano is in front of A. A turns left.”, is it plausible to say “The piano is right of A”?). On the other hand, we evaluate complementary spatial cognitive abilities such as cognitive mapping / spatial orientation, spatial perception, spatial visualization, selective spatial attention and visuospatial working memory (L196, L291 - 298).

---

> > ### Comment · Reviewer_Mbyv · 2024-11-25
> >
> > Thank you to the authors for this response. All of my concerns have been addressed. I have also read the other reviews and author responses, and it seems that the concerns raised in other reviews were addressed fairly well, though not all reviewers have responded. I will raise my score and vote for acceptance.

---

### Official Review · Reviewer_pZuW · 2024-11-04

**Soundness:** 2
**Presentation:** 2
**Contribution:** 3
**Rating:** 5
**Confidence:** 4

**Summary:**

This paper proposes the SPACE Benchmark, designed to evaluate spatial cognition capabilities in large models by focusing on both large-scale and small-scale spatial cognition. Large-scale spatial cognition pertains to environmental reasoning, where the environment stays constant while the viewer’s perspective shifts. Conversely, small-scale spatial cognition involves the perception and imagery of changing spatial states of objects, with a fixed viewer’s perspective. The benchmark comprises 15 tasks distributed across image-based and text-based experiments, designed to test VLMs and LLMs. Each task contains questions between 30 and 100+. The findings indicate that current models underperform relative to human-level reasoning across various spatial tasks, suggesting that these models have yet to achieve robust spatial reasoning capabilities. Notably, some models, such as GPT-4 and GPT-4v, demonstrate superior performance in text-only tasks compared to multimodal tasks.

**Strengths:**

1. It is interesting to test both VLM and LLM models’  large scale 2D and 3D spatial reasoning abilities using three scenarios: Ego Image/ Video, Bird’s Eye View (BEV) images, and BEV text. It also allows for the assessment of models across different modalities, including textual descriptions, single images, and video content. This design reflects the complexity of real-world spatial understanding and demonstrates the comprehensive nature of the evaluation.
2. This article uses interactive tasks for route retracing and shortcut searching where the model receives the current observation, decides which action to take, and receives corresponding updated observations

**Weaknesses:**

A. Providing specific details such as the resolution of the videos and images used in the experiments would be better because models might take in different sizes of input images, therefore model performance can vary significantly based on input resolution. if an image is downsampled incorrectly, it may disrupt the original aspect ratio, potentially impairing the performance in tasks that require precise spatial reasoning, such as distance and direction estimation. This issue might partially explain why LLMs outperform VLMs in some scenarios.

To better explain this, could try to
	1. Specify the exact resolutions used for images and videos in their experiments
	2. Discuss how they handled any resizing or aspect ratio changes when preparing inputs for different models
	3. Consider running an ablation study with different image resolutions to quantify the impact on performance, especially for tasks like distance estimation

B. Only two VLMs are tested for image relevant tasks. And those two models even belong to the same family (GPT). I would recommend to test more SOTA VLMs to validate your conclusions.
C. As I stated in the following questions, the images used for experiments might be too small to draw a reliable conclusion.

**Questions:**

A. In the direction estimation task, take Figure 6 for example, pretend that you are standing below landmark F and facing F, while N is right above F. Why does the angle become 78 degrees? The two landmarks seem to be almost on the same line.

B. The article omits some details of data processing and requires greater clarification. For instance:
	1. In the section on Large-Scale Spatial Cognition, while the Bird’s-Eye View (BEV) is mentioned as covering 2.5m x 2.5m surrounding area, the dimensions of both the 2D and 3D views,  the duration of videos as well as number of sampled images of each video for the Ego image are not provided. It might affect the models' performance if the sampled images are over dense and include too much redundancy or images are too sparse therefore missing key frames containing landmarks.
        2. For BEV text, the authors state that they “carefully select the encoding to ensure compatibility with text tokenizers” but do not detail this process. How is this encoding achieved, and how is the dimension of the text array determined? It would be helpful to know how many different array dimensions were tested to understand the effect of text encoding choices. Additionally, since LLMs reportedly perform better with BEV text, could this be due to the much smaller dimensionality of text arrays compared to BEV images?
	3. In Table 3, only the number of questions is presented per task, without image counts. Although most tasks include 30-50 questions, there appear to be fewer than 10 images for some tasks, which limits the robustness of any conclusions drawn. To strengthen the study’s validity, I recommend providing the exact image counts per task and ensuring a sufficient number of images for each task. This would enhance the reliability of conclusions and support a more balanced dataset for each task.

---

> ### Author Response · Authors · 2024-11-24
> **Response from authors to reviewer pZuW [Part 1 / 2]**
>
> We thank the reviewer for their valuable feedback. As we understand it, the reviewer has two main concerns:
>
> **Concern 1:**  The image and video inputs may not be provided to the model correctly, resulting in poor results.
>
> Please see common response above.
>
> **Concern 2:**  We only evaluate two VLMs for image relevant tasks.
>
> Please note that we evaluated five VLMs in our submission (see Table 2). We evaluate two closed-source models (GPT-4o and GPT-4v), and three state-of-the-art open-source models (Pixtral 12B, Phi 3.5 vision and Llava interleave 7B). Among these models, only GPT-4o and GPT-4v support evaluating on long videos (up to 240 frames for ego videos) as noted in L418 - 420. We now additionally include results from Claude 3.5 Sonnet on small-scale spatial cognition tasks in Table 2, bringing the total number of VLMs to six (see updated paper). It obtains results similar to GPT-4o on majority of the tasks. It performs considerably better at the multimodal versions of SAtt and CSWM, and on the text-only version of CBTT. We were unable to evaluate Claude 3.5 Sonnet on large-scale spatial cognition tasks due to API failures on requests with long videos.
>
> We address other questions below.
>
> **What image resolutions / aspect ratios do we use?** For majority of our experiments, we use square images. The image resolution and aspect ratios are task-dependent. They are listed in Table 5 in the updated paper. Note that our human baseline uses the exact same images and resolutions for each task.
>
> **How do we handle aspect ratios?** We provide the images to models as is without pre-processing. For most models (especially closed-source ones), the processing of the image beyond the input stage is outside our control. We rely on the model creators to correctly process the images. As the reviewer points out, rescaling images can affect metric distance estimation in both 2D and 3D. Our tasks are designed to be independent of the image resolution, and require only relative judgements (including the distance estimation task as noted in L210 - 217). We do not change other properties, such as aspect ratio.
>
> **What are the durations of videos and number of sampled images?** The video duration depends on the path taken and whether the videos are of BEV images or ego images.
>
> For BEV videos, a video frame is generated after every step (i.e., a one-step movement in the grid). Landmarks are clearly visible in this case and cannot be missed. The number of frames in a video frames from 13 to 72. We do not subsample the video frames when providing them to the model.
>
> For ego videos, a video frame is sampled after every navigation action (i.e., turn-left, turn-right, move-forward). Following prior navigation literature (Savva et al., ICCV 2019), we use a forward movement step of 0.25m and turn angle of 30 degrees. This sufficiently fine-grained action space keeps the number of frames manageable (ranges from 61 - 240), while ensuring that the landmarks are not missing from the video. We further ensure that landmarks are not missed by sampling the trajectories as follows. To visit a landmark, the agent navigates close to the landmark (within 1m) and turns around to face it. When looking at it, we stop the agent for N=10 steps at the landmark to ensure it is prominently visible in the video for a longer time. The agent then moves away to visit a new landmark. We include some examples of walkthrough videos in the updated supplementary materials zip.  Since GPT-4o and GPT-4v were unable to process 240 frames at once, we subsampled the videos by a factor of two to evaluate them on our benchmark.
>
> Savva, Manolis, et al. "Habitat: A platform for embodied ai research." Proceedings of the IEEE/CVF international conference on computer vision. 2019.
>
> We have clarified our image pre-processing details in Appendix A.3 in the updated paper.
>
> **How do we carefully select the encoding to ensure compatibility with text tokenizers?** We use comma-separated arrays of characters or four-letter words. Please see the common response for more details.
>
> **How is dimension of text array determined? Does this affect the tokenization?** The array size varies from 3x3 to 20x20 for all tasks other than Maze Completion Task (MCT). The largest array size for MCT was 31x31. No, the dimensions of the 2D text array do not affect the tokenization.

---

> ### Author Response · Authors · 2024-11-24
> **Response from authors to reviewer pZuW [Part 2 / 2]**
>
> **Do LLMs perform better with BEV text than VLMs with BEV images because text arrays are smaller than BEV images?**
> From Table 1, GPT4o and GPT4v achieve average scores of 28.8 and 25.9, respectively, on BEV image, and 32.6 and 27.6, respectively, on BEV text. While these differences are non-trivial, they’re still too small to arrive at the above conclusion. Furthermore, the image tokenization process is significantly different from the text tokenization process. So, we cannot compare these processes directly.
>
> **What is the number of images per task in Table 3?** We have updated the table (now Table 6 in the updated paper) to include the number of images and videos generated per task.
> Large-scale spatial cognition: There are 30 videos for ego image and BEV image tasks in large-scale spatial cognition. These videos form the basis for the questions and navigation tasks. Additional images are rendered for the navigation tasks conditioned on the agent’s actions.
> Small-scale spatial cognition: There are 1557 images for the QA tasks (all tasks other than MCT and CSWM). The number of images per task varies from 20 for PTT to 300 for SAdd. Note that some tasks have multiple images for the same image (e.g., MFPB, WLT, SAtt and CBTT), while others have multiple questions for the same image (e.g., PTT and MRT).For interactive tasks like MCT and CSWM, we render images conditioned on the agent’s actions.
>
> **Why does the angle become 78 degrees in Figure 6?**
> In Figure 6, the coordinate systems of the top-down visualization (shown on the left) and the rendered BEV images (shown on the top-right) are different. The top-down visualization and the rendered BEV images are rotated versions of each other, i.e., you rotate the top-down visualization by 90 degrees clockwise to get BEV image. For the first question, the observer is standing below F in the BEV image. Correspondingly, the observer would be standing to the right of F in the top-down visualization. From the BEV image perspective, N is to the right on the same row as F. Note, that the visualization on the left is provided to the reader of the manuscript only. Models or our human baseline see only the BEV images. We have clarified this in the captions for Figures 6 and 7.

---

### Author Response · Authors · 2024-11-24
**Common response**

We thank the reviewers for their insightful feedback and questions. Reviewers appreciated the comprehensive nature of our benchmark (reviewers pZuW, Mbyv, 61bg and HHQw), the comprehensive evaluation with lots of models (reviewer 61bg), the clarity of our paper writing (reviewers Mbyv and HHQw), the importance our study to the research community (reviewer Mbyv and HHQw), and the novelty (reviewer HHQw) and timeliness (reviewer 61bg) of our benchmark.  Reviewers also raised a few concerns that we address in our responses. We have updated our paper and supplementary material to reflect the feedback from the reviewers. The changes in the paper are highlighted in blue text. We address a key concern shared by multiple reviewers below and respond to individual concerns in separate messages.

# Do models understand the visual and textual inputs in SPACE? TL; DR - Yes, they do.

A common concern was that models may fail on our benchmark because they cannot understand the inputs (i.e., visual and textual inputs may not be ecological to them), and not because they lack spatial cognition. Our experimental evidence suggests that this is not the case. We took great care in designing our benchmark to be as ecological as possible to models while maintaining the true spirit of the animal cognition experiments. For this rebuttal, we performed additional experiments to confirm that models do understand our inputs and can solve other tasks using the same inputs. Yet, they fail at tasks related to spatial cognition. We provide more details below.

**For textual experiments**, it is important that LLMs perceive each element of the input array as a separate token. We tested multiple encodings to arrive at “comma-separated” character or 4-word arrays (see Figure 9 for examples) to ensure correct tokenization of the array elements. We tested a large number of tokenizers from HuggingFace and https://gpt-tokenizer.dev/ (including all models we benchmark) to confirm tokenization works as expected. Our results in Table 2 also indicate that this encoding makes sense to state-of-the-art LLMs like Claude 3.5 Sonnet, GPT-4o, Mistral 123B, and GPT-4v. These models excel at tasks like selective attention task (SAtt), spatial addition task (SAdd) and corsi block-tapping task (CBTT), and perform well above chance on other tasks like judgement of line orientation (JLO), perspective taking test (PTT) and cambridge spatial working memory task (CSWM).

**For vision-language models**, especially closed ones like Claude 3.5 Sonnet, GPT-4o and GPT-4v, the image pre-processing and tokenization strategies and the training data distribution are unknown to us. Nevertheless, we attempted to design our benchmark to be easily understood by VLMs.

* For the large-scale spatial cognition experiments with ego images, we put in a considerable amount of effort to generate photorealistic 3D scenes and images (L242 - 252). We wanted to ensure that the rendered images are in the domain with the internet-style training data typically used for training VLMs. For this, we generated 3D scenes using PBR textures often used by game designers for realistic graphic rendering. We then used ImageNet images hung on walls as landmarks.
* For the BEV images and the small-scale spatial cognition experiments, we render images at sufficiently high resolution so that models can perceive the inputs clearly.

To evaluate whether VLMs can understand our inputs, we design additional tests unrelated to spatial cognition on the same inputs used in our benchmark. In each test, we pose a series of multiple-choice questions evaluating a model’s fine-grained understanding of the visual (and textual) inputs. These tests are described in Appendix A.1 in the updated paper. We evaluate GPT-4o and GPT-4v on these tests. The results are shown in Tables 3 and 4. Our results on these tests indicate that state-of-the-art models can understand multimodal and text-only inputs provided in our benchmark. They perform well in most of the tests that do not require spatial cognition but have a few shortcomings (e.g., GPT-4v is poor at localizing landmarks in ego images, GPT-4o and GPT-4v cannot effectively understand rotations of water containers or count unique characters / objects in a grid). Importantly, the average results on each test are much higher than the SPACE task counterparts for the same inputs. That is, for the same inputs, these models excel at non-SPACE tasks and fail at SPACE tasks. These results indicate that the failures on SPACE are most likely due to the lack of spatial cognition and not due to a lack of understanding of the visual and textual inputs.

---

### Meta-Review · Area_Chair_KnYk · 2024-12-08

**Metareview:**

This submission presents an evaluation of spatial cognition in frontier AI models through SPACE, a benchmark of 15 tasks adapted from cognitive science. The benchmark evaluates both large-scale spatial abilities like environmental mapping and small-scale capabilities like mental rotation, offering parallel visual and text-only presentations. Results show current models perform significantly below human baselines on these fundamental cognitive tasks. Responsive reviewers were positive on the submission, and I assess that the authors have substantively addressed the concerns of the last reviewer, and so I recommend acceptance.

**Additional Comments On Reviewer Discussion:**

The reviewers questioned whether poor performance indicates spatial cognition deficits or input processing difficulties, how SPACE relates to prior work, and its ecological validity. The authors demonstrated that models can process the inputs while specifically struggling with spatial reasoning, established SPACE's broader scope versus existing benchmarks, and documented how the task design balances model compatibility with experimental fidelity.

---

### Decision · Program_Chairs · 2025-01-22

Accept (Poster)